# Influence of aerosol copper on HO₂ uptake: A novel parameterized equation

Huan Song[1,2], Xiaorui Chen[1,2], Keding Lu[1,2*], Qi Zou[1,2], Zhaofeng Tan[2,3], Hendrik Fuchs[2,3], Alfred Wiedensohler[4], Daniel R. Moon[5,6], Dwayne E. Heard[5], María-Teresa Baeza-Romero[7], Mei Zheng[1], Andreas Wahner[2,3], Astrid Kiendler-Scharr[2,3], Yuanhang Zhang[1,2]

1. State Key Joint Laboratory or Environmental Simulation and Pollution Control, College of Environmental Sciences and Engineering, Peking University, Beijing, China
2. International Joint Laboratory for Regional Pollution Control, Jülich, Germany, and Beijing, China
3. Institute of Energy and Climate Research, IEK-8: Troposphere, Forschungszentrum Jülich GmbH, Jülich, Germany
4. Leibniz Institute for Tropospheric Research, 04318 Leipzig, Germany
5. School of Chemistry, University of Leeds, Leeds, LS2 9JT, UK
6. LISA, Université Paris-Est Créteil, Université de Paris, Faculté des Sciences et Technologie, 61 avenue du Général de Gaulle, 94010 Créteil Cedex, France
7. Universidad de Castilla-La Mancha, Escuela de Ingeniería Industrial y Aeroespacial de Toledo, 45071, Toledo, Spain

*Correspondence to*: Keding Lu (k.lu@pku.edu.cn)

**Abstract.** Heterogeneous uptake of hydroperoxyl radicals (HO₂) onto aerosols has been proposed to be a significant sink of HOx hence impacting the atmospheric oxidation capacity. Accurate calculation of the HO₂ uptake coefficient $\gamma_{HO_2}$ is key to quantifying the potential impact of this atmospheric process. Laboratory studies show that $\gamma_{HO_2}$ can vary by orders of magnitude due to changes in aerosol properties, especially aerosol soluble copper (Cu) concentration and aerosol liquid water content (ALWC). In this study we present a state-of-the-art model called MARK to simulate both gas and aerosol phase chemistry for the uptake of HO₂ onto Cu-doped aerosols. Moreover, a novel parameterization of HO₂ uptake was developed that considers changes in relative humidity (*RH*) and condensed phase Cu ion concentrations and which is based on a model optimization using previously published laboratory and new laboratory data included in this work. This new parametrization would be applicable to wet aerosols and it would complement current IUPAC recommendation for cloud droplets. The new parameterization is as follows:

$$\frac{1}{\gamma_{HO_2}} = \frac{1}{\alpha_{HO_2}} + \frac{3 \times v_{HO_2}}{4 \times 10^6 \times R_d H_{corr} RT \times (5.87 + 3.2 \times \ln(\text{ALWC}/[\text{PM}] + 0.067)) \times [\text{PM}]^{-0.2} \times [Cu^{2+}]^{0.65}} + \frac{v_{HO_2} l}{4RTH_{org}D_{org}\varepsilon}$$

All parameters used in the paper are summarized in Table A1. Using this new equation, field data from Wangdu campaign were used to evaluate the impact of the HO₂ uptake onto aerosols on the ROx (=OH + HO₂+RO₂) budget. Highly variable values for HO₂ uptake were obtained for North China Plain (median value <0.1).

## 1 Introduction

The atmospheric cleaning capacity of the troposphere is largely determined by the concentrations of the hydroxyl radical, which are closely linked with the concentrations of the hydroperoxyl ($HO_2$) radical. In the established chemical mechanism, the coupling of OH and $HO_2$ is strongly determined by the reaction of OH + VOCs (volatile organic compounds)/CO/HCHO/$CH_4$/$H_2$/$SO_2$ and $HO_2$ + NO (Seinfeld, 1986). The reactivity from aerosol uptake cannot compete with the known gas phase reactivity of OH, whereas it may compete with the reactivity of NO toward $HO_2$ under some conditions such as low NO (Tang et al., 2017). For high aerosol mass load, the reaction rate of $HO_2$ with aerosol particles could be fast enough to influence the concentration of HOx radicals, and consequently, reduce ozone production from $HO_2$+NO (Kanaya et al., 2009; Li et al., 2019b).

From a global perspective, the impact of $HO_2$ uptake on the calculated HOx concentrations is diagnosed to be about 10~40 % (Jacob, 2000; Whalley et al., 2015; Whalley et al., 2010; Mao et al., 2010; Li et al., 2019b; Li et al., 2019a) and often the value of $\gamma_{HO2}$ (the heterogeneous uptake coefficient (Schwartz, 1984; Schwartz, 1986)) is assumed to be a single value, 0.2 (Tie et al., 2001; Martin et al., 2003). The impact of $HO_2$ uptake is lowered when a parameterized equation of $\gamma_{HO_2}$ is used without considering the influence of transition metal ions (TMIs) (Thornton et al., 2008), however, still, a significant impact on the calculated [OH] and $O_3$ production rate are suggested for air masses over Chinese megacity areas (Macintyre and Evans, 2011). A model study (Xue et al., 2014) considering the aerosol uptake of $HO_2$ showed an impact on the simulated $HO_2$ concentrations and local $O_3$ production rates in Chinese urban regions: Beijing, Shanghai, and Guangzhou. Furthermore, researchers have proposed that in the North China Plain (Li et al., 2019a; Li et al., 2019b), the reduced $HO_2$ uptake owing to reduction of aerosol surface area is considered to be the key reason for the increased surface ozone concentration over the last few years when a value of 0.2 was used for $\gamma_{HO_2}$.

Previous studies show that the value of $\gamma_{HO_2}$ from the laboratory, field, and modeling studies spans several orders of magnitude, ranging from <0.002 for dry aerosols (Cooper and Abbatt, 1996; Taketani et al., 2008; George et al., 2013) to 0.2 for liquid deliquesced aerosols. Much higher values of $\gamma_{HO_2}$ have been measured and calculated for Cu-doped aerosols (Mozurkewich et al., 1987; Taketani et al., 2008; Thornton et al., 2008; Cooper and Abbatt, 1996; Lakey et al., 2016b; George et al., 2013). For fine particles, the reactions of $HO_2$ with soluble Cu ions may be fast enough, thus the uptake coefficient is limited by the mass accommodation coefficient $\alpha$. Due to the widespread distribution of $Cu^{2+}$ ion in ambient particles, the absence of an accurate evaluation of $\gamma_{HO_2}$ is one of the largest uncertainties for the determination of the impact of $HO_2$ uptake on pressing atmospheric issues such as ozone formation.

In this study, we reanalyzed several datasets of the aerosol uptake of $HO_2$ from laboratory studies reported in the literature, a new dataset for $HO_2$ uptake coefficient onto Cu-dopped ammonium sulphate aerosols at 43% relative humidity and proposed a novel parameterized equation (abbreviated as *NEq.* in the paper) for the prediction of $\gamma_{HO_2}$ that best fits all the laboratory results. Furthermore, for  Wangdu field campaign, we also calculated $\gamma_{HO_2}$ according to the *NEq.* and the impact of $HO_2$ uptake on HOx (=OH + $HO_2$) budget was evaluated.

## 2 Materials and Methods

### 2.1 The Model

A Multiphase Reaction Kinetic Model (MARK) is developed in this study for the simulation of $\gamma_{HO_2}$ for the laboratory experiments. The reaction mechanism and reaction rate constants are summarized in Table S.1 – S.4 in the Supplementary Information (SI). The MARK model is currently capable of simulating inorganic deliquescent aerosol at ambient pressure and temperature. The model directly calculates the quasi-first order gas phase uptake loss rate, $k_{het}$ (s$^{-1}$), in Eq. (1). In this model, aerosol liquid water content (ALWC) [g cm$^{-3}$] is more pertinent than surface density because of the influence of the *RH* on the uptake process (Kuang et al., 2018; Bian et al., 2014).

$$\frac{d[HO_2]}{dt} = -k_{het} \times [HO_2] \tag{1}$$

$$k_{het} = \left(\frac{R_d}{D_g} + \frac{4}{\gamma v_{HO_2}}\right)^{-1} \times \frac{3ALWC}{\rho R_d} \tag{2}$$

The units of aqueous reagents are converted to [molecule cm$^{-3}$] in the model by $k_{mt}$. To combine both gas phase molecular diffusion and liquid phase interface mass transport processes the approach adopted is using one variable called $k_{mt}$ (Schwartz, 1984; Schwartz, 1986), which is used in the calculation for gas-liquid multiphase reactions in many modelling studies (Lelieveld and Crutzen, 1991; Chameides and Stelson, 1992; Sander, 1999; Hanson et al., 1994). The definition of $k_{mt}$ is given by equation (3):

$$k_{mt} = \left(\frac{R_d^2}{3D_g} + \frac{4R_d}{3v_{HO_2}\alpha}\right)^{-1} \tag{3}$$

The rate of gas phase reactants (*X*) diffusing and dissolving to the condensed phase can be calculated in the framework of aqueous phase reactions as $k_{mt\_X} \times ALWC$ (where *X* is the reactant molecule). Moreover, the conversion rate of aqueous phase reactants to gas phase can be calculated as $\frac{k_{mt\_X}}{H^{cc} \times RT}$. The unit of $k_{mt}$ is s$^{-1}$, as $k_{mt}$ contains the conversion from m$_{air}$$^{-3}$ of the gas phase molecule concentrations to m$_{aq}$$^{-3}$ of the aqueous phase molecule concentrations and in the other direction. For larger particles (radius >1µm), $k_{mt}$ is mainly determined by gas phase diffusion of HO$_2$. For smaller particles (radius <1µm) $k_{mt}$ is mainly determined by the accommodation process. The MARK model can simultaneously simulate gas and liquid two-phase reaction systems in the same framework.

The aerosol particle condensed phase is not an ideal solution. Consequently, an effective Henry's law constant $H^{cc}$ should be applied in the model calculation, that takes into account the effects of solution pH and "*salting out*" effect in the small gas phase molecule (such as HO$_2$, OH, O$_2$ ect.) due to the existence of electrolytes in the solution (Ross and Noone, 1991). This study uses the ISORROPIA II thermodynamic model (Fountoukis and Nenes, 2007) to calculate the ALWC and components concentrations for metastable deliquescent aerosols. The effective Cu$^{2+}$ concentration in the aqueous phase, which is strongly influenced by non-ideal solution ionic strength, is also calculated following Ross and Noone (Ross and Noone, 1991).

## 2.2 Corrections to $\gamma_{HO_2}$ in the MARK model

### 2.2.1 Henry's law of gas phase reactants

The aerosol particle condensed phase solution is not an ideal solution as commented before. The addition of an electrolyte to water interferes with the gas dissolution and the organization of water molecules around the gas. This frequently results in a decrease in the solubility, or a "*salting out*" effect. This *salting out* effect is frequently a linear function of the molar ionic strength $I$. $H_0$ is estimated to be about 3900 M atm$^{-1}$ at 298K for $HO_2$ (Thornton et al., 2008; Golden et al., 1990; Hanson et al., 1992) and its temperature dependence is given accordingly to the IUPAC recommendation (Ammann et al., 2013; IUPAC

Task Group on Atmospheric Chemical Kinetic Data Evaluation, http://iupac.pole-ether.fr.). $H_0$ should be corrected by the solution pH and the "*salting out*" effect. In the MARK model, these two corrections are incorporated as $H^{cc}$:

$$H^{cc} = H_0 \times \left(1 + \frac{K_{eq}}{[H+]}\right) \times A_{HO_2} = 9.5 \times 10^{-6}\, exp\left(\frac{5910}{T}\right) \times \left(1 + \frac{K_{eq}}{[H^+]}\right) \times A_{HO_2} \tag{4}$$

The activity coefficient $A$ for $HO_2$ and other neutral small molecules such as $H_2O_2$ and $O_2$ can be expressed as (Ross and Noone, 1991):

$$A = 10^{-0.1 \times I} \tag{5}$$

According to this correction, $H^{cc}$ of $HO_2$ increases with $RH$ and decreases quickly after $[Cu^{2+}]$ reaches 0.1M in aerosol liquid phase, which limits $\gamma_{HO_2}$ at high $Cu^{2+}$ concentration.

### 2.2.2 Aerosol particle condensed phase $Cu^{2+}$ molality calculation

Inorganic species in ambient aerosol particles may be in the form of aqueous ions, or in the form of precipitated solids in

thermodynamic equilibrium with atmospheric gases and water vapor. The salts in the metastable aerosol are all dissolved in the aqueous phase. For meta-stable aerosols, this paper uses thermodynamic models to calculate ALWC and aerosol particle condensed phase component concentrations. In this work ISORROPIA II (Fountoukis and Nenes, 2007; Capps et al., 2012) thermodynamic equilibrium model for inorganic aerosol systems is used to take into account this.

In ambient aerosol, since the Fe concentration is about $10-100$ times (Mao et al., 2013a) higher than that of Cu, for an

aerosol pH ranging from $3-6$, the solubility of Fe (primarily $Fe^{2+}$) is rather small (Fang et al., 2017; Hsu et al., 2010a; Baker and Jickells, 2006; Oakes et al., 2012). The reaction rates of Fe/Mn for liquid phase $HO_2$ in aerosol is about 100 times slower than it is for Cu. For these reasons, the influence of aerosol Fe and Mn on $HO_2$ uptake can be neglected compared to Cu or scaled as equivalent $[Cu^{2+}]$. Thus, in this paper, we only focused the crucial influence of aerosol coppper on $HO_2$ uptake.

At low relative humidity, the aqueous phase is highly concentrated (i.e. with a high ionic strength), and the solution is

strongly non-ideal, consequently the activity coefficient and "*salting out*" effect must be taken into account for calculation of aerosol chemistry. The ion activity coefficient refers to the effective concentration of ions participating in an electrochemical reaction in an electrolyte solution.

Based on Ross and Noone (Ross and Noone, 1991), for an ion ($x_i$) of charge $z_i$ (i=x,y,z...), the activity coefficient ($\varphi_x$) is

$$log\ \varphi_x = -z_x^2 D - \sum_y \varepsilon(x, y, I)m_y \tag{6}$$

where $D$ is given by equation (7):

$$D = \frac{0.5109\sqrt{I}}{1+1.5\sqrt{I}} \tag{7}$$

$I$ is the ionic strength of a solution [M], which can be calculated as following equation:

$$I = \frac{1}{2} \cdot \sum m_i \cdot z_i^2 \tag{8}$$

$\varepsilon(x, y, I)$ is referred to as "interaction coefficients", and the summation extends over all ions ($y$) in the solution at a molality of $m_y$. For ions of similar charge, $\varepsilon(x, y, I)$ is set to zero. For ions of unequal charge, $\varepsilon(x, y, I)$ may be calculated from the logarithm solution mean activity coefficient $log(A_\pm)$ (Clegg et al., 1998) of the single electrolyte at the same $I$ according equation (9):

$$\varepsilon(x, y, I) = \frac{(log(A_\pm)+z_x z_y D)(z_x+z_y)^2}{4I} \tag{9}$$

In the condensed phase of aerosol particle, the effective molality of an ion $x_i$ ($[x_i]_{equ}$) can be calculated as:

$$[x_i]_{equ} = [x_i] \times \varphi_x \tag{10}$$

In the aerosol particle condensed phase, an effective concentration rather than the total concentration of Cu ion should be calculated in catalytic aqueous reactions with $HO_2$. The effective concentration of Cu ion can be calculated as:

$$[Cu^{2+}]_{equ} = [Cu^{2+}] \times \varphi_{Cu^{2+}} \tag{11}$$

$[Cu^{2+}]$ is the aerosol condensed phase soluble copper concentration.

## 2.2.3 The conversion formula of $[\overline{HO_2}]$ and $[HO_{2(r)}]$

Gas phase $HO_2$ molecules dissolve in the particle condensed phase and diffuse from the surface of a particle toward the center in parallel with aqueous phase reactions. We need to evaluate $[\overline{HO_2}]$, the assumed averaged steady-state $HO_2$ concentration over the volume of the particle. $[HO_{2(r)}]$ is $HO_2$ concentration at the surface of particles. The ratio of these two concentrations can be calculated as (Schwartz, 1986; Schwartz, 1984):

$$\frac{[\overline{HO_2}]}{[HO_{2(r)}]} = 3 \times \left(\frac{coth(q)}{q} - \frac{1}{q^2}\right) \tag{12}$$

where $q$ is given by equation (13):

$$q = R_d \times \left(\frac{k_{eff}}{D_{aq}}\right)^{0.5} \tag{13}$$

and $D_{aq}$ is the aqueous phase diffusion coefficient [cm$^2$s$^{-1}$], $k_{eff}$ is the comprehensive liquid phase reaction rate coefficient which encompasses both $HO_2$ dissolution equilibrium reactions and liquid phase chemical-physical reactions during $HO_2$ uptake process. In the copper-doped aerosol particle, because of the high value of $k_{eff}$ and small Count Median Diameter ($R_d$) (usually smaller than 1μm), the ratio $\frac{[\overline{HO_2}]}{[HO_{2(r)}]}$ is close to 1. At a diameter of 1μm, and a relative humidity between 40% and 90%, the condensed phase copper ion concentration varies from $10^{-5}$ to 1M, the average ratio of the surface $HO_2$ concentration

and the condensed phase HO$_2$ concentration is 0.89. At 400nm diameter particles for $RH = 40\%$ to $90\%$, the ratio is larger than 0.95. The ratios are calculated by simulation of $k_{eff}$ and the accordingly calculations by Equation (12) and (13). Thus, in this model, we assume the surface concentration of HO$_2$ equals to the condensed phase average HO$_2$ concentration.

## 2.3 Laboratory results for the HO$_2$ accommodation coefficient

The accommodation coefficient of HO$_2$ used in the model was determined for copper-doped inorganic aerosol particles using values taken from various previous laboratory studies. The accommodation coefficient of HO$_2$ ($\alpha_{HO_2}$) is approximately 0.5 in sulfate aerosol and even higher for chlorine or nitrate aerosol because of the catalytic effect of Cu$^{2+}$ on aqueous HO$_2$/O$_2^-$ (Table 1). In this situation, the aqueous reactions are fast enough for the uptake process be limited primarily by the mass transport process (accommodation).

With the wide distribution of Cu$^{2+}$ in aerosol particles, a high accommodation coefficient of HO$_2$ presents the possibility of HO$_2$ uptake as an important sink of HO$_x$ radicals. According to existing research results, the upper limitation of $\alpha_{HO_2}$ with aqueous sulfate aerosol particles is around 0.5. Thus, the MARK model typically selects the accommodation coefficient $\alpha_{HO_2}$ as 0.5. We also tested the influence of the accommodation coefficient on calculated HO$_2$ uptake coefficient in a field campaign, details please see the Supplementary Information.

Table 1: $\gamma_{HO_2}$ determined under laboratory conditions for copper-doped inorganic aerosols.

| Aerosol type | RH/% | Estimation of [Cu] in aerosol (mol L-1) | $\alpha_{HO_2}$ | Ref. |
|---|---|---|---|---|
| NH$_4$HSO$_4$ | 75% | 0.0059−0.067* | 0.40±0.21 | (Mozurkewich et al., 1987) |
| (NH$_4$)$_2$SO$_4$ | 45% | 0.5 | 0.53±0.13 | (Taketani et al., 2008) |
| (NH$_4$)$_2$SO$_4$ | 42% | 0.16 | 0.5±0.1 | (Thornton and Abbatt, 2005a) |
| (NH$_4$)$_2$SO$_4$ | 53−65% | 0.5−0.7* | 0.4±0.3 | (George et al., 2013) |
| (NH$_4$)$_2$SO$_4$ | 65% | 0.57 | 0.26±0.02 | (Lakey et al., 2016b) |
| (NH$_4$)$_2$SO$_4$ | 51% | 0.0027 | 0.096±0.024 | (Zou et al., 2019) |
| (NH$_4$)$_2$SO$_4$ | 43% | 0.38 | 0.355±0.023 | This work |
| NaCl | 53% | ~0.5 | 0.65±0.17 | (Taketani et al., 2008) |
| KCl | 75% | 5% of KCl solution | 0.55±0.19 | (Taketani et al., 2009) |
| LiNO$_3$ | 75% | 0.03−0.0063* | 0.94±0.5 | (Mozurkewich et al., 1987) |

*Cu concentration is in molality (M).

**2.4 The experimental setup and methodology of the latest results of $\gamma_{HO_2}$**

In this study, we also conclude the latest results which measured at Leeds. The experimental setup and methodology used to make the new measurements of γ(HO₂) reported here have been described in detail elsewhere (Moon et al., 2018b; Lakey et al., 2016b; George et al., 2013) and so only brief details are given here. In summary, the experiments were performed by moving an HO₂ injector backwards and forwards along the concentric axis of a laminar aerosol flow tube hence changing the contact time between HO₂ and the aerosols. Measurements of [HO₂] were performed using laser induced fluorescence (LIF) spectroscopy at low-pressure (the fluorescence assay by gas expansion (FAGE) technique (Heard and Pilling, 2003)) and the total aerosol surface area was determined with a Size Mobility Particle Sizer (SMPS) at the end of the flow tube. Aerosols were formed using a constant output atomiser (TSI, 3076) and the aerosol concentration and hence surface area could be varied, being controlled using a high efficiency particulate air (HEPA) filter in a bypass arrangement. Atomiser solutions were prepared by dissolving 0.01 moles of ammonium sulphate (AS) (Fisher scientific, >99%) with varying amounts of copper (II) sulphate (Fisher scientific, >98%) in 500 mL of Milli-Q water. The data were analysed as described in George et al 2013. The pseudo first-order loss rate coefficient (*k'*) was obtained from the gradient of a plot of ln(HO₂ signal) against the interaction time between HO₂ and the aerosol before sampling by the FAGE detector. The uptake coefficient (γ(HO₂)) was obtained from the linear least-squares gradient of the plot of *k'* against the surface area concentration of aerosols in the flow tube. The error given on all measurements of γ(HO₂) represents 2σ of the uncertainty of the fitted gradient. A correction to *k'* was applied to taking into account non-plug flow conditions in the flow tube using the Brown method.

**3 Results and Discussion**

**3.1 Parameter sensitivity analysis of the MARK model**

Hygroscopic inorganic particles are one of the most important components of PM₂.₅ in ambient air. The annual average contribution of inorganic aerosol to PM₂.₅ is between 25% and 48% across China (Tao et al., 2017), especially $NH_4^+$, $SO_4^{2-}$, $NO_3^-$ and other inorganic ions. In laboratory studies of radical heterogeneous reactions, $(NH_4)_2SO_4$ aerosol is most widely studied because of its simple components, easy way to generate and as they are important component for urban aerosols (Cheng et al., 2012; Yin et al., 2005). A simplified approach was used to explore the mechanism of HO₂ heterogeneous uptake to derive a parameterized equation for the uptake coefficient, $\gamma_{HO_2}$.

In this study, $(NH_4)_2SO_4$ aerosol uptake reactions of HO₂ are simulated by the MARK model, and good correlation between simulation results and experimental results are obtained especially considering the influence of both [Cu²⁺] and *RH*.

Figure 1 shows the influences of both factors, RH and condensed phase pH together with Cu⁺² concentration on the heterogeneous process of HO₂. As the *RH* rises, the $\gamma_{HO_2}$ exhibits a logarithmic growth. Higher *RH* means a higher water

content which dilutes the bulk phase ions thus promotes the activity coefficients of reactant ions in the aerosol particle condensed phase and the solubility of the gas phase reactant such as OH, HO₂ and H₂O₂.

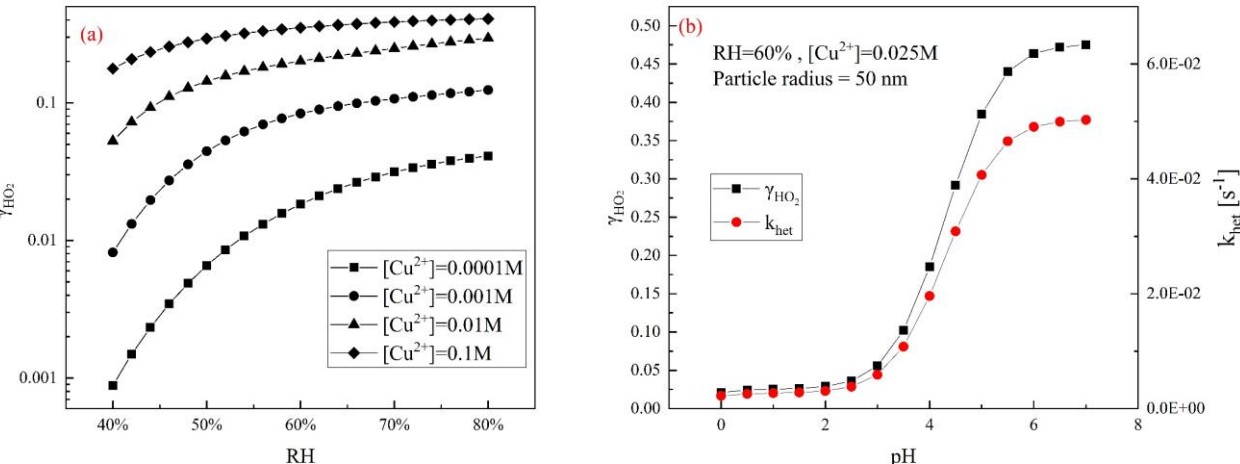

Figure 1: Influence of various parameters upon $\gamma_{HO_2}$ predicted by the MARK model. (a) $\gamma_{HO_2}$ increases with the *RH* at different [Cu²⁺]; (b) $\gamma_{HO_2}$ denoted by black squares and black line and $k_{het}$ in red circles and red line increase with aerosol particle condensed phase pH.

$\gamma_{HO_2}$ presents a sigmoid-shaped growth with aerosol particle condensed phase pH. In the model, it is found that as the pH rises, the uptake coefficient rises rapidly because HO₂ is a weak acid (*pKa* = 4.7) and has a low solubility in an acidic environment. The higher condensed phase pH is favorable for the dissolution equilibrium of the gas phase HO₂. This trend is consistent with the observed second-order rate constant of HO₂/O₂⁻ reviewed by Bielski et al. 1985 (Bielski et al., 1985). Moreover, aqueous phase reaction rates of HO₂/O₂⁻ and Cu²⁺/Cu⁺ increase with the increasing of condensed phase pH because in an alkaline environment HO₂ is more dissociated to O₂⁻ which has quicker reaction rate with Cu²⁺/Cu⁺. The pH of the ambient atmospheric aerosol is measured generally below 5 even when the concentration of NH₃ is high as in Beijing and Xi'an (Ding et al., 2019; Guo et al., 2017) with a range of 3-5. At this range, $\gamma_{HO_2}$ is highly affected by aerosol condensed phase pH mainly because of the change of HO₂ solubility.

**3.2 Model Validation**

Although the MARK model simulation results in this paper are not obtained by adjusting parameters to fit the experimental data points, the MARK model fitted well with these results under different ambient *RH* and Cu²⁺ concentrations.

At present, there are experimental measurements of $\gamma_{HO_2}$ at different RH (Thornton et al., 2008; Taketani et al., 2008, 2009; Taketani and Kanaya, 2010; Taketani et al., 2012; Matthews et al., 2014; Thornton and Abbatt, 2005a) but there is no an experimental systematic study of this dependence where only RH is changed and not other parameters. Many researches proposed that $\gamma_{HO_2}$ is higher for aqueous inorganic aerosol than for dry inorganic aerosol. Although the previous experiments

did not directly measure the dependence of RH, the change of the uptake coefficient met the simulation trend (see Figure 2). For hygroscopic inorganic aerosols, *RH* significantly affects the aerosol liquid water content, changing its ionic strength, aqueous reagent activity coefficients, and the solubility of the gas phase reactant such as OH, $HO_2$ and $H_2O_2$.

Aerosol condensed phase copper ion concentration is another important factor of $HO_2$ uptake by adjusting the aqueous reaction rates between $HO_2/O_2^-$ and Cu. As shown in Fig. 2. when the condensed phase copper ion concentration is less than

$1\text{-}2\times10^{-4}$ M, the heterogeneous uptake of $HO_2$ is not significant. This threshold is consistent with the results of previous researches (Mozurkewich et al., 1987; Lakey et al., 2016b). The threshold is also consist in different heterogeneous media of aerosol and droplets. As the copper concentration increases, $\gamma_{HO_2}$ rapidly rises to the limit of the accommodation coefficient determined by the $HO_2$ solubility.

What is more, laboratory measurement uncertainties will directly influence the evaluation of the deviation between the

modelled $HO_2$ uptake coefficient and the measured results because all the parameters inputted in the MARK model are in reference to the measurement conditions. However, it is difficult to calculate the detailed uncertainties from all factors that influence $\gamma_{HO_2}$ because of the nonlinear reaction system. Uncertainties of the experimental conditions such as RH and particle diameters are combined into the reported values of $\gamma_{HO_2}$. Taking all these into account, we calculated an averaged uncertainty for the experimental values of $\gamma_{HO_2}$ in different ranges of Cu ions concentration. Laboratory measurement uncertainty has the

largest value of 35.1% in the range of $1\times10^{-4}$ to 0.01 M soluble copper concentration, 14.9% below $1\times10^{-4}$ M and 9.3% higher than 0.01 M. In general, good agreement is achieved between the MARK model results and the results of the previous laboratory studies, which were also classified based on a statistical parameter: root mean square error (RMSE) (Figure 2). In this paper, the relative error of each measured data point is considered to calculate the weighted average in RMSE:

$$\text{RMSE} = \sqrt{\frac{\sum_{i=1}^{n}\left(\left(\log_{10}u_{i_{measured}} - \log_{10}u_{i_{model}}\right)^2 (\omega_i)^2\right)}{\sum_{i=1}^{n}(\omega_i)^2 \cdot n}} \qquad (14)$$

$u_{i_{model}}$ is the MARK model result at each $Cu^{2+}$ concentration and *RH*, $u_{i_{measured}}$ is the central value of each measurement result and $\omega_i$ is its corresponding relative error.

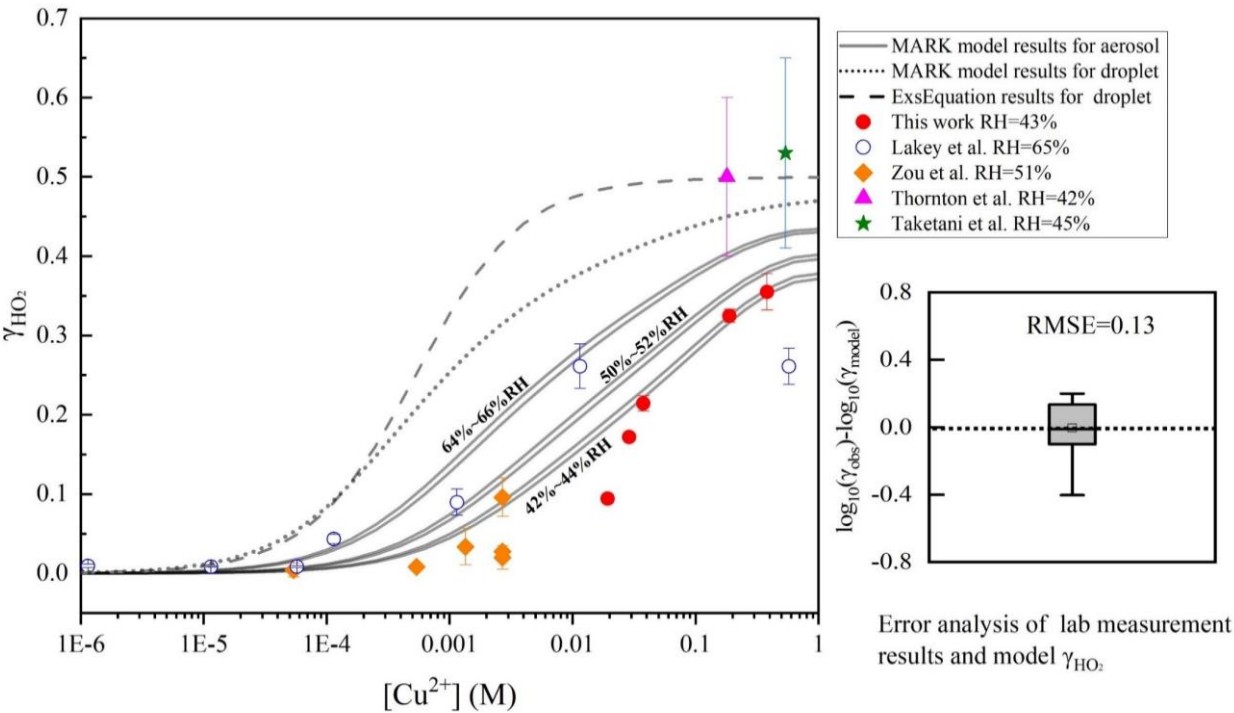

Figure 2: Dependence of $\gamma_{HO_2}$ on aerosol copper concentration. Red filled circles denote the results at 43% *RH* measured at
Leeds included in this paper. Blue hollow circles at 65% *RH* (Lakey et al., 2016b). Yellow filled diamonds denote results at
51% *RH* (Zou et al., 2019), filled purple triangle at 42% *RH* (Thornton and Abbatt, 2005a) and filled green star at 45% *RH*
(Taketani et al., 2008). The grey dashed line denotes the results of the classical parameterized equation (named as *CEq.* in this
paper) $\gamma_{HO_2}$ with dilute solution droplets (Thornton et al., 2008; Hanson et al., 1992; Hanson et al., 1994; Jacob, 2000; Kolb
et al., 1995), which was confirmed by researches of reactive gas molecular uptake on dilute solution droplets(Hu et al., 1995;
Magi et al., 1997) and on aqueous surfaces (Utter et al., 1992; Müller and Heal, 2002). The solid grey lines represent the model
results of the MARK model in this study at various *RH* (two lines represent the range of *RH* from 64% to 66%, 50% to 52%
and 42% to 44%) and the short dotted line represents the result in the MARK model of HO$_2$ with dilute solution droplets. The
root mean square error (RMSE) between the MARK modelled values and the full dataset is 0.13. The aerosol pH is set as 4.5
based on the aqion 7.0.8 interface considering the participation of Cu ion (for details please see https://www.aqion.de/).


### 3.3 Comparison of the classical parameterized equation and the MARK model

The classical parameterized equation (*CEq.*) (Thornton et al., 2008; Hanson et al., 1992; Hanson et al., 1994; Jacob, 2000;
Kolb et al., 1995; Ammann et al., 2013;IUPAC Task Group on Atmospheric Chemical Kinetic Data Evaluation,

solution droplets (Hu et al., 1995; Magi et al., 1997) and on aqueous surfaces (Utter et al., 1992; Müller and Heal, 2002):

$$\frac{1}{\gamma_{HO_2}} = \frac{1}{\alpha_{HO_2}} + \frac{v_{HO_2}}{4H_{corr}RT\sqrt{D_{aq}k_{TMI}[TMI]}[coth(\frac{R_d}{l_{rd}} - (\frac{l_{rd}}{R_d}))]} \tag{15}$$

$$l_{rd} = \sqrt{\frac{D_{aq}}{k_{TMI}[TMI]}} \tag{16}$$

When the classical parameterized equation (*CEq.*) is applied to the calculate $HO_2$ uptake coefficient with aerosol, *CEq* has

higher deviation of $\gamma_{HO_2}$ between the measured results compared to the MARK model. All input parameters are the same except that the MARK model involved more liquid phase reactions instead of only considering the second order rate coefficient ($k_{TMI}$) of $HO_2$ and $O_2^-$ with transition metal ions as the *CEq.* did. $k_{TMI}$ is the most important parameter in the calculation of uptake coefficient. Based on the research by Bielski in 1985 (Bielski et al., 1985), we used the effective rate constant of $HO_{2\_total}$ ($=HO_{2(aq)}+ O_{2(aq)}^-$) with Cu ions as $1.5\times10^7$ $M^{-1}$ $s^{-1}$ rather than the more commonly used value of $1\times10^9$ $M^{-1}$ $s^{-1}$ considering the

pH limitation (pH is about 3-5 in ambient aerosol particle condensed phase as discussed above). The prior value ($1.5\times10^7$ $M^{-1}$ $s^{-1}$) reflects the rate of reaction between $HO_2$ and $Cu^{2+}$, more prevalent in acidic aerosol such as ammonium sulphate, and the latter ($1\times10^9$ $M^{-1}$ $s^{-1}$) between $O_2^-$ and $Cu^{2+}$ ions, which is more prevalent in aerosols with a pH greater than the $pK_a$ of $HO_2$, such as NaCl (Bielski et al., 1985). This treatment within the calculation can bring predictions more in line with experimental results in the *CEq.* as shown in the dashed line in Figure 2.

IUPAC (Ammann et al., 2013;IUPAC Task Group on Atmospheric Chemical Kinetic Data Evaluation, http://iupac.pole-ether.fr.) proposed the effective rate coefficient $k^1$ for the reaction of $HO_{2\_total}$ ($=HO_{2(aq)}+ O_{2(aq)}^-$) with Cu ions as $5x10^5$ $M^{-1}$ $s^{-1}$ to achieve the best fit based on the calculation results from Lakey et al. (2016b). This assumption is not in accordance with the aqueous reaction rate coefficient from other databases mentioned below and needs further laboratory measurements to confirm it. According to the aqueous reaction rate coefficient from NIST and the latest measurement result (Lundström et al., 2004;

Huie, 2003), the rate coefficient of $HO_2$ with $Cu^{2+}$ is $1\times10^8$ or $1.2\times10^9$ $M^{-1}$ $s^{-1}$ at pH= 2 and pH=1, respectively. These two rate coefficients were quantified in a low pH environment (pH=2 for $1.2\times10^9$ $M^{-1}$ $s^{-1}$ and pH=1 for $1\times10^8$ $M^{-1}$ $s^{-1}$). At the same time, the reaction rate of $O_2^-$ with $Cu^{2+}$ is $8\times10^9$ $M^{-1}$ $s^{-1}$ for pH in the range 3-6.5 (Huie, 2003). At higher pH, the reaction rate of $HO_2$ with $Cu^{2+}$ may change, but it is unknown whether it will decrease by four orders of magnitude. Further kinetics experiments are needed at varying pH to verify the reaction rate coefficient of $Cu^{2+}$ ions with $HO_2$ and $O_2^-$ in aqueous solution.

The rate constants used in the MARK model are shown in the Table S. 1 in the SI. The reaction rate of $Cu^{2+}$ with $HO_2/O_2^-$ is $1\times10^8$ and $8\times10^9$ $M^{-1}$ $s^{-1}$ in the MARK model. We also test the MARK model with dilute solution droplets as shown in Figure 2 the short-dotted line.

The classical parameterized equation (*CEq.*) is more applicable to calculate uptake coefficient of reactive gas molecular with diluted solution droplets such as cloud or rain droplets. The MARK model uses the same framework with the *CEq.* and

considered more parameters influencing uptake process such as the activity coefficients of reactive reagents and the effects of

valence states in aerosol particle condensed phase. Considering the small RMSE between the MARK model and the laboratory studies, we proposed a novel parameterized equation (*NEq.*) to better describe the influence of [Cu²⁺] and *RH* on $\gamma_{HO_2}$.

## 3.4 A novel parameterized equation of $\gamma_{HO_2}$

When the full reaction system reaches steady-state, the reaction of HO₂ in the aqueous particle phase can be expressed as the following reaction scheme (Schwartz, 1984; Schwartz and Freiberg, 1981; Schwartz, 1987):

$$HO_{2(g)} \rightleftharpoons HO_{2(r)} \rightleftharpoons HO_{2(a)} \xrightarrow{k_{eff}} Products \tag{17}$$

Gas phase HO₂₍g₎ molecule transports onto the surface of the aerosol particles, HO₂₍r₎ then dissolves at the condensed phase to give HO₂₍a₎. The reactions between Cu²⁺/Cu⁺ and HO₂ can be seen as catalytic reactions, because in the model simulations,
the total amount of [Cu²⁺]+[Cu⁺] does not change with reaction time. The rate of HO₂ aqueous reaction with copper ions is noted as $k_{eff}$. For fine particles, we can safely assume that the interface concentration $[HO_{2(r)}]$ is equal to the condensed phase average $\overline{[HO_2]}$ concentration due to rapid diffusion in the liquid phase (details have been discussed in section 2.2.3). For the submicrometer aerosol particles with which most uptake reaction occurs, the influence of the gas phase diffusion limitation can be neglected. Hanson et al. (1994) proposed the definition of the uptake coefficient as $\gamma = \alpha(1 - \frac{c_{a,surf}}{H^{cc}c_{g,surf}})$ where $c_{a,surf}$
is the suface concentration of the reactant, $c_{g,surf}$ is the gas phase concentration. In the process of HO₂ uptake, we deduce the parameterized equation (*NEq.*) of $\gamma_{HO_2}$ in the framework of the resistance model:

$$\frac{1}{\gamma} = \frac{1}{\alpha_{HO_2}} + \frac{3 \times v_{HO_2}}{4 \times R_d \times H_{corr} \times RT k_{eff}} \tag{18}$$

$$k_{eff} = f(\text{ALWC}, \text{PM}) \times [Cu^{2+}]_{equ} \tag{19}$$

$$f(\text{ALWC}, PM) = 10^6 \times \left(5.87 + 3.2 \times ln\left(\frac{\text{ALWC}}{[\text{PM}]} + 0.067\right)\right) \times [\text{PM}]^{-0.2} \tag{20}$$

$[Cu^{2+}]_{equ} = [Cu^{2+}]^{\varphi} = [Cu^{2+}]^{0.65} \tag{21}$

From Eq. (18), it can be deduced that $\gamma_{HO_2}$ can be calculated by optimizing $k_{eff}$ under different ambient environmental conditions from the MARK model results. The MIPFIT model (Markwardt, 2009; Lewis et al., 2009) in the IDL software program is used to optimize $k_{eff}$ using the Levenberg-Marquardt algorithm. Because the equation is empirical, the initial value of $k_{eff}$ is set as 1. $k_{eff}$ is related to the condensed phase soluble copper concentration [Cu²⁺] with an exponential
relationship to the parameterization of the catalytic reactions, which is denoted in Eq. (19). The exponent of [Cu²⁺] is globally fitted using the MIPFIT method. It is found that the overall $R^2$ is higher than 0.97 and the residual is minimized when the exponent is 0.65. $f(\text{ALWC}, [\text{PM}])$ has a negative exponential relationship to [PM], and has a positive linear relationship to *RH*.

We further calculated the RMSE of the modeled data and *NEq.*(Eq.15) data under different *RH* conditions. The range of values shows the difference between the modeled data and *NEq.* data at different Cu²⁺ concentration. At low *RH* and
consequently relatively low ALWC, $\gamma_{HO_2}$ is more sensitive to [Cu²⁺] especially at low [Cu²⁺]. This sensitivity cannot be fully

represented in the parameterized equation. What is more, at low [$Cu^{2+}$] and low *RH*, the value of $\gamma_{HO_2}$ is smaller than in other conditions, so that the uncertainty of $\gamma_{HO_2}$ becomes larger.

All the RMSE values are smaller than 0.2, which indicates a minor deviation from the laboratory results in our $\gamma_{HO_2}$ equation. In the typical ambient urban atmospheric environment, with an aerosol mass concentration of 10-300 µg m$^{-3}$, aqueous $Cu^{2+}$ concentration of $10^{-5}$-1 molar concentration, and a relative humidity between 40%-90%, the *NEq.* can be used. Beyond the range, the application of the *NEq.* may cause a large deviation. The HO$_2$ uptake under dry conditions needs further investigation in the future, but probably it is not of high priority because the effective reaction volume becomes 10% or less of the aerosol volume for dry conditions and the HO$_2$ uptake may then be neglected for typical tropospheric conditions (Taketani et al., 2008; Kanaya et al., 2009; Taketani and Kanaya, 2010; Thornton et al., 2008; George et al., 2013).

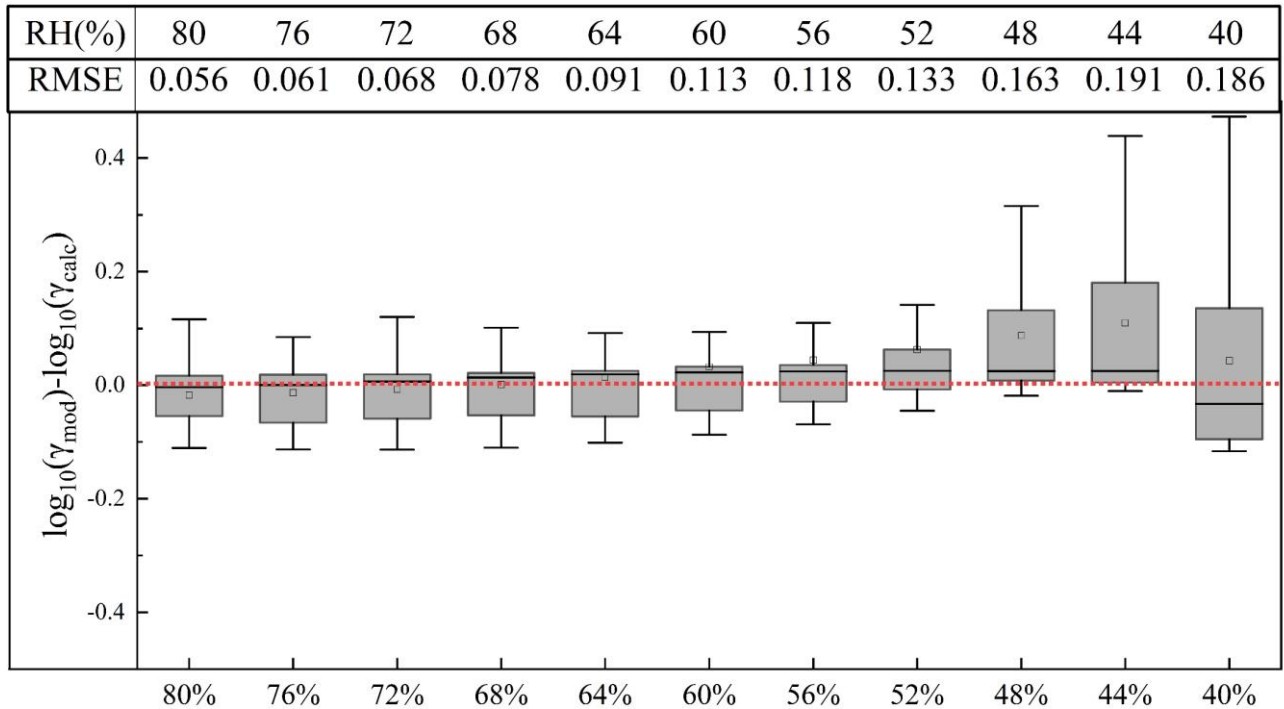

Figure 3: Comparison of the MARK modeled $\gamma_{HO_2}$ to the corresponding calculated values from the *NEq.* $log_{10}(\gamma_{mod})$ is the logarithmic value of modeled $\gamma_{HO_2}$ and $log_{10}(\gamma_{cal})$ is the calculated value from the *NEq.*

## 3.5 Evaluation of the impact of the new HO$_2$ uptake parametrization in the Wangdu campaign

Many model studies (Lakey et al., 2015; Mao et al., 2013b; Martinez et al., 2003; Tie et al., 2001; Whalley et al., 2015) suggest that heterogeneous uptake of HO$_2$ radical affects the global distribution of trace gases and the atmospheric oxidant capacity especially in regions with high aerosol loading or low NOx concentration. The importance of aerosol chemistry as a

sink for ozone precursors in North China Plain has been suggested in many model studies (Li et al., 2019b; Lou et al., 2014). The competition of $HO_2$ with aerosol and gas phase reactants is crucial when evaluating the influence of heterogeneous reactions on the atmospheric oxidant capacity.

Based on the results of a comprehensive field campaign performed in summer 2014 in a rural site (Wangdu) in the North China Plain(Tan et al., 2020), the $HO_2$ uptake coefficient and the ratios of the $HO_2$ uptake loss rates ($TR_{HO2uptake}$) to the sum of the ROx termination rates ($TR_{ROxsinks}$) are calculated with direct measurements of the ROx radicals, trace gas species, ALWC and the aerosol condensed phase component concentrations (please see the SI for details). The experimental determined ROx termination rates include reaction channels from OH + $NO_2$, OH + NO, $HO_2$ + $HO_2$, $HO_2$ + $RO_2$ and $RO_2$ + NO. Considering

the solubility and size distribution of particle metal copper (Fang et al., 2017; Hsu et al., 2010a; Mao et al., 2013a) we can estimate $\gamma_{HO_2}$ in daytime and night.

### 3.5.1 Average results of observed meteorological parameters and trace gases concentration in the Wangdu campaign

Wangdu is located in the center of the Beijing-Tianjin-Hebei area and it is a regional site. The observations were carried out in the summer with serious photochemical smog pollution events (Tan et al., 2017; Tan et al., 2020). Table 2 summarizes the

meteorological and chemical conditions in this field campaign. In terms of parameters such as temperature, pressure and relative humidity, the Wangdu area is a high-temperature and high-humidity environment with a monsoon climate.

Table 2: Average daytime results of observed meteorological parameters and trace gases concentration in Wangdu campaign from June 10th, 2014 to July 6th, 2014.

| Parameters | Average values | $1\sigma$ Accuracy |
|---|---|---|
| Temperature /°C | 27±4 | ±0.05% |
| Pressure /hPa | 1000±5 | ±0.05% |
| $RH$/% | 61±18 | ±0.05% |
| $O_3$/ppb | 55.6±9.0 | ±5% |
| $NO_x$/ppb | 10±13.6 | ±20% |
| HONO/ppb | 0.8±0.24 | ±20% |
| CO/ppm | 0.6±0.19 | <5% |
| Isoprene/ppb | 0.5±0.11 | ±15%-20% |
| HCHO/ppb | 7±0.69 | ±5% |


### 3.5.2 Calculation of soluble copper ion concentration

During this campaign, the total concentration of heavy metal ions in fine particles (smaller than 2.5$\mu m$) was measured using a commercial instrument based on non-destructive X-ray fluorescence technique (Xact 625, Cooper Environmental). Since the concentration of soluble copper concentration rather than total copper concentration is used in the model, it is necessary to analyze the ratio of soluble copper to total copper in the aerosol particles. For particle radius smaller than 2.5$\mu m$, which are the most contributing bins of aerosols in $HO_2$ uptake, the mass fraction of Cu is about 33%−100% compared with other two size bins in ambient aerosols (2.5-10 $\mu m$, >10$\mu m$) (Mao et al., 2013a). According to previous research results, the dissolution ratio of copper in aerosol particles varies from 20% to 70% in different regions, being solubility lower in smaller particles (Fang et al., 2017; Hsu et al., 2004; Hsu et al., 2010b). Therefore, when using the *NEq.* to calculate the $HO_2$ heterogeneous uptake coefficient, it is necessary to reduce the copper concentration considering the solubility and the distribution in the accumulation mode of aerosol particles. We take 50% copper is soluble in the particle condensed phase and 50% copper is in the accumulation mode. Thus, we assume 25% of total aerosol metal copper concentration is soluble in the accumulation mode when calculating $\gamma_{HO_2}$ in Wangdu campaign. The hourly resolution total copper concentration (ng m$^{-3}$) is divided by the aerosol volume concentration and the atomic mass of copper (64) to obtain the total copper molar concentration in the aerosol (mol L$^{-1}$). $\gamma_{HO_2}$ rather depends on copper concentration so we also evaluate the influence of copper solubility on the uptake coefficient. What is more, the unequally distribution of copper ions will also influence the $HO_2$ uptake coefficient (details in the SI).

### 3.5.3 $\gamma_{HO_2}$ estimated at Wangdu field campaign

By inputting the soluble copper concentration, aerosol mass concentration, aerosol particle geometric mean diameter and the corresponding relative humidity and temperature into the *NEq.*, we can obtain an estimation of $\gamma_{HO_2}$ in suburban Wangdu, which is shown in Fig. 4 (a) and (b). The time resolution is 1 hour. The aerosol pH is calculated using the thermodynamic model ISORROPIA-II (Fountoukis and Nenes, 2007) and the averaged value is 3.41± 0.69 (1σ). Average aerosol mass concentration is 67.2±39.7 μg m$^{-3}$, the average Cu concentration is 35.8±57.7 ng m$^{-3}$. The results of a fit to a Gaussian function results in a $\gamma_{HO_2}$ value of 0.116 ± 0.086 (1σ) the Wangdu campaign ($\gamma_{HO_2}$ will increase from 0.065±0.051 (1σ) at 10% solubility to 0.196±0.142 (1σ) at 70% solubility for the summary of day and night data).

Tan et al. (2017) had compared the measured and modelled OH, $HO_2$ and $RO_2$ radicals in the Wangdu campaign. However, in this paper, they did not discuss the influence of $HO_2$ uptake. A very recent publication (Tan et al., 2020) calculated $\gamma_{HO_2}$ in the Wangdu campaign based on the comparison of field measurement data for $HO_2$ and concentrations calculated by the box model. The paper proposes that all $\gamma_{HO_2}$ calculated in this way from the Wangdu campaign can be fitted to a Gaussian distribution around the value of 0.08 ± 0.13 (1σ). This value isin the range of our estimation in this paper considering the influence of aerosol morphology and the indirect measurement uncertainty (please see the SI).

The experimentally determined ROx termination rates include reaction channels from OH + NO$_2$, OH + NO, HO$_2$ + HO$_2$, HO$_2$ + RO$_2$, RO$_2$ + NO. The ratio ($R_1$) of HO$_2$ uptake loss rate ($L_{HO2uptake}$) to the whole RO$_x$ loss rate ($L_{ROx}$) is calculated by Equation (22) and (23).

$$L_{HO2\text{uptake}} = 0.25 \cdot v_{HO_2} \cdot [ASA] \cdot [HO_2] \tag{22}$$

$$R_1 = \frac{L_{HO2uptake}}{L_{ROx}} \tag{23}$$

[$ASA$] is the aerosol surface area [$\mu m^2\, cm^{-3}$].

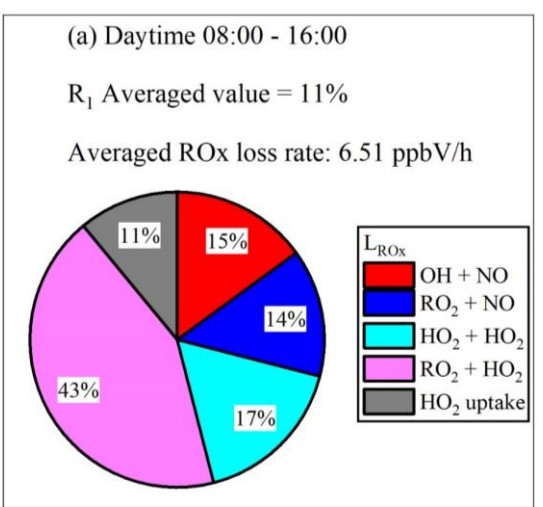 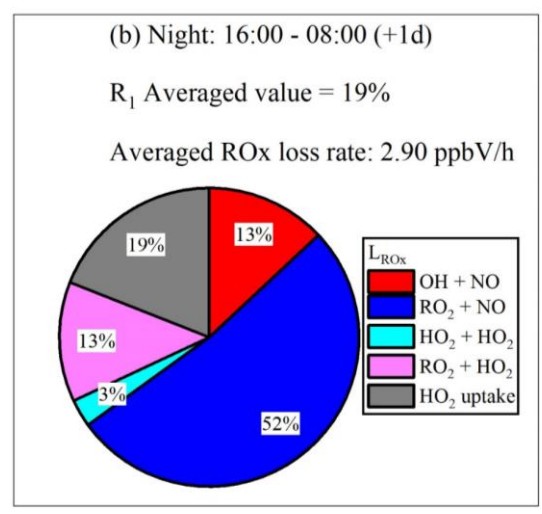

Figure 4: $R_1$ calculated by the *NEq.*. Pie charts show the values of $R_1$ and the loss rates for ROx during daytime (a) and nighttime (b). The averaged daytime (08:00−16:00) ROx radical loss rate is 6.5 ppbV/h and that for nighttime (16:00−08:00 (+1d)) is 2.9 ppbV/h.

No significant difference of $\boldsymbol{\gamma_{HO_2}}$ is observed during daytime and night. The HO$_2$ uptake coefficient is slightly higher at night due to the higher *RH* (57.6% at day and 67.4% at night). However, because of the high uncertainty of the uptake coefficient, such a high trend cannot be concluded to other cases. HO$_2$ heterogeneous uptake reactions with aerosol particles have small impact on ROx radical termination at daytime as shown in Fig. (4 a). However, HO$_2$ uptake may be important in the termination of ROx radicals at night shown in Fig. (4 b). The daytime ratio $R_1$ is lower than it is at night because of the lack of photochemical reactions, thus a longer HO$_2$ lifetime at night. The high proportion of RO$_2$+NO during night is due to high [NO] at dawn.

The RO$_2$ concentration is also important when evaluating the impact of HO$_2$ uptake. Using the modeled value of RO$_2$ concentration in Wangdu campaign, a higher proportion of HO$_2$ uptake to about 21% of ROx sinks in daytime can be calculated.

However, using the modified field measured $RO_2$ concentration in Wangdu campaign, $HO_2$ uptake is less important in the budget of ROx as shown in Fig (4 aa), which is in line with the results from Tan et al. (2020).

### 3.5.4 Discussion of uncertainties of $\gamma_{HO_2}$ estimated at Wangdu field campaign

The impact of $HO_2$ aerosol uptake on the ROx budget is complicated by large uncertainties in the $HO_2$ uptake coefficient under ambient conditions. The *NEq.* is applicable under the assumption of steady-state concentrations and with metastable or liquid aerosol particles (if the ambient *RH* over a completely liquid aerosol decreases below the deliquescence *RH*, the aerosol may not crystalize immediately but may constitute a supersaturated aqueous solution (i.e., in the metastable state) (Song et al., 2018)). The approximate calculation of $HO_2$ concentration gradients within the aerosol particle condensed phase also cause deviations for larger particles.

Organic content of an aerosol particle may affect several important parameters in the uptake model (Lakey et al., 2016b; Lakey et al., 2015). For example, the aerosol pH, hygroscopic properties of the aerosol, the rate of diffusion of $HO_2$ within the aerosol and a reduction in the concentration of $Cu^{2+}$ via the formation of complexes that could affect the ability of Cu to undergo redox reactions with $HO_2$ and $O_2^-$. Hence, it is expected that the presence of organic matter would change the value of $\gamma_{HO_2}$. We tested the core-shell morphology of aerosol particles influence on $HO_2$ uptake in the Wangdu campaign (details in the SI). Organic matter will lower the uptake coefficient about 25% to 40% under the assumption of 20% -50% $PM_{2.5}$ mass is organic matter.

Another uncertainty comes from aerosol particles morphology. The bulk diffusion coefficient of $HO_2$ and other reactive molecules should be lower in the situation of semi-solid particles (Berkemeier et al., 2016; Shiraiwa et al., 2010; Mikhailov et al., 2009) and would change with the water activity and the organic components (Price et al., 2015). For crystalline or amorphous solid aerosol particles, $HO_2$ will undergo surface reactions and diffuse across the surface rather than be accommodated within the aerosol bulk. The MARK model has limitations in the calculation of $\gamma_{HO_2}$ with semi-solid aerosol particles. In the Wangdu campaign, $\kappa_{sca}$ (optical aerosol hygroscopicity parameter) ranges from 0.05 to 0.35 with an average of 0.22. The ambient *RH* during the Wangdu campaign shows significant diurnal variations and varies greatly from 15% to 97%, with an average value of 61% (Kuang et al., 2019) indicating that the percentage of solid aerosol particles is relatively low and hence do not significantly influence $\gamma_{HO_2}$. Anyway, aerosol particles morphology relative to an aqueous phase will influence the uptake coefficient of $HO_2$. The uptake process would vary with mixing state of the particles, thus the predicted $\gamma_{HO_2}$ values here may be biased as a result but represents an average over bulk aerosols.

The interaction between organics and soluble copper and the influence of organics on aerosol properties will lead to further uncertainty in the calculation of the uptake coefficient. Lakey et al. (Lakey et al., 2016a; Lakey et al., 2015; Lakey et al., 2016b) have also shown that the addition of an organic compound to $Cu^{2+}$ doped aerosols such as oxalic acid, which forms oxalate ions $(C_2O_4)^{2-}$ in the aerosol, results in a lower value of $\gamma_{HO_2}$ as such ions forms a complex with the TMI.

As noted above, the value ($0.116 \pm 0.086$ ($1\sigma$)) estimated by the *NEq.* represents the upper limitation of $\gamma_{HO_2}$ in the Wangdu field campaign.

## 4 Summary and conclusions

Taketani et al. collected the filter samples of aerosol in Mts. Tai and Mts. Mang, North China (Taketani et al., 2012) and re-aerosolized from the water extracts of sampled particles. The measured uptake coefficients for Mt. Tai samples ranged between 0.09 and 0.40, while those at Mt. Mang were between 0.13 and 0.34. Li et. al (Li et al., 2019b) suggest that the rapid decrease of $PM_{2.5}$ in China has slowed down the reactive uptake rate of $HO_2$ radicals by aerosol particles and could have been the main reason for the increase in ozone in the North China Plain in the recent years. They apply a value of the uptake coefficient of 0.2 in their model calculations. However, the results of the MARK model and of the *NEq.* in this paper suggest that the $HO_2$ uptake coefficient could be smaller and highly variable for typical conditions in the North China Plain. Further research is needed to study the effects of heterogeneous uptake of $HO_2$ on gas phase and heterogeneous physicochemical reactions under different environmental conditions in different regions. The novel parameterized equation proposed in this paper provides an effective way for more detailed calculation of the effects of $HO_2$ heterogeneous reactions on the atmospheric radical budget, ozone production and particulate matter generation. This is the first attempt to parameterize the heterogeneous uptake coefficient of $HO_2$ with aerosol particles in China campaign. This equation estimates the $\gamma_{HO_2}$ in a comprehensive field campaign which is in agreement with the simulation results from the comparison of gas phase radical concentrations (Tan et al., 2020). Overall, we can conclude that the $HO_2$ uptake process needs to be considered in photochemical box models for the study of the HOx radical budget. The exact value is highly variable with respect to the change of copper concentrations in the aerosol particle condensed phase and other factors. The measurement of condensed phase soluble copper and other TMIs, organic content, as well as the aerosol liquid water should be added for future field campaigns for the study of the HOx radical budget.

## Appendix A

Table A1 Description and units of parameters used in the MARK model and the parameterized equations

| Parameter | Description | Unit |
|---|---|---|
| | Used in the parameterized equation | |
| $\gamma_{HO_2}$ | $HO_2$ uptake coefficient | - |
| $\alpha_{HO_2}$ | Mass accommodation coefficient of $HO_2$ which is the probability that a $HO_2$ molecule colliding with the aerosol surface leads to dissolution, reaction or volatilization | - |

| | | |
|---|---|---|
| $v_{HO_2}$ | Mean molecular speed of HO$_2$ | cm s$^{-1}$ |
| $R_d$ | Count Median Radius of the aerosols | cm |
| $R_c$ | radius of the aqueous core | cm |
| $H_{corr}$ | Henry's constant corrected for solution pH | mol cm$^{-3}$ atm$^{-1}$ |
| | $H_{corr} = H_0 \times \left(1 + \dfrac{K_{eq}}{[H^+]}\right)$ | |
| $H_0$ | physical Henry's law constant | mol cm$^{-3}$ atm$^{-1}$ |
| $H^{cc}$ | effective Henry's law constant | mol cm$^{-3}$ atm$^{-1}$ |
| $H_{org}$ | Henry's law constant of HO$_2$ for organic coating | mol cm$^{-3}$ atm$^{-1}$ |
| $R$ | gas constant | cm$^3$ atm K$^{-1}$ mol$^{-1}$ |
| $T$ | temperature | K |
| $RH$ | relative humidity ranging from 0.4 to 0.9 | 0-1 |
| ALWC | aerosol liquid water content | g cm$^{-3}$ |
| [PM] | Mass concentration of PM$_{2.5}$ | μg cm$^{-3}$ |
| $D_g$ | gas phase diffusion coefficient of HO$_2$ | cm$^2$s$^{-1}$ |
| $D_{aq}$ | aqueous phase diffusion coefficient | cm$^2$s$^{-1}$ |
| $D_{org}$ | solubility and diffusivity of HO$_2$ in the organic coating | cm$^2$s$^{-1}$ |
| $\varepsilon$ | ratio of the radius of the aqueous core ($R_c$) and the particle ($R_d$). | - |
| $l$ | Thickness of organic coating which is calculated from the volume ratio of the inorganics to total particle volume with the assumption of a hydrophobic organic coating (density, 1.27 g cm$^{-3}$) on the aqueous inorganic core (with a density of 1.77 g cm$^{-3}$). | cm |
| Used in the corrections in the MARK model or the classical parameterized equation | | |
| $\rho$ | density of the aerosol particles | g cm$^{-3}$ |
| $I$ | Solution molar ionic strength | M |
| $A$ | activity coefficient for gas phase HO$_2$ and other neutral small molecules | - |
| $\varphi_x$ | activity coefficient of ion in solution | - |
| $m_y$ | molality of an ion in solution | M |
| $\varepsilon(x, y, I)$ | "interaction coefficients", the summation extends over all ions ($y$) in the solution at a molality of $m_y$ | - |
| $[x_i]_{equ}$ | effective molality of an ion $x_i$ | M |
| $[\overline{HO_2}]$ | averaged steady-state HO$_2$ concentration over the volume of the particle | M |
| $[HO_{2(r)}]$ | HO$_2$ concentration at the surface of particles | M |

| | | |
|---|---|---|
| $K_{eq}$ | solution equilibrium constant for HO$_2$ in the gas phase | M$^{-1}$ s$^{-1}$ |
| $k_{eff}$ | comprehensive liquid phase reaction rate coefficient which encompasses both HO$_2$ dissolution equilibrium reactions and liquid phase chemical-physical reactions during HO$_2$ uptake process | M$^{-1}$ s$^{-1}$ |
| $k_{TMI}$ | second order rate coefficient ($k_{TMI}$) of HO$_2$ and O$_2^-$ with transition metal ions used in the classical equation | M$^{-1}$ s$^{-1}$ |
| k$^1$ | effective rate coefficient used in the classical equation proposed by IUPAC | M$^{-1}$ s$^{-1}$ |

**Author Contribution**

Keding Lu conceived the study. Huan Song and Keding Lu developed the MARK model for multiphase simulations. Xiaorui Chen and Qi Zou improved the codes of the MARK model. Zhaofeng Tan, Hendrik Fuchs, Keding Lu, Alfred Wiedensohler, Mei Zheng, Andreas Wahner, Astrid Kiendler-Scharr, Yuanhang Zhang contributed to the related measurements of the Wangdu field campaign. Dwayne E. Heard, Daniel R. Moon and María-Teresa Baeza-Romero contributed the laboratory studies of HO$_2$ uptake coefficients, and they have contributed to writing the manuscript. Huan Song performed the model simulations and prepared the manuscript with Keding Lu and Zhaofeng Tan which was enhanced by contributions from all the co-authors.

**Competing Interest**

The authors have no conflict of interests.

**Data Availability**

Data supporting this publication are available upon request for the corresponding author (k.lu@pku.edu.cn).

**Acknowledgments**

This study was supported by the Beijing Municipal Natural Science Foundation for Distinguished Young Scholars (Grants No. JQ19031), the National Natural Science Foundation of China (Grants No. 21976006, 91544225, 21522701, 91844301; the National Research Program for Key Issue in Air Pollution Control (Grants No. DQGG0103-01).

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
