# Peer review of "Influence of aerosol copper on HO2 uptake: A novel parameterized equation"

_Atmospheric Chemistry and Physics, 2020_

## Short Comment (SC1) · 27 Mar 2020

A comment on the paper:

Song, H., Chen, X., Lu, K., Zou, Q., Tan, Z., Fuchs, H., Wiedensohler, A., Zheng, M., Wahner, A., Kiendler-Scharr, A., and Zhang, Y, Influence of aerosol copper on HO2 uptake: A novel parameterized equation, Atmos. Chem. Phys. Discuss., https://doi.org/10.5194/acp-2020-218, in review, 2020.

The paper uses laboratory measurements of HO2 uptake coefficients experimentally obtained by the University of Leeds for copper ion doped ammonium sulphate (AS) aerosols, and data from other groups. However, the origin of the data from the University of Leeds is not clear and the citation for the data is not correct. In Figure 2 for

example, the reference Moon et al. 2018 is used to cite both sets of measurements at 43% and 65% RH. Moon et al. (2018) as cited by the paper does not contain any data for the uptake of HO2 onto copper doped aerosols, rather Moon et al 2018 concerns the uptake of HO2 onto TiO2 aerosols. Song et al. (2020), reference this paper as:

Moon, D. R., Taverna, G. S., Anduix-Canto, C., Ingham, T., Chipperfield, M. P., Seakins, P. W., Baeza-Romero, M.-T., and Heard, D. E.: Heterogeneous reaction of HO2 with airborne TiO2 particles and its implication for climate change mitigation strategies, Atmos. Chem. Phys., 18, 327-338, 10.5194/acp-18-327-2018, 2018.

but this is clearly not relevant to Figure 2 or the data shown in Table 1.

So it is not clear what the source of the data are from the University of Leeds for this figure and also that used in Table 1 that are cited incorrectly as Moon et al. 2018.

In a previous paper:

Qi Zou, Huan Song, Mingjin Tang, Keding Lu, Measurements of HO2 uptake coefficient on aqueous (NH4)2SO4 aerosol using aerosol flow tube with LIF system, Chinese Chemical Letters 30 (2019) 2236–2240.

the authors also show in Figure 3 of that paper some University of Leeds data for HO2 uptake onto copper doped AS aerosols at RH=43% and RH=65%. However, in that paper (Zou et al., 2019), two PhD theses from the University of Leeds are cited as the source of data, namely Matthews PhD thesis (2014) for RH=65%, and Moon PhD thesis (2018) for RH=43%.

For the current paper (Song et al. ACPD, 2020), it seems that PhD theses have been used again for the data used in Figure 2. These two PhD theses are not mentioned in the paper, rather Moon et al (ACP, 2018) is cited, which cannot be correct.

There is also another issue regarding the University of Leeds data used in Song et al. 2020.

[Figure]

The Leeds data shown for RH=65% in Song et al. (ACPD, 2020) and in Zou et al. (CCL 2019) seem to be different for the 2 papers? In Figure 2 of Song et al (2020) the RH=65% HO2 uptake coefficient levels out at around 0.4 at high copper ion concentrations. However, in Zou et al al. (2019), the RH=65% data seem to level out around 0.25. These are clearly two different data sets, one of which is cited in Song et al. (ACPD, 2020) as coming from Moon et al (2018), which is incorrect, and one of which in Zou et al. (CCL, 2019) is cited as coming from the PhD Thesis of Matthews (2014).

The Leeds data in the Matthews thesis, the HO2 uptake coefficient as a function of copper ion concentration for AS aerosols at RH=65% have been published in Lakey et al (JPCA 2016) (see Figure 5 of that paper) and resemble the data shown in Zou et al (2019), but do not resemble the data for RH=65% that are shown in Song et al.. (2020), which appears to be from a different dataset.

Pascale S. J. Lakey, Ingrid J. George, Maria T. Baeza-Romero, Lisa K. Whalley and Dwayne E. Heard, Organics Substantially Reduce HO2 Uptake Onto Aerosols Containing Transition Metal ions, J. Phys. Chem. A, 2016, 120, 9, 1421-1430.

We can only think that the data shown in Fig 2 of Song et al (2020) for RH=65% have come from the PhD Thesis of D Moon (Leeds, 2018), data which have not been published, and these data may have been taken using different experimental conditions (e.g. HO2 concentration, aerosol-HO2 interaction time in the flow-tube, different liquid samples used etc.), and may require final analysis/checks etc. before publication.

Now turning to the Leeds HO2 uptake data at RH=43% shown in Figure 2 of Song et al (2020) and also in Zou et al (2019) as a function of copper ion concentration. These data have not been published, and as mentioned above, Moon et al., (ACP, 2018) which is cited by Song et al 2020 is an incorrect citation. These data appear to come from the PhD thesis of D Moon, and again these data have not been subject to the same final analysis/ checks etc. for journal publication.

The Song et al 2020 paper contains data from the University of Leeds PhD theses, and

although uptake data at RH=65 % have been published in Lakey et al (2016), these are not the data shown in Fig 2 of Song et al (2020). Use seems to have been made in Song et al (2020) at both RH=43% and RH=65% of data from the thesis of D Moon. The data used by Song et al. (2020) at RH=65% should be those from Lakey et al., (2016), and not from the Moon PhD thesis, and also it would be sensible for the data at RH=43% to be used following consultation with Leeds, as final analysis and checks on these data may need to be performed for publication. Indeed, manuscripts are in preparation which will use these data.

Finally, on an unrelated point, the value given in Table 1 for the uptake coefficient of HO2 onto (NH4)2SO4 by George et al 2013 is correct in the table at 0.4, but the uncertainty given in Table 1 is +/-0.2 whereas in George et al the uncertainty reported is +/-0.3. The concentration of copper is as well reported in George et al. as molal (Mol kg-1) rather than in molar, M (Mol litre-1).

I. J. George, P. S. J. Matthews, L. K. Whalley, B. Brooks, A. Goddard, M. T. Baeza-Romero and D. E. Heard, Measurements of uptake coefficients for heterogeneous loss of HO2 onto submicron inorganic salt aerosols, Phys. Chem. Chem. Phys., 2013, 15, 12829.

---

## Short Comment (SC2) · 28 Mar 2020

I am not an assigned reviewer of this manuscript, but was curious to learn about its topic.

1) Being a kineticist, I looked at the Table 2 (Kinetic data for the simulation of reactions in aqueous aerosols). I found that Table confusing:

1a) Reaction R2 is given as $Cu^+ + 2\,H^+ + O_2^- \rightarrow Cu^{2+} + H_2O_2$

Is the rate expression for this reaction $v = k[Cu^+][H^+]^2[O_2^-]$? If so, that should be made clear, but the reaction would be unimportant given the value of the rate constant. It seems more likely that the rate expression is $v = k[Cu^+][O_2^-]$; if so, then perhaps the reaction should be written:   $Cu^+ + (2\,H^+) + O_2^- \rightarrow Cu^{2+} + H_2O_2$

with a note on the Table specifying that species in parentheses do not contribute to the rate equation.

Similar questions apply to R5.

1b)     Reaction R9 has $O_2^-$ reducing $Cu^{2+}$ to $Cu^+$ while reaction R2 has $O_2^-$ oxidizing $Cu^+$ to $Cu^{2+}$, both with rate constants at the diffusion limit. Perhaps I am exhibiting my ignorance of aqueous-phase chemistry, but I find this hard to believe.

1c) Reactions R1 and R3 do not exhibit mass balance.

On another topic: Perhaps the title shou

Some minor points:

a) In my experience "K" (upper case) is used for equilibrium constants and "k" (lower case) for rate constants, while the present manuscript uses the opposite convention.

b) Table 3 appears to have equilibrium data but no kinetic data. Also, I assume that the redox chemistry of iron is included in the model, even though it is not included in Table Perhaps the table should be retitled something like "Equilibria for copper and $HO_x$ chemistry in aqueous aerosols"

c) I noticed on line 94 that "ironic" is used for "ionic"

---

## Referee Comment (RC1) · Anonymous Referee #1 · 4 Apr 2020

Review of "Influence of aerosol copper on HO2 uptake: A novel parameterized equation" by Song et al.

The authors present a modeling study of the HO2 uptake coefficient as a function of aerosol copper concentration and relative humidity. Their methods seem thorough and robust for liquid particles containing ammonium sulfate and copper. They are able to explain the previous large discrepency between experimental measurements and models, which is extremely useful in helping to push our understanding of HO2 uptake coefficients forwards. They also provide a parameterization, based on their more complex model, which could easily be implemented in regional and global models. They are able to show that a wide range of HO2 uptake coefficients are likely to be required for atmospheric aerosols, rather than the fixed value which is often currently used in

models. However, I feel that there are many places throughout the manuscript where clarifications and additional details would be helpful. The authors could also state the limitations of their model and their implications more clearly. The manuscript would also benefit from being proof read. Overall, this is a really useful study, which should be published after the comments below have been addressed.

Comments: 1. Please clarify how the ionic strength in Equation 4 is calculated. Is this calculated in the MARK model and what would a typical value be?

2. For all tables please add units where these are missing.

3. In Table 4 what are the values of kmt or how are these calculated?

4. Are all reactions in the tables included in the model? If so how does this relate to keff in Equation 11?

5. It's stated that in the model it is assumed that the surface concentration and the bulk concentration equal each other. It is also stated that this is only valid for particles with a radius less than 200 nm. However, the model and resulting parameterization are then applied to particles which are larger than this and many particles in the atmosphere are larger than this. I wonder why the authors don't seem to have used the correction in equations 10 and 11 and what impact this will have on their final results and the applicability of their parameterization to future studies?

6. In Figure 2 what is the main cause for the decrease in the uptake coefficients between the original parameterization and the new model results. Is the difference mainly due to the different rate coefficient being used, the use of activity coefficients or something else?

7. For Figure 2 what is the aerosol pH and how is it calculated?

8. Figure 2 seems to be missing some previously published data point(s) from Lakey et al., JPCA (2016). It seems that the point at the highest copper concentration in that work (which is not shown in Figure 2) would not fit the modeled line. The authors

should include any previously published missing points for completeness. Are they able to model or at least speculate as to why this data point does not fit their model.

9. George et al. PCCP (2013) noticed higher uptake coefficients for lower HO2 concentrations for copper doped particles. Did the authors do any sensitivity tests with different HO2 concentrations and do they see any difference?

10. Figure 3: Is lg on y axis log10?

11. Figure 3: Please explain this figure in a more detailed fashion. It is unclear to me what the markers are and why there is a range of values. Why does there seem to be a larger difference between the model and the parameterization at low relative humidity?

12. Figure 4: Is the data shown measurements or a simulation?

13. Figure 5: This figure is not mentioned at all in the text and as such I don't know what the difference is between Figures 4 and 5.

14. Why not combine Figures 4 and 5 for better comparisons?

15. Can the authors speculate as to why the HO2 uptake coefficient is higher at night (Figure 6)?

16. In Figure 6 what is the main cause of the distribution in HO2 uptake coefficients? Is it due to different copper concentrations in the particles or something else?

17. Were any HO2 measurements made during the Wangdu field campaign and if so was any box modeling of the Wangdu campaign performed to determine whether there was a discrepency between measured and modeled HO2 uptake coefficients? Were predicted HO2 uptake coefficients in the range that was expected? If no HO2 measurements were made, could the authors clarify why they chose this particular field campaign to apply their model to?

18. Line 313: The authors may want to clarify that 'aerosol properties' may include phase state and that previous measurements have shown lower uptake coefficients

for semi-solid and solid particles (e.g. Lakey et al. ACP (2016)). The authors should also clarify somewhere that one of the major limitations of their model is that they assume steady-state concentrations and do not consider concentration gradients which will occur and could change over time for semi-solid particles.

19. The authors fix the solubility of copper at 25%. In reality solubility can vary considerably. How sensitive is this parameter in their model?

20. Another limitation of the model is that they don't consider reactions between different metal ions (such as Reaction 4 in Mao et al. ACP (2013)) which they have stated. However, could they also speculate how this could impact the estimated uptake coefficients for atmospheric aerosols (e.g. is it expected that this would increase the uptake coefficient)?

21. Please check the references carefully as many seem to wrong (e.g. Schwartz and Meyer 1986 line 108 and references in Figure 2).

---

## Author Comment (AC1) · 9 Apr 2020

Response to Comment of Dwayne Heard

We thank Professor Heard a lot for the critical comments on the Leeds data of Figure 2 especially those related to the unpublished data from the Moon PhD thesis which we took previously from an online website. Since his comment was submitted, we have then made personal communications with Prof. Dwayne Heard. So now the finalized data from Moon's PhD thesis for RH=43% are provided directly from Prof. Dwayne Heard and we will add the Leeds providers of the data as co-authors on the paper.

In the following, detailed responses are given so now both the data and the citation of both Figure 2 and Table 1 are up to date according to the comments.

**Comment:**
The paper uses laboratory measurements of $HO_2$ uptake coefficients experimentally obtained by the University of Leeds for copper ion doped ammonium sulphate (AS) aerosols, and data from other groups. However, the origin of the data from the University of Leeds is not clear and the citation for the data is not correct. The Song et al 2020 paper contains data from the University of Leeds PhD theses, and although uptake data at RH=65 % have been published in Lakey et al (2016), these are not the data shown in Fig 2 of Song et al (2020). Use seems to have been made in Song et al (2020) at both RH=43% and RH=65% of data from the thesis of D Moon. The data used by Song et al. (2020) at RH=65% should be those from Lakey et al., (2016), and not from the Moon PhD thesis, and also it would be sensible for the data at RH=43% to be used following consultation with Leeds, as final analysis and checks on these data may need to be performed for publication. Indeed, manuscripts are in preparation which will use these data.

**Response:**
Figure 2 is now updated based on the Leeds data provided by Prof. Heard on 3rd April 2020 for the new data at RH=43% and the published data at RH=65% from Lakey et al., 2016. Since the simulation results only show the copper ion concentration from $10^{-5}$ M to 1M concentration, the first point of the uptake coefficient at 65% RH measured by Lakey et al (2016) at $10^{-6}$ M is not shown in the picture, and so there are six data points shown from Lakey et al., (2016). The five data points for 43% RH are all new in Figure 2 and were measured by aerosol flow tube experiments and provided by Leeds. These data are now labelled "This work" in Figure 2 and Table 1 as the providers of the data from the Univesity of Leeds will now be added to the paper as co-authors".

[Figure]

Figure 2: Dependence of $\gamma_{HO_2}$ on aerosol copper ion concentration. Red circles denote the results at 43% RH measured at the University of Leeds using an aerosol flow tube experiment. Blue hollow circles at 65% RH were measured by Lakey et al., (2016). Yellow diamonds denote results at 51% RH measured by Zou et al., (2019). Pink triangle at 42% RH was measured by Thornton and Abbatt, (2005). Green star at 45% RH was measured by Taketani et al.(2008). The grey dotted line denotes the current parameterized equation of which the influence of RH was not considered ( e.g. Hanson et al., (1994)) and the grey solid lines represent the model results of MARK model in this study for a range of RH. The root mean square error (RMSE) between the modeled values and the full dataset in the figure is 0.30 indicating a modest deviation in MARK model calculations.

And in Table 1, the citations are also corrected:

Table 1: $\gamma_{HO_2}$ under lab conditions for Cu(II)-doped inorganic aerosols.

| Aerosol type | RH/% | Estimation of [Cu] in aerosol/M | $\alpha$ | Ref. |
|---|---|---|---|---|
| NH$_4$HSO$_4$ | 75% | 0.0059−0.067[*] | 0.40±0.21 | (Mozurkewich et al., 1987) |
| (NH$_4$)$_2$SO$_4$ | 45% | 0.5 | 0.53±0.13 | (Taketani et al., 2008) |
| (NH$_4$)$_2$SO$_4$ | 42% | 0.16 | 0.5±0.1 | (Thornton and Abbatt, 2005) |
| (NH$_4$)$_2$SO$_4$ | 53−65% | 0.5−0.7[*] | 0.4±0.3 | (George et al., 2013) |
| (NH$_4$)$_2$SO$_4$ | 65% | 0.57 | 0.26±0.02 | (Lakey et al., 2016) |
| (NH$_4$)$_2$SO$_4$ | 51% | 0.0027 | 0.096±0.024 | (Zou et al., 2019) |
| (NH$_4$)$_2$SO$_4$ | 43% | 0.38 | 0.355±0.023 | This work |
| NaCl | 53% | ~0.5 | 0.65±0.17 | (Taketani et al., 2008) |
| KCl | 75% | 5% of KCl solution | 0.55±0.19 | (Taketani et al., 2009) |
| LiNO$_3$ | 75% | 0.03−0.0063[*] | 0.94±0.5 | (Mozurkewich et al., 1987) |

[*]Cu concentration is in molal (Mol kg$^{-1}$).

**Comment:**

Finally, on an unrelated point, the value given in Table 1 for the uptake coefrient of $HO_2$ onto $(NH_4)_2SO_4$ by George et al 2013 is correct in the table at 0.4, but the uncertainty given in Table 1 is +/-0.2 whereas in George et al the uncertainty reported is +/-0.3. The concentration of copper is as well reported in George et al. as molal (Mol kg-1) rather than in molar, M (Mol litre-1).

**Response:**

The units and the uncertainty are corrected accordingly. Please see Table 1 above. The measurements of Mozurkewich et al., 1987 are also in Molal units, and this is also now indicated in the Table.

**References**

George, I. J., Matthews, P. S. J., Whalley, L. K., Brooks, B., Goddard, A., Baeza-Romero, M. T., and Heard, D. E.: Measurements of uptake coefficients for heterogeneous loss of HO2 onto submicron inorganic salt aerosols, Physical Chemistry Chemical Physics, 15, 12829-12845, 2013.

Hanson, D. R., Ravishankara, A. R., and Solomon, S.: HETEROGENEOUS REACTIONS IN SULFURIC-ACID AEROSOLS - A FRAMEWORK FOR MODEL-CALCULATIONS, J. Geophys. Res.-Atmos., 99, 3615-3629, 10.1029/93jd02932, 1994.

Lakey, P. S. J., George, I. J., Baeza-Romero, M. T., Whalley, L. K., and Heard, D. E.: Organics Substantially Reduce HO2 Uptake onto Aerosols Containing Transition Metal ions, Journal of Physical Chemistry A, 120, 1421-1430, 10.1021/acs.jpca.5b06316, 2016.

Mozurkewich, M., Mcmurry, P. H., Gupta, A., and Calvert, J. G.: Mass Accommodation Coefficient for Ho2 Radicals on Aqueous Particles, Journal of Geophysical Research-Atmospheres, 92, 4163-4170, 1987.

Taketani, F., Kanaya, Y., and Akimoto, H.: Kinetics of heterogeneous reactions of HO2 radical at ambient concentration levels with (NH4) 2SO4 and NaCl aerosol particles, The Journal of Physical Chemistry A, 112, 2370-2377, 2008.

Taketani, F., Kanaya, Y., and Akimoto, H.: Heterogeneous loss of HO2 by KCl, synthetic sea salt, and natural seawater aerosol particles, Atmos. Environ., 43, 1660-1665, 2009.

Thornton, J., and Abbatt, J. P. D.: Measurements of HO2 uptake to aqueous aerosol: Mass accommodation coefficients and net reactive loss, J. Geophys. Res.-Atmos., 110, 10.1029/2004jd005402, 2005.

Zou, Q., Song, H., Tang, M., and Lu, K.: Measurements of HO2 uptake coefficient on aqueous (NH4)2SO4 aerosol using aerosol flow tube with LIF system, Chinese Chemical Letters, https://doi.org/10.1016/j.cclet.2019.07.041, 2019.

---

## Referee Comment (RC2) · Anonymous Referee #2 · 14 Apr 2020

The manuscript of Song et al deals with an important theme in atmospheric science; the interaction of HO2 with particles containing dissolved copper and the modelling of the impact of this heterogeneous reaction on e.g. O3 production.

Song et al have analysed laboratory data and derived an empirical expression that they then implemented in a model. They suggest that their parameterisation is superior to taking a constant value of 0.2 for the uptake coefficient. This is most likely true but why do they not compare to other parameterisations of this process, e.g. that proposed by IUPAC which also considers laboratory studies up to 2016 and which has a physical basis (i.e. aqueous phase rate coefficients for reaction of HO2 / O2- with Cu).

They also do not consider in any detail the potential for organic content of particles to reduce the uptake coefficient considerably as this is mentioned only in passing on

line 312-315. Thus aspect will be central to improving the modelling of HO2 uptake to particle matter, as has been shown e.g. for N2O5. The sentence on line 149-150 in this regard is highly misleading: "Without the interference of organic matter, it is convenient to explore the mechanism of HO2 heterogeneous uptake and derive its parameterized equation, which provides a good reference for the heterogeneous uptake of HO2 in the actual atmosphere environment". The co-authors from Jülich will be intimately familiar with this aspect of heterogeneous chemistry and should be able to provide input.

The manuscript is illogically organised and difficult to follow, has different definitions of the same term, poor language quality and incorrectly cites the published datasets of Lakey et al (2016) (see author comment from D. Heard). This work might be publishable after major revision and reorganisation.

Some specific points are listed below:

L11 hydroxyl peroxy radicals = hydroperoxyl radicals L12 gamma_HO2 is defined but the expression after line 17 simply lists gamma. L15 parameterisation of HO2 ? Presumably of HO2 uptake

Does it make sense to list the expression in the abstract, where none of the terms (e.g. ALWC, [PM], Rd) involved are defined

L39-40 Meaning obscure. I think the authors refer to the reduction of aerosol mass over the last few years. The information that the uptake coefficient used (in calculating surface ozone) was 0.2 is superfluous here.

L57 L is defined as the aerosol liquid water content. In the abstract it is ALWC.

L68 MARKM model is otherwise referred to as MARK

L74 Equilibrium constant have capital "K". Rate coefficients have lower case "k".

L/3 H_0 is estimated (Thornton et al, 2008) to be 3900 M atm-1. Why is this cited in different units to the effective solubility (Hˆcc). How good is this "estimate" and on what

data is it based (I believe Hanson 1992, who also lists a T-dependence) ?

L101 Define [xi]_equ. In the line above only [xi] is mentioned.

L105 "steady-state" HO2 concentration. Why "steady-state" ?

L110 k_eff is listed in the equation 11. This appears to be defined 57 lines later, but not always in the same manner.

Tab1 (and abstract) the accommodation coefficient is not defined, as far as I can see.

Tab1 The data of Lakey 2015/2016 and Zou 2019 are not mentioned (Moon = Lakey ???)

Tab2 Add units, do not capitalize K and move to supplementary information Tab3 K should be capitalized (eqm.)

L146 annual average contribution. Does this refer to a global average ?

Fig1 Please explain why the uptake coefficient continues to increase at pH > 5 whereas k_eff decreases.

L166 k_eff is defined as a comprehensive reaction rate constant. . .. . .during heterogeneous uptake. On L210, K_eff is defined as the rate of HO2 aqueous reaction with copper ions.

L175 The parameterisations listed do not include that of IUPAC, which includes more recent laboratory data than those listed.

L180 Based on the data of Lakey et al, IUPAC list a rate coefficient for HO2 (O2-) with Cu ions of 5x10^5. This is orders of magnitude lower that those listed.

Fig2 Why are the data of Lakey not listed (or are these Moon ??)

Fig2 Taketani also have uptake coefficients in the absence of Cu that are just as high as the single point at about 0.5 M. Why are the other datapoints of Taketani selectively omitted here?

L188 what is the "current parameterised equation". From whom is it?

L201 This text ignores the fact that the IUPAC parameterisation accurately reproduces the lab studies from Lakey et al. Is the empirical parameterisation that Song et al propose really superior to the IUPAC one, which has a physical basis?

L213 where does this expression for the uptake coefficient come from?

L265 section 3.4.2. could be move to SI

L312 The work of Lakey et al is cited. Where are their data? How much "lower" can the uptake coefficient be in the presence of organics?

L351 move section 3.4.5 to SI

---

## Author Comment (AC2) · 17 Apr 2020

Response to Comment of Theodore S. Dibble

We thank a lot for the short comment by Prof. Theodore S. Dibble on the chemical reaction equations and rates in Table 2 and Table 3.

Comment:
1) Being a kineticist, I looked at the Table 2 (Kinetic data for the simulation of reactions in aqueous aerosols). I found that Table confusing:

1a) Reaction R2 is given as $Cu^+ + 2H^+ + O_2^- \rightarrow Cu^{2+} + H_2O_{2(aq)}$.
Is the rate expression for this reaction $v = k[Cu^+][H^+]2[O_2^-]$? If so, that should be made clear, but the reaction would be unimportant given the value of the rate constant. It seems more likely that the rate expression is $v = k[Cu^+]2[O_2^-]$; if so, then perhaps the reaction should be written: $Cu^+ + (2H^+) + O_2^- \rightarrow Cu^{2+} + H_2O_{2(aq)}$ with a note on the Table specifying that species in parentheses do not contribute to the rate equation.
Similar questions apply to R5.

1c) Reactions R1 and R3 do not exhibit mass balance.

Response:
Thanks a lot for pointing out the possible problems in the chemical reaction equations. The box model ensures the mass balance of interrelated chemical reactions. Reaction rates of R1, R2 and R5 are not affected directly from the concentration of $H^+$ ion while the ratio of different valence state of the copper ion is affected by the solution acidity. In Table 2, R1, R2, R3 and R5 are corrected accordingly.

Table 2: Kinetic data for the simulation of reactions in aqueous aerosols.

| No. | Reactions | $k_{298}$ | $E_a$ |
|------|-----------|-----------|-------|
| R1 | $Cu^+ + HO_{2(aq)} + (H^+) \rightarrow Cu^{2+} + H_2O_{2(aq)}$ | $2.2 \times 10^9$ | |
| R2 | $Cu^+ + (2H^+) + O_2^- \rightarrow Cu^{2+} + H_2O_{2(aq)}$ | $9.4 \times 10^9$ | |
| R3 | $Cu^+ + OH_{(aq)} \rightarrow Cu^{2+} + OH^-_{(aq)}$ | $3 \times 10^9$ | |
| R5 | $Cu^+ + (H^+) + O_{3(aq)} \rightarrow Cu^{2+} + O_{2(aq)} + OH_{(aq)}$ | $3 \times 10^7$ | |

Comment:
1b) Reaction R9 has $O_2^-$ reducing $Cu^{2+}$ to $Cu^+$ while reaction R2 has $O_2^-$ oxidizing $Cu^+$ to $Cu^{2+}$, both with rate constants at the diffusion limit. Perhaps I am exhibiting my ignorance of aqueous-phase chemistry, but I find this hard to believe.

Response:
$O_2^-$ can be catalytically dismutated by metals and metal complexes thus can function either as a univalent oxidant or reductant with transition metal ions such as iron, copper and manganese (Bielski and Cabelli, 1991). Among the transition metals, Cu is the most likely catalytic sink of $O_2^-$. The rate constant of R2 and R9 are from CAPRAM 2.4 and Jacob, 2000 based on the aerosol bulk estimated pH range (Ervens et al., 2003; Jacob, 2000).

Comment:

Some minor points:

a) In my experience "K" (upper case) is used for equilibrium constants and "k" (lower case) for rate constants, while the present manuscript uses the opposite convention.

b) Table 3 appears to have equilibrium data but no kinetic data. Also, I assume that the redox chemistry of iron is included in the model, even though it is not included in Table Perhaps the table should be retitled something like "Equilibria for copper and HOx chemistry in aqueous aerosols"

c) I noticed on line 94 that "ironic" is used for "ionic"

Response:

a) Rate symbol "K" (upper case) and "k" (lower case) in Table 2 and Table 3 are corrected accordingly.

b) The redox chemistry of iron is not included in the present model for the diagnosis of copper influence but we indeed studied the iron chemistry in separate model runs which don't have observations to compare. Nevertheless, we think your proposed tile for Table 3 is more accurate and we changed as suggested.

c) We revised accordingly.

Bielski, B., and Cabelli, D.: Highlights of current research involving superoxide and perhydroxyl radicals in aqueous solutions, International journal of radiation biology, 59, 291-319, 1991.

Ervens, B., George, C., Williams, J. E., Buxton, G. V., Salmon, G. A., Bydder, M., Wilkinson, F., Dentener, F., Mirabel, P., Wolke, R., and Herrmann, H.: CAPRAM 2.4 (MODAC mechanism): An extended and condensed tropospheric aqueous phase mechanism and its application, J. Geophys. Res.-Atmos., 108, 10.1029/2002jd002202, 2003.

Jacob, D. J.: Heterogeneous chemistry and tropospheric ozone, Atmos. Environ., 34, 2131-2159, 10.1016/s1352-2310(99)00462-8, 2000.

---

## Author Comment (AC4) · 21 Jul 2020

We thank the reviewer for his/her helpful comments. The referee's comments are first given in black type, followed by our response to each in turn in blue type. Any changes to the manuscript in response to the comments are then given in quotation marks in red type and the line number isin the Microsoft-word version of revised MS without revision.

Our detailed reponse is shown as the attached file.

Please also note the supplement to this comment:
https://www.atmos-chem-phys-discuss.net/acp-2020-218/acp-2020-218-AC4-supplement.pdf

[Figure]

**Supplement:**

Response to the comments of referee #1:

We thank the reviewer for their helpful comments. The referee's comments are first given in black type, followed by our response to each in turn in blue type. Any changes to the manuscript in response to the comments are then given in quotation marks in red type and the line number isin the Microsoft-word version of revised MS without revision. The line number may be different in the PDF version, so please see the section number mentioned in the Response. Figure Response 1 and Figure Response 2 only appear in the Responses and not the revised MS, in order to illustrate the responses to the referee comments.

We have now added D. Moon, M. Baeza-Romero and D. Heard as co-authors to this paper since their unpublished experimental data have been included in this paper and they have contributed to enhance the manuscript.

Comment:

1. Please clarify how the ionic strength in Equation 4 is calculated. Is this calculated in the MARK model and what would a typical value be?

Response:

The ionic strength ($I$, mol L$^{-1}$), is calculated in the model via Equation 8 in section 2.2.2 on line 124. Typical values for $I$ are in the range of 2.16 – 17.75 mol L$^{-1}$ based on the ion concentrations in the aerosol bulk and the $RH$ ranging from 40% - 90%.

Comment:

2. For all tables please add units where these are missing.

Response:

We added the units in Table 2, 3, 4 and 5 which is now Table S. 1, S. 2, S. 3, and S. 4 in the Supplementary Information of the revised MS.

Comment:

3. In Table 4 what are the values of $k_{mt}$ or how are these calculated?

Response:

We added more information about the definition and importance of $k_{mt}$ in part 2.1 on line 78 around Equation (3):

"An approach to combine both gas phase molecular diffusion and liquid phase interface mass transport processes is through one variable called $k_{mt}$ (Schwartz, 1984;Schwartz, 1986), which is used in the calculation for gas-liquid multiphase reactions in many modelling studies (Lelieveld and Crutzen, 1991;Chameides and Stelson, 1992;Sander, 1999;Hanson et al., 1994). The definition of $k_{mt}$ is:

$$k_{mt} = (\frac{R_d{}^2}{3D_g} + \frac{4R_d}{3v_{HO_2}\alpha})^{-1} \qquad (3)$$

$k_{mt}$ is used to connect the gas phase reactions and the aerosol condensed phase reactions. The rate of gas phase reactants ($X$) diffusing and dissolving to the condensed phase can be calculated in the framework of aqueous phase reactions as $k_{mt\_X} \times ALWC$ (where $X$ is the reactant molecule). Moreover, the conversion rate of aqueous phase reactants to gas phase can be calculated as $\frac{k_{mt\_X}}{H^{cc} \times RT}$ where $H^{cc}$ is the effective Henry's law constant [M atm$^{-1}$]. The unit of $k_{mt}$ is s$^{-1}$, as $k_{mt}$

contains the conversion from $m_{air}^{-3}$ of the gas phase molecule concentrations to $m_{aq}^{-3}$ of the aqueous phase molecule concentrations and backward. For larger particles (radius >1μm), $k_{mt}$ is mainly determined by gas phase diffusion of $HO_2$. For smaller particles (radius <1μm) $k_{mt}$ is mainly determined by the accommodation coefficient ($\alpha$). The MARK model can simultaneously simulate gas and liquid two-phase reaction systems in the same framework."

The typical value of $k_{mt}$ of $HO_2$ for small particles with the radius of 50 nm is $3.85 \times 10^5$ s$^{-1}$, and for larger particles with the radius of 1 μm is $1.93 \times 10^4$ s$^{-1}$.

Comment:

4. Are all reactions in the tables included in the model? If so how does this relate to keff in Equation 11?

Response: All reactions in the tables are included in the model. $k_{eff}$ is the comprehensive reaction rate coefficient encompasses both $HO_2$ dissolution equilibrium reactions and liquid phase chemical-physical reactions during $HO_2$ uptake process. $k_{eff}$ is mostly affected by the concentration of copper, the $HO_2/O_2^-$ cycle in the liquid phase and $HO_2$ dissolution equilibrium. $HO_2/O_2^-$ reacting with Cu ions (R 1, R 2, R 8 and R 9) may directly affect $k_{eff}$ thus change the model result of $\gamma_{HO_2}$. $OH_{(aq)}$, $O_{2(aq)}$, $O_{3(aq)}$, $H_2O_{2(aq)}$ will also influence the reaction system because they have direct relationship with the dissolution equilibrium and reactions of $HO_2$, $O_2$, $H_2O_2$ OH and $O_3$. $H_2O_2$ is also a reaction product of $HO_2/O_2^-$ reacting with Cu ions, its concentration will also affect the reaction system. Sulfate ion, ammonium ion and nitrate ion may not directly affect the system, while they will change the concentration of aqueous $HO_2$, $O_2$, $H_2O_2$, OH, $O_3$ and their solubility. What ismore, $HO_2$ uptake process may influence the oxidation rate of $SO_2$ and other reagents, to which more research is needed.

Comment:

5. It's stated that in the model it is assumed that the surface concentration and the bulk concentration equal each other. It is also stated that this is only valid for particles with a radius less than 200 nm. However, the model and resulting parameterization are then applied to particles which are larger than this and many particles in the atmosphere are larger than this. I wonder why the authors don't seem to have used the correction in equations 10 and 11 and what impact this will have on their final results and the applicability of their parameterization to future studies?

Response:

The size of the particles is an important factor within the MARK model when considering gas phase diffusion to the particle surface and $HO_2$ desorption. While the $HO_2$ radical concentration is not a factor that influence $\gamma_{HO_2}$ in the MARK model.

In ambient urban situations, the Count Median Diameter ($R_d$) of aerosol particles is smaller than 1μm in most instances. The ratio of $\overline{[HO_2]}$ to $[HO_{2(r)}]$ is 0.89 calculated by the MARK model simulation of $k_{eff}$ with a RH 40%-90%, the copper ion concentration varies from $10^{-5}$ to 1M at 1μm diameter particles. The ratio will be even higher and close to 1 with smaller particles (>0.95 at 400 nm diameter). The MARK model is valid for particles with $R_d$ smaller than 400 nm and may cause small deviations for particles with $R_d$ smaller than 1μm. Thus, in this paper, we assume the surface concentration of $HO_2$ equals to the condensed phase average $HO_2$ concentration.

A model considering the influences of aerosol particle size distribution and $HO_2$ concentration gradients on $\gamma_{HO_2}$ is currently under development.

We changed the statement in the origin MS in part 2.2.3 on line 148 and modified as: "In the copper-doped aerosol particle, because of the high value of $k_{eff}$ and small Count Median Diameter ($R_d$) (usually smaller than 1μm), the ratio is close to 1. At a diameter of 1μm, and a relative humidity between 40% and 90%, the condensed phase copper ion concentration varies from $10^{-5}$ to 1M, the average ratio of the surface $HO_2$ concentration and the condensed phase $HO_2$ concentration is 0.89. At 400nm diameter particles for $RH$ = 40% to 90%, the ratio is larger than 0.95. The ratios are calculated by simulation of $k_{eff}$ and the accordingly calculations by Equation (12) and (13)."

Comment:

6. In Figure 2 what is the main cause for the decrease in the uptake coefficients between the original parameterization and the new model results. Is the difference mainly due to the different rate coefficient being used, the use of activity coefficients or something else?

Response:

The parameterization proposed by the IUPAC uses only one rate constant as the second-order reaction rate $k^{II}$ of $Cu^{2+}$ and $HO_2$. We use $1.5\times10^7$ L $mol^{-1}$ $s^{-1}$ as the secondary reaction rate $k^{II}$ rather than the more commonly used value of $1\times10^9$ L $mol^{-1}s^{-1}$ in the calculation of the original parameterization.

We added the reason of using $1.5\times10^7$ L $mol^{-1}$ $s^{-1}$ in the calculation in the revised MS in Section 3.2 on line 241:

"The prior value ($1.5\times10^7$ $M^{-1}$ $s^{-1}$) reflects the rate of reaction between $HO_2$ and $Cu^{2+}$, more prevalent in acidic aerosol such as ammonium sulphate, and the latter between $O_2^-$ and $Cu^{2+}$ ions, which is more prevalent in aerosols with a pH greater than the $pK_a$ of $HO_2$, such as NaCl (Bielski et al., 1985). This treatment within the calculation can bring predictions more in line with experimental results (Figure 2 grey dotted line) compared to the high value of $1\times10^9$ L $mol^{-1}s^{-1}$ used in the existing parameterized equation."

And on line 256 the main reason of the differences between original parameterization and the MARK model:

"The main reason for the differences between the original parameterization and the MARK model is the effect of including the activity coefficients of Cu ion and HO2 and the effects of reactions of different valence states of copper ions."

7. For Figure 2 what is the aerosol pH and how is it calculated?

Response:

The main components of the aerosols used in the laboratory measurements of $\gamma_{HO_2}$ shown in Figure 2 are ammonium sulfate and a small amount of copper sulfate. According to the calculation based on the aqion 7.0.8 interface (for details please see https://www.aqion.de/), the pH is around 4.54 considering aerosol dehydration with 2M aerosol bulk sulfate concentration and 1M Cu ion at 25°C. In the MARK model, we set aerosol pH as 4.5 when compared to the laboratory results. We have re-calculated $\gamma_{HO_2}$ at pH=4.5 and added data points from Lakey et al., JPCA (2016) based on the short comments from Pro. Heard. In the revised MS, Figure 2 is modified as follows:

[Figure]

"Figure 2: Dependence of $\gamma_{HO_2}$ on aerosol copper concentration. Red filled circles denote the results at 43% *RH* measured at Leeds. Blue hollow circles at 65% *RH* (Lakey et al., 2016). Yellow filled diamonds denote results at 51% *RH* (Zou et al., 2019), filled purple triangle at 42% *RH* (Thornton and Abbatt, 2005)) and filled green star at 45% *RH* (Taketani et al., 2008)). The grey dotted line denotes the current parameterized equation (Thornton et al., 2008;Hanson et al., 1992;Hanson et al., 1994;Jacob, 2000;Kolb et al., 1995) and the solid grey lines represent the model results of MARK model in this study at various *RH*. The root mean square error (RMSE) between the MARK modelled values and the full dataset (0.23). Aerosol pH is set as 4.5 based on aqion 7.0.8 interface considering the participation of Cu ion (for details please see https://www.aqion.de/)."

Comment:

8. Figure 2 seems to be missing some previously published data point(s) from Lakey et al., JPCA (2016). It seems that the point at the highest copper concentration in that work (which is not shown in Figure 2) would not fit the modeled line. The authors should include any previously published missing points for completeness. Are they able to model or at least speculate as to why this data point does not fit their model.

Response:

We have now added these data points from Lakey et al., JPCA (2016) at the highest and the lowest copper concentration used to Figure 2 for completeness. Please also see the response to the Short Comment from Professor Heard, University of Leeds. Prof. Heard, Dr. Moon and Dr. M. Baeza-Romero from Leeds are now added as co-authors to the paper, with the correct data from Lakey et al., (2016) used for *RH*=65% and from the PhD thesis of Dr. Moon for *RH*=43% included. Moreover, in order to model the results from Lakey et al., JPCA (2016), we change the accommodation coefficient to 0.26 in the MARK model as they recommended in the paper, and get the result as follows:

[Figure]

Figure Response. 1 Comparison of the dependence of $\gamma_{HO_2}$ on aerosol copper ion concentration with $\alpha_{HO_2}$ as 0.26 and 0.5 at 65% *RH*. The solid line denotes the MARK model results, and the blue hollow circles denote the results at 65% *RH* measured by Lakey et al. (2016).

Model results using $\alpha_{HO_2}$ as 0.5 fits well with the results from Lakey et al. (2016) when $[Cu^{2+}]$ smaller than 0.1M while have larger deviation for higher $[Cu^{2+}]$. $\alpha_{HO_2}$ as 0.26 fits unsatisfactorily at $[Cu^{2+}]$ around 0.01M. Considering the ambient aerosol condensed phase $[Cu^{2+}]$ (for example, ranging from 0.003 to 0.012 M in Wangdu campaign), in this MS we still use $\alpha_{HO_2}$ as 0.5 to get the novel $HO_2$ uptake parametrization.

The accommodation coefficient may change in the process of the experiments for some reasons. For example, the reaction time and $HO_2$ initial concentration (see next comment by the reviewer). Aerosol phase state is also an important factor influenced $\alpha_{HO_2}$. Moreover, during the efflorescence of aerosol particles in drying nafion tube, different concentrations of copper ions may have a certain effect on particle phase state which will influence $\alpha_{HO_2}$. This part should be further studied.

9. George et al. PCCP (2013) noticed higher uptake coefficients for lower $HO_2$ concentrations for copper doped particles. Did the authors do any sensitivity tests with different $HO_2$ concentrations and do they see any difference?

Response:

The $\gamma_{HO_2}$ measured in the flow tube experiments indeed depends both on the $HO_2$ concentration and also the reaction time between $HO_2$ and the aerosols.. However, sensitivity tests in the MARK model show no $\gamma_{HO_2}$ decreasing trend with increasing $[HO_2]_0$ in the absence of Cu ions, and $\gamma_{HO_2}$ will slightly increase with the $[HO_2]_0$ in the presence of Cu ions in the MARK model.

A possible explanation for the results from George et al. PCCP (2013) could be the Fenton-like reactions of Cu ions and $H_2O_2$ that is an additional source of $HO_2$. More $H_2O_2$ will be generated with greater light intensity and may accumulate along the flow tube with the reaction of $HO_2$ with aerosol for $H_2O_2$ is one product of $HO_2$ uptake. However, $H_2O_2$ and Cu ions reactions need to be of the same order of magnitude or no more than one magnitude lower than that of $HO_2$ reaction with Cu ions to make obvious differences in the measurement of $\gamma_{HO_2}$. In the MARK model, $H_2O_2$ only reacts with $Cu^+$ and the reaction rate constant is $7\times10^3$ L mol$^{-1}$ s$^{-1}$ which is too small to influence $\gamma_{HO_2}$ with the changes of $H_2O_2$ concentration, and so it may explain the lack of sensitivity of the uptake coefficient with $HO_2$ concentration.

10. Figure 3: Is lg on y axis log10?

11. Figure 3: Please explain this figure in a more detailed fashion. It is unclear to me what the markers are and why there is a range of values. Why does there seem to be a larger difference between the model and the parameterization at low relative humidity?

Response:

We have changed the y-axis label in accordance with the referee´s comment in Figure 3 in the revised MS. We calculated the RMSE of $\gamma_{HO_2}$ predicted by MARK and the corresponding calculated values from the new parameterized equation at different *RH* and Cu ion concentrations.

We added the explanation of the larger difference between the model and the parameterization at low relative humidity in line 290 on the second last paragraph of section 3.3 in the revised MS:

"The range of values shows the difference between the modeled data and parameterized equation data at different Cu concentration. At low *RH* and consequently relatively low ALWC, $\gamma_{HO_2}$ is more sensitive to $[Cu^{2+}]$ expecially at low $[Cu^{2+}]$ ($<10^{-4}$M). This sensitivity can not be fully represented in the parameterized equation. What is more, at low $[Cu^{2+}]$ and low *RH*, the value of $\gamma_{HO_2}$ is smaller than in other conditions, small changes of $\gamma_{HO_2}$ will cause larger RMSE values. "

Comment:

12. Figure 4: Is the data shown measurements or a simulation?

13. Figure 5: This figure is not mentioned at all in the text and as such I don't know what the difference is between Figures 4 and 5.

14. Why not combine Figures 4 and 5 for better comparisons?

Response:

Figure S. 1 shows the averaged particle surface-area size distribution (PSASD) and the particle number size distribution (PNSD) of aerosol measured in the Wangdu field campaign. We deleted Figure 5 in the original MS because the information of aerosol size distribution is redundant. In the SI of the revised MS, we have added Figure S.1 as follows:

[Figure]

Figure S.1: The dry-state particle number size distribution (PNSD) (black line) and particle surface-area size distribution (PSASD) (grey line) of aerosol for conditions encountered during the Wangdu field campaign.

15. Can the authors speculate as to why the $HO_2$ uptake coefficient is higher at night (Figure 6)?
Response:
We add the table below in the SI for revised MS as Table S. 5. Table S.5 shows the median and average values of the copper concentration, PM loading and $RH$ during the day and at night. During the day the copper concentration is larger, but the lower $RH$ may limit $\gamma_{HO_2}$. Thus, $k_{het}$, the quasi-first order reaction rate constant of $HO_2$ heterogeneous uptake is slightly higher at night compared to the daytime, contributing to the higher $\gamma_{HO_2}$ predicted at night.

Table S.5. The median and average values used in the calculation of $\gamma_{HO_2}$ in Wangdu

|  | Value | Cu [ng/m$^3$] | PM$_{2.5}$ mass [μg/m$^3$] | $RH$ [%] | $\gamma_{HO_2}$ | $k_{het}$[s$^{-1}$] |
|---|---|---|---|---|---|---|
| Day | median | 33.42 | 77.9 | 55.4 | 0.119 | 0.017 |
|  | average | 44.66 | 85.0 | 57.6 | 0.126 | 0.020 |
| Night | median | 19.01 | 70.6 | 68.9 | 0.134 | 0.021 |
|  | average | 34.16 | 67.9 | 67.4 | 0.147 | 0.023 |

16. In Figure 6 what is the main cause of the distribution in $HO_2$ uptake coefficients? Is it due to different copper concentrations in the particles or something else?
Response:
The distribution of $\gamma_{HO_2}$ is mainly due to the different copper concentrations and ambient $RH$. Although the PM mass is also a parameter in the empirical equation proposed by this MS, it shows small partial correlation on $\gamma_{HO_2}$. Figure S 2 below shows the partial correlation coefficient between $\gamma_{HO_2}$, field measured [$HO_2$], [OH], $TR_{HO2uptake}$ and $R_1$ with aerosol mass loading in Wangdu

campaign. This figure is now added in the SI of the revised MS.

[Figure]

"Figure S.2. Impact of the $HO_2$ uptake evaluated with the novel empirical equation for conditions encountered during the Wangdu field campaign. Partial correlation of logarithmic values of $TR_{HO2uptake}$ and $R_1$ with respect to aerosol loading were calculated. The partial correlation coefficient in panel (a) means that $TR_{HO2uptake}$ has a small partial correlation with aerosol loading. No partial correlation of $R_1$, $[HO_2]$ and $[OH]$ to aerosol loading is observed. The different coloured dots show different $[NO_2]$. Panel (c) is the distribution of $log_{10}R_1$."

17. Were any $HO_2$ measurements made during the Wangdu field campaign and if so was any box modeling of the Wangdu campaign performed to determine whether there was a discrepency between measured and modeled HO2 uptake coefficients? Were predicted HO2 uptake coefficients in the range that was expected? If no HO2 measurements were made, could the authors clarify why they chose this particular field campaign to apply their model to?

Response:

Yes, $HO_2$ and other radical concentration measurements were made during the comprehensive campaign in Wangdu.

We added the following statements in the revised MS in section 3.4.3 the second paragraph on line 347:

"Tan et al. (2017) had compared the measured and modelled OH, $HO_2$ and $RO_2$ radicals in the Wangdu campaign. However, in this paper, they did not discuss the influence of $HO_2$ uptake. A very recent publication (Tan et al., 2020) calculated $\gamma_{HO_2}$ in the Wangdu campaign based on the comparison of field measurement data for $HO_2$ and concentrations calculated by the box model. The paper proposes that all $\gamma_{HO_2}$ calculated in this way from the Wangdu campaign can be fitted to a Gaussian distribution around the value of $0.08 \pm 0.13$. This value is similar to our estimation in this paper considering the indirect measurement uncertainty (please see the SI)."

What is more, we recalculated the ratio ($R1$) of $HO_2$ uptake to ROx termination rate with measured $RO_2$ concentration which is now in consist to the dataset of (Tan et al., 2020). In the original MS, we used the modeled $RO_2$ radical concentrations which is underestimated compared to the measured results. Thus, $R1$ decreases obviously due to the competition of larger proportion of $RO_2+HO_2$, $RO_2+NO$ and $RO_2+RO_2$ reactions in the ROx radical termination budget.

18. Line 313: The authors may want to clarify that 'aerosol properties' may include phase state and that previous measurements have shown lower uptake coefficients for semi-solid and solid particles (e.g. Lakey et al. ACP (2016)). The authors should also clarify somewhere that one of the major limitations of their model is that they assume steady-state concentrations and do not consider concentration gradients which will occur and could change over time for semi-solid particles.

Response:

We changed the word "properties" as "aerosol particle condensed phase component concentrations" in the revised MS in Part 2.2.2 on line 116 at the first paragraph and Part 3.4 on line 314 at the second paragraph to avoid the inaccurate statement of aerosol properties including phase state that is not considered in the MARK model used in this MS.

We added the following statement in the revised MS in section 4 on line 377:

"The novel empirical equation is applicable under the assumption of steady-state concentrations and with metastable or liquid aerosol particles (if the ambient $RH$ over a completely liquid aerosol decreases below the deliquescence $RH$, the aerosol may not crystalize immediately but may constitute a supersaturated aqueous solution (i.e., in the metastable state) (Song et al., 2018)). The approximate calculation of $HO_2$ concentration gradients within the aerosol particle condensed phase also cause deviations for larger particles. The bulk diffusion coefficient of $HO_2$ and other reactive molecules should be lower in the situation of semi-solid particles (Berkemeier et al., 2016;Shiraiwa et al., 2010;Mikhailov et al., 2009) and would change with the water activity and the organic components (Price et al., 2015). This aspect needs further studies. For crystalline or amorphous solid aerosol particles, $HO_2$ will undergo surface reactions and diffuse across the surface rather than be accommodated within the aerosol bulk. The MARK model has limitations in the calculation of $\gamma_{HO_2}$ with semi-solid aerosol particles. In the Wangdu campaign, $\kappa_{sca}$ (optical aerosol hygroscopicity parameter) ranges from 0.05 to 0.35 with an average of 0.22. The ambient $RH$ during the Wangdu campaign shows significant diurnal variations and varies greatly from 15% to 97%, with an average value of 61% (Kuang et al., 2019) indicating that the percentage of solid aerosol particles is relatively low and hence do not significantly influence $\gamma_{HO_2}$.
"

19. The authors fix the solubility of copper at 25%. In reality solubility can vary considerably. How sensitive is this parameter in their model?

Response:

The MARK model is sensitive to Cu ion concentrations that exceed $1\times10^{-4}$ M. We tested the sensitivity of soluble copper ion concentration in the Wangdu campaign between the value of 10% to 70% (Fang et al., 2017;Hsu et al., 2004;Hsu et al., 2010). $\gamma_{HO_2}$ will increase 1.57 times from 0.075±0.031 at 10% solubility to 0.193±0.079 at 70% solubility based on the GaussAmp fitting of data from Wangdu campaign. Even at 70% solubility (which is unlikely true for most situations), the central value of $\gamma_{HO_2}$ is lower than 0.2. The figure below shows the variation of the uptake coefficient with Cu ion solubility.

[Figure]

Figure Response.2 Sensitivity analysis of Cu solubility in the calculation of $\gamma_{HO_2}$ for conditions encountered during the Wangdu campaign.

In the revised MS part 3.4.2 we discussed the influence factors of Cu solubility in the aerosol.

20. Another limitation of the model is that they don't consider reactions between different metal ions (such as Reaction 4 in Mao et al. ACP (2013)) which they have stated. However, could they also speculate how this could impact the estimated uptake coefficients for atmospheric aerosols (e.g. is it expected that this would increase the uptake coefficient)?
Response:
According to our understanding, there is no direct published laboratory measurement evidence of Cu-Fe redox coupling mechanism in HO$_2$ uptake. We speculate that the upper limitation of $\gamma_{HO_2}$ may not change because of the low solubility of Fe and the influence of organic matters. However, whether the product of HO$_2$ uptake is H$_2$O$_2$ or H$_2$O will affect atmospheric oxidation capacity, as outlined by Mao et all (Mao et al., 2013). This should be studied in the future. In the revised MS Section 4, we discussed the influence of other TMI on $\gamma_{HO_2}$.

21. Please check the references carefully as many seem to wrong (e.g. Schwartz and Meyer 1986 line 108 and references in Figure 2).
Response:
We have checked the references in the updated manuscript, and the references in Figure 2 are also checked based on the Short Comment from Professor Heard (please see the first short comment). On line 43 in Section 1, line 79 in Section 2.1, line 142 in Section 2.2.3 and line 263 in Section 3.3, we changed the reference as Schwartz (1984) and Schwartz (1986). Moreover, we removed inaccurate references on line 33, 206 in the original MS and in Table S.3 and part S.1 in the revised MS. We also checked the references for the misuse of capitalization and subscripting.

[revised manuscript text omitted]

---

## Author Comment (AC5) · 21 Jul 2020

Response to the comments of referee #2

We thank the reviewer for their helpful comments. The referee's comments are first given in black type, followed by our response to each in turn in blue type. Any changes to the manuscript in response to the comments are then given in quotation marks in red type and the line number is in the Microsoft-word version of revised MS without revision. The line number may be different in the PDF version, so please see the section number mentioned in the Response.

We have now added D. Moon, M. Baeza-Romero and D. Heard as co-authors to this paper since their unpublished experimental data have been included in this paper and they have contributed to enhance the manuscript.

In summary, the comments made by the referee and our responses to them can be classified into the following categories:

1 The comparison of the results from the empirical equation proposed here with the existing parameterization proposed by IUPAC.

2 Discussion of the influence of the organic content of particles.

3 Correction of different definitions of the same terms and the obscure statements.

4 The corrections of citations in the manuscript.

5 Responses to other specific points.

6 Explanation of part of the SI

Comments are numbered for categorization. For example, 1.1 refers to the first comment in the first category above.

The manuscript of Song et al deals with an important theme in atmospheric science; the interaction of $HO_2$ with particles containing dissolved copper and the modelling of the impact of this heterogeneous reaction on e.g. $O_3$ production.

We thank the reviewer for recognising this.

1.1 Song et al have analysed laboratory data and derived an empirical expression that they then implemented in a model. They suggest that their parameterisation is superior to taking a constant value of 0.2 for the uptake coefficient. This is most likely true but why do they not compare to other parameterisations of this process, e.g. that proposed by IUPAC which also considers laboratory studies up to 2016 and which has a physical basis (i.e. aqueous phase rate coefficients for reaction of $HO_2/O_2^-$ with Cu).

In the original manuscript Figure 2, we compared the novel empirical expression to parameterizations proposed by IUPAC, however we did not include this reference and we simply called it the Resistance Model. In the revised manuscript section 3.2 beginning (line 212), we referred to the sources of the existing parameterization as "(Thornton et al., 2008;Hanson et al., 1992;Hanson et al., 1994;Jacob, 2000;Kolb et al., 1995;Ammann et al., 2013;IUPAC Task Group on Atmospheric Chemical Kinetic Data Evaluation, http://iupac.pole-ether.fr.)".
.

2.1 They also do not consider in any detail the potential for organic content of particles to reduce the uptake coefficient considerably as this is mentioned only in passing on line 312-315. Thus aspect

will be central to improving the modelling of $HO_2$ uptake to particle matter, as has been shown e.g. for $N_2O_5$. The sentence on line 149-150 in this regard is highly misleading: "Without the interference of organic matter, it is convenient to explore the mechanism of $HO_2$ heterogeneous uptake and derive its parameterized equation, which provides a good reference for the heterogeneous uptake of $HO_2$ in the actual atmosphere environment". The co-authors from Jülich will be intimately familiar with this aspect of heterogeneous chemistry and should be able to provide input.

We agree that this sentence is misleading, and whilst recognising that the model has considerable uncertainties, we feel that extending the current IUPAC parameterisation to include the concentration of $Cu^{2+}$ ions (measurements of which in aerosol are available from the field) represents an advance. We have changed the relevant text on line 176 in section 3.1 the first paragraph as follows in the revised MS:

"A simplified approach was used to explore the mechanism of $HO_2$ heterogeneous uptake in order to derive a parameterized equation for the uptake coefficient, $\gamma_{HO_2}$, and which did not consider any potential role of organic matter present in the aerosol because of the reasons detailed below.

Using laboratory measurements of $\gamma_{HO_2}$ for particles containing a single organic component (Lakey et al., 2016;Lakey et al., 2015), it was concluded that the organic content of an aerosol particle may affect several important parameters in the model. For example, the aerosol pH, hygroscopic properties of the aerosol, the rate of diffusion of $HO_2$ within the aerosol and a reduction in the concentration of $Cu^{2+}$ via the formation of complexes that could affect the ability of Cu to undergo redox reactions with $HO_2$ and $O_2^-$. Hence, it is expected that the presence of organic material would change the value of $\gamma_{HO_2}$, but incorporation of terms in the parameterisation of $\gamma_{HO_2}$ from organic material is beyond the scope of this work, and should be a focus of future studies. In fact, during a recent field measurement of the urban atmosphere using a combined laser-flash photolysis and laser-induced fluorescence (LFP-LIF) technique coupled with a versatile aerosol concentration enrichment system (VACES) in Japan, Zhou et al. showed that the average value of $\gamma_{HO_2}$ was $0.24 \pm 0.20$ (1σ) during the study period (Zhou et al., 2020). Although conditions will be different between field locations, this "field measured" value is within the broad range of our model output that does not include organic matter effects."

3.1 The manuscript is illogically organised and difficult to follow, has different definitions of the same term, poor language quality and…

In the revised MS we have tried to improve the language and to be consistent with definitions. Moreover, the revised manuscript is reorganized now according to the following outline:

Abstract
1 Introduction
2 Materials and Methods
   2.1 The Model
   2.2 Corrections on $\gamma_{HO_2}$ in the MARK model
      2.2.1 Henry's law of gas phase reactants
      2.2.2 Aerosol particle condensed phase $Cu^{2+}$ molality calculation
      2.2.3 The conversion formula of $[\overline{HO_2}]$ and $[HO_{2(r)}]$
   2.3 Laboratory results for the $HO_2$ accommodation coefficient
3 Results and Discussion

In order to remove unnecessary details, and to facilitate the flow and aid navigation of the main paper, there is now a supplement with Supplementary Information for the revised MS, which is organised as follows:

S1 Reaction mechanism and reaction rate constants

S2 Calculation of aerosol liquid water content (ALWC) and other important parameters for conditions encountered during the Wangdu campaign

S3 The uncertainty of the calculation for conditions encountered during the Wangdu campaign

We also added the forward and backward reaction rate constants in Table S. 2 in the MARK model calculation.

Please see the SI for more details.

4.1…incorrectly cites the published datasets of Lakey et al (2016) (see author comment from D. Heard). This work might be publishable after major revision and reorganisation.

In response to the Short Comment from Professor D. Heard, and in consultation with the Leeds group, we have modified the way the data are presented in the Table and Figure (there were some mistakes and incorrect citations were used). The citations in the revised MS have been corrected based on the short comment of Professor D. Heard (please see also the response to the first short comment).

Some specific points are listed below:

3.2 L11 hydroxyl peroxy radicals = hydroperoxyl radicals

L12 gamma_HO2 is defined but the expression after line 17 simply lists gamma.

In the new version it has been taken care that only "hydroperoxyl radicals" is used. We changed the wording in the abstract and line 34 in the Introduction.

3.3 L15 parameterisation of $HO_2$? Presumably of $HO_2$ uptake

On line 21 in the Abstract we have added the word "uptake" in the revised MS.

3.4 Does it make sense to list the expression in the abstract, where none of the terms (e.g. ALWC,

[PM], Rd) involved are defined

We changed the abstract in the revised MS including the definition of the main parameters of the parametrization as follows:

"Heterogeneous uptake of hydroperoxyl radicals ($HO_2$) onto aerosols has been proposed to be a significant sink of HOx and hence the atmospheric oxidation capacity. Accurate calculation of the $HO_2$ uptake coefficient $\gamma_{HO_2}$ is key to quantifying the potential impact of this atmospheric process. Laboratory studies show that $\gamma_{HO_2}$ can vary by orders of magnitude due to changes in aerosol properties, especially aerosol soluble copper (Cu) and aerosol liquid water content (ALWC). In this study we present a state-of-the-art model to simulate both gas and aerosol phase chemistry for the uptake of $HO_2$ onto Cu-doped aerosols. Moreover, a novel parameterization of $HO_2$ uptake was developed that considers changes in relative humidity (*RH*) and condensed phase Cu ion concentrations and which is based on a model optimization using previously published laboratory and new laboratory data included in this workdata. The new parameterization is as follows:

$$\frac{1}{\gamma_{HO_2}} = \frac{1}{\alpha} + \frac{3 \times v_{HO_2}}{4 \times 10^6 \times R_d H_{corr} RT \times (5.87 + 3.2 \times \ln(\text{ALWC}/[\text{PM}] + 0.067)) \times [\text{PM}]^{-0.2} \times [Cu^{2+}]^{0.65}}$$

where $\alpha$ is the mass accommodation coefficients which is the probability that a gas-phase molecule colliding with the aerosol surface leads to dissolution, reaction or volatilization, $v_{HO_2}$ is the mean molecular speed of $HO_2$ [cm s$^{-1}$]. $R_d$ is the Count Median Radius of the aerosols [cm], $H_{corr}$ is the Henry's constant [mol cm$^{-3}$ atm$^{-1}$] corrected by solution pH ($H_{corr} = H_0 \times \left(1 + \frac{K_{eq}}{[H^+]}\right)$, where $H_0$ is the physical Henry's law constant), *R* is the gas constant [cm$^3$ atm K$^{-1}$ mol$^{-1}$], *T* is the temperature [K] and [PM] is the mass concentration of particles [μg m$^{-3}$]. According to the new equation, highly variable $HO_2$ uptake coefficients (median value ~0.1) were obtained for the North China Plain and the impact of $HO_2$ uptake on the ROx (=OH + $HO_2$ + $RO_2$) budget was assessed."

3.5 L39-40 Meaning obscure. I think the authors refer to the reduction of aerosol mass over the last few years. The information that the uptake coefficient used (in calculating surface ozone) was 0.2 is superfluous here.

We mean that the role of the reduction of $HO_2$ uptake on aerosol toward that of $O_3$ production is also dependent on the selection of the $HO_2$ uptake coefficient, as well as the reduction of aerosol mass itself. We try to modify the text to become clearer as follows,

"… the reduced $HO_2$ uptake owing to reduction of aerosol surface area is considered to be the key reason for the increased surface ozone concentration over the last few years when a value of 0.2 was used for $\gamma_{HO_2}$."

3.6 L57 L is defined as the aerosol liquid water content. In the abstract it is ALWC.

We have changed all usages of L to ALWC. L has the same meaning as the aerosol liquid water content.

3.7 L68 MARKM model is otherwise referred to as MARK

The name of the model is the "MARK" model, we have been careful now to use "MARK" in the revised MS. We corrected "MARKM" to "MARK" on Section 2.2 header and the first paragraph in the SI Section S1.

3.8 L74 Equilibrium constant have capital "K". Rate coefficients have lower case "k".

We have changed the equilibrium constant on line 108 and Equation (4) as $K_{eq}$, and the rate coefficient as lowercase $k$ in SI Table S. 1 and S. 3.

3.9 L/3 H_0 is estimated (Thornton et al, 2008) to be 3900 M atm-1. Why is this cited in different units to the effective solubility (H^cc). How good is this "estimate" and on what data is it based (I believe Hanson 1992, who also lists a T-dependence)?

We have changed the units of the effective solubility as M atm$^{-1}$ in the new MS. There is no particular reason other than an oversight in writing the manuscritpt that the temperature dependence was not taken into account. Thank you for pointing it out. $H_0$ is the physical Henry's law constant, the original data used in this manuscript is from Golden et al. (1990) and Hanson et al. (1992). We agree it is better to use $H_0$ with the temperature dependent formula recommended by IUPAC as follows:

$$H_0 = 9.5 \times 10^{-6} \exp\left(\frac{5910}{T}\right) M\ atm^{-1}$$

At 298K, $H_0$ equals to 3897.13 M atm$^{-1}$ calculated from the formula, and the estimation as 3900 M atm$^{-1}$ may cause small deviation.

We changed the equation 4 in the revised MS as:

$$H^{cc} = H_0 \times \left(1 + \frac{K_{eq}}{[H^+]}\right) \times A_{HO_2} = 9.5 \times 10^{-6} \exp\left(\frac{5910}{T}\right) \times \left(1 + \frac{K_{eq}}{[H^+]}\right) \times A_{HO_2} \tag{4}$$

In our model, the T-dependence formula is used in the MARK model now. No difference between the original and revised results have been shown because at 298K, with the T dependent parametrization a value of 3897.13 M atm$^{-1}$ is obtained for $H_0$ while a value of 3900 was used before.

3.10 L101 Define [xi]_equ. In the line above only [xi] is mentioned.

In the bulk condensed phase of aerosol particle, the effective concentration $[x_i]_{equ}$, rather than total concentration of ions, should be calculated because of the high ionic strength. We added "effective" to the original sentence under the equation (9) in the revised MS and show the equation of effective $[Cu^{2+}]_{equ}$ in the aerosol particle condensed phase as Equation (11).

5.1 L105 "steady-state" HO$_2$ concentration. Why "steady-state"?

As discussed below (please see response to Comment 1.5), the parameterization proposed by IUPAC is originally from the heterogeneous modeling with liquid droplets and modified by the Resistance Model (Danckwerts, 1951;Schwartz, 1984;Schwartz, 1986;Ammann et al., 2013;Davidovits et al., 2006). The Resistance Model is based on the assumption of steady-state solutions (liquid water cloud droplets). The novel parameterization proposed by Song et al. is still built on the basic framework of the Resistance model, thus only steady-state HO$_2$ concentration can be calculated and in consequence this novel parameterization has the limitation of steady-state assumption.

In the revised MS on line 361 we added statement of the limitation of the novel empirical equation.

3.11 L110 k_eff is listed in the equation 11. This appears to be defined 57 lines later, but not always in the same manner.

We defined $k_{eff}$ as "the comprehensive reaction rate coefficient encompasses both HO$_2$ dissolution equilibrium reactions and liquid phase chemical-physical reactions during HO$_2$ uptake

process." on line 147 just under the Equation (13) in Section 2.2.3 of the revised MS. This is the place where the definition first appeared. We deleted the conflict definitions below.

3.12 Tab1 (and abstract) the accommodation coefficient is not defined, as far as I can see.
We have added the definitions of relevant parameters of the parameterization in the revised MS Abstract. Please see response to comment 3.4.

5.2 Tab1 The data of Lakey 2015/2016 and Zou 2019 are not mentioned (Moon = Lakey ???)
Following consultation with the Leeds group, we added the published data by Lakey et al. 2016 and Zou et al. 2019 in the revised MS Table 1.
The study of Lakey et al, 2015 measured $\gamma_{HO_2}$ on single component organic aerosols and the Cu ion concentration was not high enough ($\sim 0.7$-$1.3 \times 10^{-6}$ to $5.5 \times 10^{-4}$ M) to measure $\alpha$. Thus, we did not include the data from Lakey (2015) in the Table 1. The citations used in the original MS are corrected based on the Short Comment to this paper from Prof. D. Heard and the response to that. Please see the response to the first short comment.

Table 1: $\gamma_{HO_2}$ under lab conditions for $Cu^{2+}$-doped inorganic aerosols.

| Aerosol type | RH/% | Estimation of [Cu] in aerosol/M | $\alpha_{HO_2}$ | Ref. |
|---|---|---|---|---|
| $NH_4HSO_4$ | 75% | 0.0059−0.067* | 0.40±0.21 | (Mozurkewich et al., 1987) |
| $(NH_4)_2SO_4$ | 45% | 0.5 | 0.53±0.13 | (Taketani et al., 2008) |
| $(NH_4)_2SO_4$ | 42% | 0.16 | 0.5±0.1 | (Thornton and Abbatt, 2005) |
| $(NH_4)_2SO_4$ | 53−65% | 0.5−0.7* | 0.4±0.3 | (George et al., 2013) |
| $(NH_4)_2SO_4$ | 65% | 0.57 | 0.26±0.02 | (Lakey et al., 2016) |
| $(NH_4)_2SO_4$ | 51% | 0.0027 | 0.096±0.024 | (Zou et al., 2019) |
| $(NH_4)_2SO_4$ | 43% | 0.38 | 0.355±0.023 | This work |
| NaCl | 53% | ~0.5 | 0.65±0.17 | (Taketani et al., 2008) |
| KCl | 75% | 5% of KCl solution | 0.55±0.19 | (Taketani et al., 2009) |
| $LiNO_3$ | 75% | 0.03−0.0063* | 0.94±0.5 | (Mozurkewich et al., 1987) |

*Cu concentration is in molality (mol kg$^{-1}$).

3.13 Tab2 Add units, do not capitalize K…
Table 2 is moved to the Supplemental Information of the revised MS as Table S.1. In Table S.1 we add the units of reaction rate constants as: "$k_{298}$/M$^{-n}$ s$^{-1}$" in the header.

6.1 …and move to supplementary information
This is a good suggestion. Please see the response to Comment 3.1. We moved this part and Table 3, Table 4 and Table 5 to the SI as Table S. 1, S. 2, S. 3 and S. 4.

3.14 Tab3 K should be capitalized (eqm.)
We changed to a capital $K_{298}$.

3.15 L146 annual average contribution. Does this refer to a global average?
It does not refer to a global average. It is the annual average contribution across China based on the

research of Tao et al., (Tao et al., 2017). We added "across China" in the original statement on line 173 as: "…contribution of inorganic aerosol to $PM_{2.5}$ is between 25% and 48% across China (Tao et al., 2017)…"

5.3 Fig1 Please explain why the uptake coefficient continues to increase at pH > 5 whereas k_eff decreases.

We agree that it was not clearly explained why this is the case. A higher pH will increase the solubility of $HO_2$. Moreover, since the rate of $O_2^-$ with $Cu^{2+}$ is larger than the rate of $HO_2$, with larger pH, $O_2^-$ will be more dominant over $HO_2$ thus increase the reaction rate. $\gamma_{HO_2}$ therefore is higher in alkaline environments. However, the optimization simulation of $k_{eff}$ try to avoid the influence of pH in the range of 3-6. pH influence on $\gamma_{HO_2}$ is embodied in $H_{corr}$ ( = $H_0 \times \left(1 + \frac{K_{eq}}{[H^+]}\right)$. With the fixed value of $\alpha_{HO_2}$ and sharply increasing $H_{corr}$ with pH, the combined reaction rate $k_{eff}$ peaks in the 4-5 pH range, and then quickly declines calculated by Equation (15).

In the revised MS, we deleted the original $k_{eff}$ graph (Figure 1) to avoid confusion in the understanding of the entire reaction system, and we have included the quasi-first order reaction rate constant $k_{het}$ of the gas phase $HO_2$ as in the new Figure 1.

Figure 1 and the explanation are corrected as follows in the revised MS:

[Figure]

Figure 1: Influence of various parameters upon $\gamma_{HO_2}$ predicted by the MARK model. (a) $\gamma_{HO_2}$ increases with the *RH* at different $[Cu^{2+}]$; (b) $\gamma_{HO_2}$ denoted by black squares and black line and $k_{het}$ in red circles and red line increase with aerosol particle condensed phase pH.

We also changed the analysis in the revised MS in Section 3.1 last paragraph as follows:

"$\gamma_{HO_2}$ presents a sigmoid-shaped growth with aerosol particle condensed phase pH. In the model, it is found that as the pH rises, the uptake coefficient rises rapidly because $HO_2$ is a weak acid (pKa = 4.7) and has a low solubility in an acidic environment. The higher condensed phase pH is favorable for the dissolution equilibrium of the gas phase $HO_2$.. This trend is consistent with the observed second-order rate constant of $HO_2/O_2^-$ reviewed by Bielski et al. 1985 (Bielski et al., 1985). Moreover, aqueous phase reaction rates of $HO_2/O_2^-$ and $Cu^{2+}/Cu^+$ increase with the increasing of condensed phase pH because in alkaline environment $HO_2$ is more likely becoming $O_2^-$ which has quicker reaction rate with $Cu^{2+}/Cu^+$. The pH of the ambient atmospheric aerosol is measured

generally below 5 even when the concentration of $NH_3$ is high as in Beijing and Xi'an (Ding et al., 2019;Guo et al., 2017) with a range of 3-5. At this range, $\gamma_{HO_2}$ is highly affected by aerosol condensed phase pH may mainly because of the change of solubility."

3.16 L166 k_eff is defined as a comprehensive reaction rate constant. . .. . .during heterogeneous uptake. On L210, K_eff is defined as the rate of HO2 aqueous reaction with copper ions.
Please see response to Comment 3.11 above.

1.2 L175 The parameterisations listed do not include that of IUPAC, which includes more recent laboratory data than those listed.
Please see the response to Comment 1.1.

1.3 L180 Based on the data of Lakey et al, IUPAC list a rate coefficient for HO2 (O2-) with Cu ions of 5x10^5. This is orders of magnitude lower that those listed.

We added the following explanation in Section 3.2 the last paragraph of the revised MS on line 245: "There is more discussion about this reaction rate. IUPAC (Ammann et al., 2013;IUPAC Task Group on Atmospheric Chemical Kinetic Data Evaluation, http://iupac.pole-ether.fr.) proposed the effective rate coefficient for the reaction of $HO_2$ ($O_2^-$) with Cu ions as $5x10^5$ $M^{-1}$ $s^{-1}$ to achieve the best fit based on the calculation results from Lakey et al. (2016b). This assumption is not in accordance with the aqueous reaction rate coefficient from other databases mentioned below, and needs further laboratory measurements to confirm it. According to the aqueous reaction rate coefficient from NIST and the latest measurement result (Lundström et al., 2004;Huie, 2003), the rate coefficient of $HO_2$ with $Cu^{2+}$ is $1\times10^8$ or $1.2\times10^9$ $M^{-1}$ $s^{-1}$ at pH= 2 and pH=1, respectively. These two rate coefficients were quantified in a low pH environment (pH=2 for $1.2\times10^9$ $M^{-1}$ $s^{-1}$ and pH=1 for $1\times10^8$ $M^{-1}$ $s^{-1}$). At the same time, the reaction rate of $O_2^-$ with $Cu^{2+}$ is $8\times10^9$ $M^{-1}$ $s^{-1}$ for pH in the range 3-6.5 (Huie, 2003). At higher pH, the reaction rate of $HO_2$ with $Cu^{2+}$ may change, but it is unknown whether it will decrease by four orders of magnitude. Further kinetics experiments are needed at varying pH to verify the reaction rate coefficient of $Cu^{2+}$ ions with $HO_2$ and $O_2^-$ in aqueous solution."

4.2 Fig2 Why are the data of Lakey not listed (or are these Moon ??)
Please see the response to Comment 4.1, and the response to the Short Comment by D. Heard.

5.4 Fig2 Taketani also have uptake coefficients in the absence of Cu that are just as high as the single point at about 0.5 M. Why are the other datapoints of Taketani selectively omitted here?
The focus of this manuscript is to investigate the influence of copper ions on $HO_2$ heterogeneous reactions and it proposes a new empirical parameterisation applicable to ambient copper ion containing aerosols. Therefore, we only included the experimental results of Taketani et al. obtained with copper-doped inorganic aerosols and did not include other experimental studies of inorganic aerosol not doped with copper ions, which are inconsistent with other measurements, perhaps owing to differences in experimental conditions in the laboratory (George et al., 2013) as follows. The mechanism of $HO_2$ uptake with single component aerosols (such as $(NH_4)_2SO_4$) is still not fully understood. Moreover, $HO_2$ uptake coefficient measurement is highly affected by experimental

conditions such as $HO_2$ concentration, reaction time, etc. Some data from Taketani et al. are not consistent with other measurements within the community. Taketani reported $\gamma_{HO_2}$= 0.11±0.03 at 45% RH, 0.15±0.03 at 55% RH, 0.17±0.04 at 65% RH and 0.19±0.04 at 75% RH when the $HO_2$ initial concentration was $1\times10^8$ molecule $cm^{-3}$, which are inconsistent with results from Thornton and Abbatt. Thornton and Abbatt concluded that the γ for wet particles of $(NH_4)_2SO_4$ is < 0.01 at ~42% RH and a $HO_2$ ambient concentration of ~$1\times10^8$ molecule $cm^{-3}$ from extrapolation based on their research with $HO_2$ initial concentration at $5\times10^{10}$ molecule $cm^{-3}$. George et al. (2013) reported $\gamma_{HO_2}$ as 0.003±0.005 at 55% RH and 0.01±0.01 at 65-75% RH at $HO_2$ initial concentration of $1.5\times10^8$ -$1.5\times10^9$, also much lower than the measurements of Taketani et al. The initial $HO_2$ concentration and $Cu^{2+}$ contamination will also affect $\gamma_{HO_2}$.

1.4 L188 what is the "current parameterised equation". From whom is it?
Please see the response to Comment 1.1.

1.5 L201 This text ignores the fact that the IUPAC parameterisation accurately reproduces the lab studies from Lakey et al. Is the empirical parameterisation that Song et al propose really superior to the IUPAC one, which has a physical basis?
Please see the response to Comment 1.3 above. We agree with the referee, that the IUPAC parameterization reproduces the laboratory studies from Lakey et al. with $k_{TMI}$ (defined as the second order rate coefficient for the reaction of $HO_2$ /$O_2^-$ with transition metal ions) equal to $5\times10^5$ $M^{-1}$ $s^{-1}$. However, we also note that IUPAC states "the parameterization suggested here is very sensitive to the solubility of $HO_2$ ($H_{HO2}$), its temperature dependence and on the aerosol pH", which we attempt to address in our new parameterisation.
The parameterization proposed by IUPAC is originally from the heterogeneous modeling with liquid droplets which was modified to become the Resistance Model (Danckwerts, 1951;Schwartz, 1984;Schwartz, 1986;Ammann et al., 2013;IUPAC Task Group on Atmospheric Chemical Kinetic Data Evaluation, http://iupac.pole-ether.fr.;Davidovits et al., 2006). The Danckwerts expressions with analytical solutions include the effect of Henry's law solubility on gas uptake, liquid-phase reactions of the solvated molecules, and the mass accommodation coefficient exist for a few limited conditions. In general, the coupled differential equations must be solved numerically. After Danckwerts, Schwartz et al. proposed a parameterization which came to be known as "the Resistance Model". The Resistance Model has been shown to provide a good approximation (within a few percent) to the numerical solution of the coupled differential equations. The whole framework of this parameterized equation was based on the assumption of steady-state solutions (liquid water cloud droplets) and decouple the differential equations for each heterogeneous process while does not take into account the physical and chemical characteristics of the ambient aerosol. The empirical equation proposed by Song et al. has made related improvements including the "*salting out*" effects of gas molecular and effective copper ion concentration. Although still with limitations, the novel empirical equation can be applied to the estimation of $\gamma_{HO_2}$ with aerosol particles.

3.17 L213 where does this expression for the uptake coefficient come from?
The definition of $\gamma_{HO_2}$ is from the Appendix A of Hanson et al., 1994 (Hanson et al., 1994). We add the citation on line 270 in Section 3.3 second paragraph as: "Hanson et al. (1994) proposed the definition of the uptake coefficient as $\gamma = \alpha(1 - \frac{c_{a,surf}}{H^{cc}c_{g,surf}})$ where $c_{a,surf}$ is the suface

concentration of the reactant, $c_{g,surf}$ is the gas phase concentration. In the process of HO$_2$ uptake, we deduce the parameterized equation of $\gamma_{HO_2}$ in the framework of the resistance model."

6.2 L265 section 3.4.2. could be move to SI
This is a good suggestion. Please see the response to Comment 3.1.
We move section 3.4.2. to the SI.

5.5 L312 The work of Lakey et al is cited. Where are their data?
Please see the response to Comment 5.2 and the response to the Short Comment by D. Heard.

2.2 How much "lower" can the uptake coefficient be in the presence of organics?
Please see the response to Comment 2.1.

6.3 L351 move section 3.4.5 to SI
This is a good suggestion. Please see the response to Comment 3.1.
We move section 3.4.5 to the SI.

Ammann, M., Cox, R. A., Crowley, J. N., Jenkin, M. E., Mellouki, A., Rossi, M. J., Troe, J., and Wallington, T. J.: Evaluated kinetic and photochemical data for atmospheric chemistry: Volume VI - heterogeneous reactions with liquid substrates, Atmos. Chem. Phys., 13, 8045-8228, 10.5194/acp-13-8045-2013, 2013;IUPAC Task Group on Atmospheric Chemical Kinetic Data Evaluation, http://iupac.pole-ether.fr.

Bielski, B. H., Cabelli, D. E., Arudi, R. L., and Ross, A. B.: Reactivity of HO$_2$/O$_2^-$ radicals in aqueous solution., Journal of physical and chemical reference data, 14, 1041-1100, 1985.

Danckwerts, P.: Absorption by simultaneous diffusion and chemical reaction into particles of various shapes and into falling drops, Transactions of the faraday society, 47, 1014-1023, 1951.

Davidovits, P., Kolb, C. E., Williams, L. R., Jayne, J. T., and Worsnop, D. R.: Mass accommodation and chemical reactions at gas-liquid interfaces, Chemical Reviews, 106, 1323-1354, 10.1021/cr040366k, 2006.

[revised manuscript text omitted]

---

## Author Response (AR2)

**Response to the comments of reviewer #3:**

Comment:

    This paper describes a new parameterization for the reaction probability of HO2 on Cu-containing aerosol particles. The reactive uptake of HO2 to aerosol remains an uncertain but potentially important component of atmospheric HOx fate, and has gained renewed attention of late. There is thus a reasonable need for more evaluation of how HO2 reactive uptake is described in models and this paper therefore has scientific merit and is appropriate in scope for ACP.

    We thank the reviewer for the helpful comments. The referee's comments are first given in black type, followed by our response to each in turn in blue type. Any changes to the manuscript in response to the comments are then given in quotation marks in red type. And in order to avoid the confusion between the two parameterized equations mentioned in the MS, the classical parameterization equation (proposed by IUPAC) is abbreviated as *CEq.*, and the parameterization proposed in this paper is abbreviated as *NEq.*. We have added these two abbreviations to make the language more concise in the revised MS.

Comment:

However, I have several concerns about the scientific quality and presentation in the manuscript that I think should be addressed before the paper is published. I describe these concerns below.

1) Uncertainty - overview. There is little objective discussions of measurement uncertainty and about uncertainties regarding application of the parameterization.

    For the new experimental dataset included in the paper, the final uncertainty in the uptake coefficient is calculated via the propagation of errors. The uncertainty of the averaged $HO_2$ signal is calculated as $1\sigma$ of the average of the data points recorded during the averaging period (20 s). The pseudo first-order loss rate coefficient (k') was obtained from the gradient of a plot of $\ln(HO_2$ signal) against reaction time, the error for k' was $1\sigma$ of the uncertainty of the gradient and was obtained from the plotting software, Origin Pro (v.9.0). The uptake coefficient ($\gamma(HO_2)$) was obtained from the gradient of the plot of k' against the surface area concentration of aerosols in the flow tube. The error given on all measurements of $\gamma(HO_2)$ represents $2\sigma$ of the uncertainty of the gradient also calculated in Origin Pro. For all calculations, errors were propagated using the standard laws of error propagation. The Brown correction calculation, used to correct k' for non-plug flow conditions in the flow tube, is complex and is performed by a FORTRAN subroutine called ROOT. When propagating error through Brown correction calculations the correction factor was determined by plotting the Brown corrected pseudo-first-order rate constant, k´, against the uncorrected observed rate constant minus the rate constant observed during the wall loss run resulting in a linear plot. The gradient of such a plot was the correction factor calculated by ROOT and was also applied to the uncertainty of k'. See further discussion about uncertainties in the new experimental dataset in the response to the next referee´s comment.

This is only for the Leeds data, and does not necessarily apply to other experimental data from other groups. Although the
35 MARK model simulation results in this paper are not obtained by adjusting parameters to fit the experimental data points, laboratory measurement uncertainties will directly influence the evaluation of the deviation between the modelled HO$_2$ uptake coefficient and the measured results because all the parameters inputted in the MARK model are in reference to the measurement conditions. However, it is difficult to calculate the detailed uncertainties from all factors that influence $\gamma_{HO_2}$ because of the nonlinear reaction system. What is more, uncertainties of the experimental conditions such as RH and particle
40 diameters are combined into the reported values of $\gamma_{HO_2}$. Taking all these into account, we calculated an averaged uncertainty for the experimental values of $\gamma_{HO_2}$ in different ranges of Cu ions concentration.

1.1 There is apparently a new data set on measured gamma_HO2, but there is *very little* discussion of those measurements, how they were made, how the uncertainty was determined etc. The relative uncertainty for the new data at high Cu is nearly
45 an order of magnitude lower than that reported for other measurements in the literature.

We agree with the referee that there is little discussion on the new measurements, the reason being that additional authors for these new measurements were added to the paper at the revised stage, and the original paper had no description of the experimental method. We have now included a brief description of the experimental set up used for these measurements.

With regards to the new measurements reported in this paper, the reason for the decrease in measurement uncertainty compared with our previous measurements, was a series of improvements developed in the experimental methodology that allow a better signal to noise ratio and more reproducible results to be obtained. For example, the sensitivity towards HO$_2$, quantified by calibration, was typically an order of magnitude better for these experiments, resulting in significantly smaller
55 measurement error bars at each point along the flow tube. The improved uncertainty on each data point resulted in a higher quality decay of the HO$_2$ signal down the flow tube and a better linear fit of the plot of the natural log of the HO$_2$ FAGE signal versus contact time, resulting in a lower degree of uncertainty on the fit and hence the derived pseudo first order rate constant, k', from it. Methodology included increasing the degree of averaging, allowing longer times for the stepper-motor to stabilize the sliding injector for each of its positions in the flow tube, more frequent cleaning of the atomizer, cleaning the set up with
60 deionized water between different experiments, the use of longer times for the aerosol concentration to stabilize between experiments. In addition, the aerosol flow tube was cleaned on a weekly basis with distilled water and a new coating of halocarbon wax applied for each new aerosol system investigated. It was observed that this in-depth cleaning and longer stabilization times between experiments (e.g. performed with different Cu concentrations) led to highly reproducible results which translated into significantly reduced uncertainties in the uptake coefficients. All these factors promote a reduction in the
65 uncertainties of our measurements.

The reason for the large uncertainties for Thornton et al. and Taketani et al. at the highest Cu concentrations is not clear. For the Thornton et al. dataset this could be related to the use of a completely different set up for the detection of $HO_2$ radicals which is less sensitive than FAGE. For the Taketani et al. dataset who used a very similar FAGE set up, it could be related to some of the factors listed above related to the experimental procedure. Moreover, for the Taketani et al. dataset, the quoted errors are given for two standard deviations from the least-squares fits (95% confidence limits), which is then combined with the estimated systematic uncertainties in the measurements of aerosol surface concentration (5%) and flow speed (2%). Our quoted uncertainties did not include any systematic uncertainties in our measurements as they are small compared to the random, statistical error.

In the revised MS, we have added the following text related to the experimental set up in Section 2.3 in the revised MS:

"The experimental setup and methodology used to make the new measurements of $\gamma(HO_2)$ reported here have been described in detail elsewhere (Moon et al., 2018b; Lakey et al., 2016c; George et al., 2013) and so only brief details are given here. In summary, the experiments were performed by moving an $HO_2$ injector backwards and forwards along the concentric axis of a laminar aerosol flow tube hence changing the contact time between $HO_2$ and the aerosols. Measurements of $[HO_2]$ were performed using laser induced fluorescence (LIF) spectroscopy at low-pressure (the fluorescence assay by gas expansion (FAGE) technique (Heard and Pilling, 2003)) and the total aerosol surface area was determined with a Size Mobility Particle Sizer (SMPS) at the end of the flow tube. Aerosols were formed using a constant output atomiser (TSI, 3076) and the aerosol concentration and hence surface area could be varied, being controlled using a high efficiency particulate air (HEPA) filter in a bypass arrangement. Atomiser solutions were prepared by dissolving 0.01 moles of ammonium sulphate (AS) (Fisher scientific, >99%) with varying amounts of copper (II) sulphate (Fisher scientific, >98%) in 500 mL of Milli-Q water. The data were analysed as described in George et al 2013. The pseudo first-order loss rate coefficient ($k'$) was obtained from the gradient of a plot of $\ln(HO_2$ signal) against the interaction time between $HO_2$ and the aerosol before sampling by the FAGE detector. The uptake coefficient ($\gamma(HO_2)$) was obtained from the linear least-squares gradient of the plot of $k'$ against the surface area concentration of aerosols in the flow tube. The error given on all measurements of $\gamma(HO_2)$ represents $2\sigma$ of the uncertainty of the fitted gradient. A correction to $k'$ was applied to taking into account non-plug flow conditions in the flow tube using the Brown method."

Comment:

Given that the parameterization is developed by comparing a mechanistic model to multiple data sets from different experiments - how each data point is weighted and the role of the relative uncertainty needs to be discussed. Best would be to show some portion of the raw data from the new measurements and step through how the uncertainty propagates to the reported gamma_HO2.

100

We changed the Section 3.2 in the MS and discuss the uncertainties influence in the revised MS:

"What is more, laboratory measurement uncertainties will directly influence the evaluation of the deviation between the modelled $HO_2$ uptake coefficient and the measured results because all the parameters inputted in the MARK model are in reference to the measurement conditions. However, it is difficult to calculate the detailed uncertainties from all factors that influence $\gamma_{HO_2}$ because of the nonlinear reaction system. Uncertainties of the experimental conditions such as RH and particle

105 diameters are combined into the reported values of $\gamma_{HO_2}$. Taking all these into account, we calculated an averaged uncertainty for the experimental values of $\gamma_{HO_2}$ in different ranges of Cu ions concentration. Laboratory measurement uncertainty has the largest value of 35.1% in the range of $1\times10^{-4}$ to 0.01 M soluble copper concentration, 14.9% below $1\times10^{-4}$ M and 9.3% higher than 0.01 M."

110

We used the weighted average values of each datapoint uncertainty when calculating RMSE as the following equation. RMSE is now changed to 0.13 considering the relative uncertainty of measured data.

We add the following statement in section 3.2 in the revised MS:

"In this paper, the relative error of each measured data point is considered to calculate the weighted average in RMSE:

115
$$RMSE = \sqrt{\frac{\sum_{i=1}^{n}\left(\left(log_{10}^{u_{i_{measured}}} - log_{10}^{u_{i_{model}}}\right)^2 (\omega_i)^2\right)}{\sum_{i=1}^{n}(\omega_i)^2 \cdot n}}$$

$u_{i_{model}}$ is the MARK model result at each $Cu^{2+}$ concentration and $RH$, $u_{i_{measured}}$ is the central value of each measurement result and $\omega_i$ is its corresponding relative error."

120 Comment:

1.2 It seems to me that little can be concluded about the effects of RH given the spread of measurement values at the same RH. While the model has a RH dependence built in through activity coefficients and Cu concentrations, etc, there is no real comparison of the modeled predicted RH dependence with the measurements where a plot of gamma_HO2 for a given Cu mass fraction would be shown versus RH.

125

The MARK model simulation results in this paper are not obtained by adjusting parameters to fit the experimental data points. At present, there are experimental measurements of $\gamma_{HO_2}$ at different RH but there is no experimental systematic study of this dependence where only RH is changed and no other parameters. Many researches proposed that $\gamma_{HO_2}$ is higher for aqueous inorganic aerosol than for dry inorganic aerosol. Although the previous experiments did not directly measure the RH´s

130 dependence, the change of the experimental uptake coefficients met the simulation trend (see Figure 2). Ambient RH would

affect the activity coefficients of reactant ions in the aerosol particle condensed phase and the solubility of the gas phase reactant such as OH, HO$_2$ and H$_2$O$_2$. The MARK model was presented to simulate the change of the uptake coefficient with RH in Figure 1. Moreover, we also did simulations in the MARK model of HO$_2$ uptake with dilute solution droplets with no consideration of RH and update the Figure 2 in the revised MS.

[Figure]

135

Figure 2: Dependence of $\gamma_{HO_2}$ on aerosol copper concentration. Red filled circles denote the results at 43% *RH* measured at Leeds included in this paper. Blue hollow circles at 65% *RH* (Lakey et al., 2016c). Yellow filled diamonds denote results at 51% *RH* (Zou et al., 2019), filled purple triangle at 42% *RH* (Thornton and Abbatt, 2005a) and filled green star at 45% *RH* (Taketani et al., 2008). The grey dashed line denotes the results of the existing parameterized equation (named as *CEq.* in this

140  paper) $\gamma_{HO_2}$ with dilute solution droplets (Thornton et al., 2008; Hanson et al., 1992; Hanson et al., 1994; Jacob, 2000; Kolb et al., 1995), which was confirmed by researches of reactive gas molecular uptake on dilute solution droplets (Hu et al., 1995; Magi et al., 1997) and on aqueous surfaces (Utter et al., 1992; Müller and Heal, 2002). The solid grey lines represent the model results of the MARK model in this study at various *RH* (two lines represent the range of *RH* from 64% to 66%, 50% to 52% and 42% to 44%) and the short dotted line represents the result in the MARK model of HO$_2$ with dilute solution droplets. The

145  root mean square error (RMSE) between the MARK modelled values and the full dataset is 0.13. The aerosol pH is set as 4.5 based on the aqion 7.0.8 interface considering the participation of Cu ion (for details please see https://www.aqion.de/).

Comment:

1.3 Application of the parameterization (more about the parameterization below): There is a brief discussion of "soluble Cu"

150  in ambient aerosol - but clearly this is the biggest source of uncertainty in using the parameterization. The author's choice of 20% of total Cu being soluble seems rather arbitrary. Really, this issue has been discussed already many years ago - there may

well be enough Cu measured in ambient aerosol but whether it is in a soluble form (and also in a "free ion") form remains poorly understood. This uncertainty is compounded by the affect of organic aerosol and its morphology relative to an aqueous phase, the potential for externally mixed aerosol in an urban atmosphere (i.e. the Cu is contained in only a fraction of the total surface area), etc. The authors could show the impact of the various assumptions on the predicted gamma more clearly. Were the Cu measurements size resolved? If so, how does the Cu mass distribution compare to the total aerosol surface area distribution?

Soluble copper concentration is a large source of uncertainty in using the novel parameterization (*NEq.*). There is no measurement of soluble copper concentration in aerosol in the Wangdu campaign. According to the research of Fang et al. (2017), the solubility of Cu across all size ranges of urban aerosol is related to pH. At aerosol pH from 2 to 6, the solubility is about 20%-30%. Based on the previous work mentioned in the MS section 3.4.2, we assume 25% rather than 20% (mentioned as the comments above) of total aerosol metal copper concentration is soluble in the accumulation mode when calculating $\gamma_{HO_2}$ in Wangdu campaign.

We added the statement in Section 3.5.2 last paragraph in the revised MS: "$\gamma_{HO_2}$ rather depends on copper concentration so we also evaluate the influence of copper solubility and mixing state of copper in aerosols on the uptake coefficient (details in the SI)."

We test the relation of $\gamma_{HO_2}$ to copper solubility from 10% to 70% already in the Response to comments at Interactive Discussion. We now added the discussion in the revised SI section 3. We also added the discussion of the mixing state of copper influence on $\gamma_{HO_2}$ in this part:

"S3 Discussion of the $\gamma_{HO_2}$ uncertainty when using the *NEq.* in the Wangdu campaign.

S3.2 The uncertainty from the effective copper concentration

We tested the sensitivity of PM$_{2.5}$ soluble copper ion concentration in the Wangdu campaign between the value of 10% to 70% (Fang et al., 2017; Hsu et al., 2004; Hsu et al., 2010a). $\gamma_{HO_2}$ will increase from 0.065±0.051 at 10% solubility to 0.196±0.142 at 70% solubility for the summary of day and night data based on the Gaussian fitting. The calculation is under the assumption that aerosols are completely *internally mixed*.

The influence of externally mixed aerosol copper on $\gamma_{HO_2}$ is illustrated below. Since there is no data of copper mixed state in the Wangdu campaign, we assumed a 12 bins distribution of copper concentration to evaluate the influence of Cu mixed state on HO$_2$ uptake process. The average concentration of Cu is the same as the *internally mixed* one, in which case, 25% copper is soluble in the aerosol particle condensed phase. Cu distributes in the aerosol particles at different concentrations, and the uniformity of the distribution is measured by the Standard Deviation (SD) of its concentration ratio to the averaged Cu concentration. Higher SD means more uneven distribution of Cu in the particles.

This calculation is only valid for particles smaller than 2.5μm (which is the most important size bins for $HO_2$ uptake), and Cu size distribution in aerosol particles is not considered here. Four modes of external mixtures states were tested as shown in the Table S.6.

Table S.6 Four modes of external mixture state of unbar aerosol copper and corresponding Gaussian fitted $\gamma_{HO_2}$.

| Gaussian fitted $\gamma_{HO_2}$ (1σ) | Square Deviation (SD) of the copper distribution |
| --- | --- |
| 0.110±0.079 | 0 |
| 0.105±0.073 | 0.35 |
| 0.089±0.065 | 1.18 |
| 0.079±0.056 | 1.71 |
| 0.051±0.033 | 6.24 |

With the increase of the Square Deviation of copper distribution in aerosol, the uptake coefficient becomes smaller and more centralized. Aerosol particles morphology relative to an aqueous phase will influence the uptake coefficient of $HO_2$. The uptake process would vary with mixing state and size distribution of the particles, thus the predicted $\gamma_{HO_2}$ values here may be biased as a result, but represents an average over bulk aerosols. The estimation value of $\gamma_{HO_2}$ under the assumption that $HO_2$ reacting with completely *internally mixed* aerosol in the Wangdu campaign is the upper limit value. The uneven distribution of copper in aerosol particles would lead to a further decrease in the $HO_2$ uptake coefficient. Another source of uncertainty comes from the lack of information about the copper size distribution in Wangdu campaign. This aspect needs further studies.

[Figure]

Figure S. 4 The statistical relative frequency distribution of averaged $\gamma_{HO_2}$ in different modes of copper mix state."

Comment:

What is the organic aerosol mass to inorganic mass ratio? If there was a core-shell morphology and the maximum gamma was similar to the impact of organics on HO2 uptake reported by Lakey et al, what would the current parameterization predict?

We added the following discussion in the revised SI:

"S3 Discussion of the $\gamma_{HO_2}$ uncertainty when using the *NEq.* in the Wangdu campaign.

S 3.3 The uncertainty from the core-shell morphology of aerosol particles

The presence of organic material would change the value of $\gamma_{HO_2}$. We revised the *NEq.* based on the research of Anttila et al. (2006) who treated the organic fraction in the aerosols as a coating, as given below:

$$\gamma_{org\_coat} = \frac{4RTH_{org}D_{org}\varepsilon}{v_{HO_2}l}$$

$$\frac{1}{\gamma_{HO_2\_corr}} = \frac{1}{\gamma_{HO_2\_in}} + \frac{1}{\gamma_{org\_coat}}$$

Here, the $H_{org}$ is the Henry's law constant of HO$_2$ for organic coating. $D_{org}$ is the solubility and diffusivity of HO$_2$ in the organic coating, the value is corrected by Lakey et al. (2016b) using the Stokes–Einstein equation resulting a factor of 1.22 decrease in the diffusion coefficients of HO$_2$ compared to the diffusion coefficients of H$_2$O on the sucrose aerosol particles. $\varepsilon$

215  is the ratio of the radius of the aqueous core ($R_c$) and the particle ($R_d$). The particle radius $R_d$ was the measured Count Median Radius of the aerosols [cm]. $l$ is the coating thickness [cm] of the organic matters which is calculated from the volume ratio of the inorganics to total particle volume with the assumption of a hydrophobic organic coating (density, 1.27 g cm$^{-3}$) on the aqueous inorganic core (with a density of 1.77 g cm$^{-3}$).

$\gamma_{HO_2\_in}$ and $\gamma_{HO_2\_corr}$ are the uptake coefficients calculated by the *NEq.* and the corrected value under the assumption of organic

220  coating, respectively. We tested the influence of the OM ratios in the range of 20%-70% for HO$_2$ uptake onto PM$_{2.5}$ due to the lack of direct measurement data in the Wangdu campaign.

[Figure]

Figure S.5 The ratio of $\gamma_{HO_2\_corr}$ and $\gamma_{HO_2\_in}$ as a function of the relative coating thickness in the Wangdu campaign.

225  The ratio of $\gamma_{HO_2\_corr}$ and $\gamma_{HO_2\_in}$ decreasing with the ratio of OM denotes the influence of particle core-shell morphology on HO$_2$ mass transfer process in aqueous organic solvent. Although the diffusion coefficient changes by more than 3 orders (3-7 orders) of magnitude over the investigated range of relative humidity, modeled averaged mean relative difference of HO$_2$ uptake coefficients change by only 3 times when the $l/R_d$ changes by an order of magnitude. One possible reason for this it is that the uptake coefficient being proportional to the square root of the diffusion coefficient when the uptake is controlled by

230  reaction and diffusion of HO$_2$ in the bulk (Davidovits et al., 2006; Berkemeier et al., 2013; Lakey et al., 2016a). OM (organic matter) usually accounts for 20–50% mass of PM$_{2.5}$ in Beijing and other urban areas (Wang et al., 2017; Sun et al., 2012). Thus, we proposed a possible range of HO$_2$ uptake coefficient in the Wangdu campaign as 0.62-0.74 times lower than values without correction for organic matter."

235  Comment:

Aerosol pH is likely rather uncertain, the authors should discuss a reasonable estimate of uncertainty in the ambient aerosol pH and its size dependence and how this uncertainty would impact the predicted HO2 gamma.

Aerosol pH is an important uncertainty in the implication of the *NEq.*. Aerosol pH size distribution is not measured in the Wangdu campaign; thus we used the estimated PM$_{2.5}$ pH using a thermodynamic equilibrium model as aerosol acidity.

240    ISORROPIA II, was the model used to predict aerosol pH. The averaged aerosol pH is 3.41 ± 0.69 (1σ). The averaged diurnal profiles of aerosol pH are given by Liu et al. (2017). The relative error of measured NH$_4^+$-Na$^+$-Cl$^+$-K$^+$-HNO$_3$-NH$_3$-HCl inputted in the ISORROPIA II model is 10%. The uncertainty caused by aerosol pH is already considered in the estimation of $\gamma_{HO_2}$ in the revised version. We add the following statements in section 3.5.3 in the revised MS:

"…The aerosol pH is calculated using the thermodynamic model ISORROPIA-II (Fountoukis and Nenes, 2007) and the

245    averaged value is 3. 41 ± 0.69 (1σ)."

And in the SI section S3:

"…and aerosol liquid water content (±9.1%). Measured aerosol NH$_4^+$-Na$^+$-Cl$^+$-K$^+$-HNO$_3$-NH$_3$-HCl concentrations cause 10% uncertainty."

250    Comment:

2) Parameterization development. The discussion of how the parameterization was developed is awkward and could at least use a different organization and in some cases more precise language.

After the last major revision, the manuscript does have local structural confusion and language coherence problems in origin

255    section 3.2. We now rewrite this part and added a new section (section 3.3) to discuss the different applications of the classical equation and the MARK model in the revised MS:

[revised manuscript text omitted]

   "

295

The authors should focus on the conditions and parameters which have the biggest effect first. For example, the authors do not mention the rate coefficients used to determine "keff" until the results section! In fact, the entire paragraph on the rate constants needs to be rewritten - I couldn't understand it. The authors open the paragraph saying: "The deviation of $\gamma HO2$ between the MARK model and laboratory studies is smaller than the predicted results from the existing parameterized equation (Thornton

300 et al., 2008; Hanson et al., 1992; Hanson et al., 1994; Jacob, 2000; Kolb et al., 1995; Ammann et al., 2013; IUPAC Task Group on Atmospheric Chemical Kinetic Data Evaluation, http://iupac.pole-ether.fr.) as shown in Figure 2. " But, it turns out to be closer to the measurements because the authors appear to have chosen a much smaller reaction rate coefficient? Please make that clear - what is the default prediction of the MARK model without any changes to achieve better agreement?? That

prediction should be shown on the figure somehow. Then the discussion of what the rate constants are in the literature and what the recommended values are should be discussed clearly - please distinguish HO2(total) from HO2(aq) and O2-(aq). I assume the dissolved forms are treated separately in the model. Basically - it seems the authors are arguing the rate coefficient for various forms of HO2 in a concentrated aqueous solution are different from previous values, and they should present that clearly and why they think those rate constants are actually two orders of magnitude slower and not that the measurements of gamma-HO2 are biased low or low for some other reason (e.g solubility or diffusion limitation, etc). The authors then state at the end: "The main reason for the differences between the original parameterization and the MARK model is the effect of including the activity coefficients of Cu ions and HO2 and the effects of reactions of different valence states of copper ions." This doesn't seem right if you changed the rate constant by two orders of magnitude - I would say that is the main difference between the original parameterization - please demonstrate that the change in this rate constant is not what affects the difference between the MARK and original parameterization (which I assume refers to the dashed line on figure 2).

We change the relative statement in the revised MS to make it clearer that we did not choose a smaller reaction rate coefficient in the MARK model calculations. All the reactions used in the MARK model is presented in the supplementary materials Section 1 Reaction mechanism and reaction rate constants. The reaction rate of $Cu^{2+}$ with $HO_2/O_2^-$ is $1\times10^8$ $M^{-1}$ $s^{-1}$ and $8\times10^9$ $M^{-1}$ $s^{-1}$. All input parameters are the same except that for the MARK model involved more liquid phase reactions instead of only considering the second order rate coefficient ($k_{TMI}$) for the reaction of $HO_2$ and $O_2^-$ with transition metal ions as the *CEq.* did. Based on the research of Bielski in 1985 (Bielski et al., 1985), we reduced the input rate constant to $1.5\times10^7$ $M^{-1}$ $s^{-1}$ rather than the more commonly used value of $1\times10^9$ $M^{-1}$ $s^{-1}$ (relative discussions are in the MS). This treatment within the calculation can bring predictions more in line with experimental results in the *CEq.*. The *CEq.* can provide good estimation of reactive gas molecular uptake coefficient on dilute solution droplets (Hu et al., 1995; Magi et al., 1997) and on aqueous surfaces (Utter et al., 1992; Müller and Heal, 2002), while may overestimate the uptake coefficients with ambient aerosol (please see the previous response).

Comment:
The authors compare the model predictions of gamma HO2 to the data measured on ammonium sulfate aerosol doped with Cu. What was the assumed pH for the model predictions of the laboratory data? What is the basis of that prediction and could the pH vary between the different laboratory experiments if it wasn't explicitly measured? A flow tube that isn't regularly cleaned could develop and NH3 background which makes the pH possibly higher than estimated for ammonium sulfate in the absence of excess ammonia.

The main components of the aerosols used in the laboratory measurements of $\gamma_{HO_2}$ shown in Figure 2 are ammonium sulfate with a small amount of copper sulfate. According to the calculation based on the aqion 7.0.8 interface (for details please see https://www.aqion.de/), the pH is around 4.54 considering aerosol dehydration with 2M aerosol bulk sulfate concentration and

1M Cu ion at 25°C. In the MARK model, we set aerosol pH as 4.5 when calculating $\gamma_{HO_2}$. We also input pH as 4.5 in the *CEq.*. The MARK model and the *CEq.* results have no uncertainty from pH in the calculations. The accurate measurement or calculation of aerosol pH is an important factor in the implication of the novel parameterization (*NEq.*) proposed in this paper as described above.

In the experimental measurements included in this paper a careful procedure for cleaning the aerosol flow tube was used as mentioned above that could also avoid a NH₃ background.

Comment:

The authors discuss that they assumed only HO2 was reacting with Cu in these experiments (at least that is what I discerned - but the language in this section is very unclear as it mentions pH of ambient aerosol being typically 3-6 and so they used the HO2 + Cu rate constant which is two orders of magnitude smaller than O2- + Cu).

The *CEq.* is assuming that only $HO_2/O_2^-$ are reacting with $Cu^{2+}$ ions rather than the MARK model. The MARK model is a box model with lots of other reactions described in detail in the SI Section 1. The MARK model considered other factors such as the activity coefficient and effective Henry's constant. A low reaction rate constant is used in the calculation of the *CEq.* but not in MARK model. Please see response above.

Comment:

The authors mention that the "model selects a mass accommodation coefficient of 0.5" - but this is an input to the model, correct? Is the model prediction iterated for each measurement or how is it that the model selects? The mass accommodation coefficient alone is another source of significant uncertainty - the authors could better illustrate the true uncertainty of the model predictions based on mass accommodation and pH assumptions.

The HO₂ mass accommodation coefficient ($\alpha_{HO_2}$) is chosen as 0.5 based on the previous work mentioned in Section 2.4. This is an input value to the MARK model as well as in the *CEq.* calculations of $\gamma_{HO_2}$ with copper-doped ammonium sulfate aerosol. An important reason for choosing 0.5 as the fixed value is that for the copper-doped ammonium sulphate aerosol, when the concentration of copper ion is high enough, the heterogeneous uptake process of HO₂ is mainly limited by mass transfer rate rather than liquid phase chemical reaction rate, which shows the effect of $\alpha_{HO_2}$. On the other hand, according to the resistance model theory $\alpha_{HO_2}$ does not influence the liquid phase reactants' reaction rates. In the process of optimizing the comprehensive liquid phase reaction rate coefficient $k_{eff}$ the influence of $\alpha_{HO_2}$ on the uptake coefficient can be embodied in the term of $1/\alpha_{HO_2}$, thus there is no correlation between $k_{eff}$ and $\alpha_{HO_2}$.

The uncertainty of $\alpha_{HO_2}$ mainly manifests in the implication of the *NEq.*. We modified the original text and set three gradients of $\alpha_{HO_2}$ to simulate the uptake coefficient: 0.2, 0.5 and 0.8. We added the following part in the revised SI:

370 "S3 Discussion of the $\gamma_{HO_2}$ uncertainty when using the *NEq.* in the Wangdu campaign.

S3.1 The uncertainty from the HO$_2$ mass accommodation coefficient ($\alpha_{HO_2}$).

The HO$_2$ mass accommodation coefficient ($\alpha_{HO_2}$) is influenced by many factors including the aerosol organic component, particle size distribution, RH and temperature, etc. There is no direct measurement result of $\alpha_{HO_2}$ in the Wangdu campaign or any other field campaign currently due to experimental difficulties. $\alpha_{HO_2}$ is a source of significant uncertainty when using the

375 *NEq.* to estimate $\gamma_{HO_2}$. Here we set five gradients of $\alpha_{HO_2}$ to simulate the mean $\gamma_{HO_2}$ and the results of the fit to a Gaussian function result in $\gamma_{HO_2}$ median values of 0.088 ± 0.022 (1σ) at $\alpha_{HO_2}$ = 0.2 and 0.125± 0.041 (1σ) at $\alpha_{HO_2}$ = 0.8.

[Figure]

Figure S.3 Gaussian fitting results of $\gamma_{HO_2}$ under different $\alpha_{HO_2}$ in the Wangdu campaign, estimated by the *NEq.*."

380

Comment:

3) Presentation - as noted in the above comments, more clarity is needed in what assumptions are made, under what conditions, and what the impact of those assumptions is on the main conclusions.

I find little need to show the distribution of gamma values for day and night - the two distributions look essentially the same.

385 The only different is that HO2 reactive uptake becomes a larger fraction of the total HOx loss (because RO2 + NO goes towards zero). This seems a minor distinction to make given the bigger uncertainties in the applicaiton of the parameterization as discussed above that would be better to communicate to the broader community that might want to employ the parameterization.

We deleted the panel (a) and (b) in Figure 4 now and give the Gaussian fitted value of HO$_2$ uptake coefficient for the summary of day and night data.

390 A new section "3.5.4 Discussion of uncertainties of $\gamma_{HO_2}$ estimated at Wangdu field campaign" in the revised MS has been included that discussed the uncertainties of the *NEq.* proposed by this paper.

The value estimated by the *NEq.* represents the upper limitation of $\gamma_{HO_2}$ in the Wangdu field campaign considering the large uncertainties mainly from the aerosol properties. More work needs to be done to evaluate the true value of $\gamma_{HO_2}$. The *NEq.*

395 proposed by this paper provides a novel way for more detailed calculation of the effects of $HO_2$ heterogeneous reactions on the atmospheric radical budget, ozone production and particulate matter generation.

[revised manuscript text omitted]

*Cu concentration is in molality (M).

**2.4 The experimental setup and  methodology of the latest results of $\gamma_{HO_2}$**

In this study, we also conclude the latest results which measured at Leeds. The experimental setup and methodology used to make the new measurements of $\gamma(HO_2)$ reported here have been described in detail elsewhere (Moon et al., 2018b; Lakey et al., 2016c; George et al., 2013) and so only brief details are given here. In summary, the experiments were performed by

moving an $HO_2$ injector backwards and forwards along the concentric axis of a laminar aerosol flow tube hence changing the contact time between $HO_2$ and the aerosols. Measurements of $[HO_2]$ were performed using laser induced fluorescence (LIF) spectroscopy at low-pressure (the fluorescence assay by gas expansion (FAGE) technique (Heard and Pilling, 2003)) and the total aerosol surface area was determined with a Size Mobility Particle Sizer (SMPS) at the end of the flow tube. Aerosols were formed using a constant output atomiser (TSI, 3076) and the aerosol concentration and hence surface area could be varied,

being controlled using a high efficiency particulate air (HEPA) filter in a bypass arrangement. Atomiser solutions were prepared by dissolving 0.01 moles of ammonium sulphate (AS) (Fisher scientific, >99%) with varying amounts of copper (II) sulphate (Fisher scientific, >98%) in 500 mL of Milli-Q water. The data were analysed as described in George et al 2013. The pseudo first-order loss rate coefficient ($k'$) was obtained from the gradient of a plot of $\ln(HO_2$ signal) against the interaction time between $HO_2$ and the aerosol before sampling by the FAGE detector. The uptake coefficient ($\gamma(HO_2)$) was obtained from

the linear least-squares gradient of the plot of $k'$ against the surface area concentration of aerosols in the flow tube. The error given on all measurements of $\gamma(HO_2)$ represents $2\sigma$ of the uncertainty of the fitted gradient. A correction to $k'$ was applied to taking into account non-plug flow conditions in the flow tube using the Brown method.

[revised manuscript text omitted]

$$TR_{HO2uptake} = k_{uptake} \times [HO_2] \tag{20}$$

$$R_1 = \frac{TR_{HO2uptake}}{TR_{HOxsinks}} \tag{21}$$nighttime (b). The averaged

 $\gamma_{HO_2}$  daytime (08:00 −16:00) ROx radical loss rate is 6. ppbV/h and that for nighttime (16:00 −08:00 (+1d)) is 2.9 ppbV/h.

 No significant difference of $\gamma_{HO_2}$ is observed during daytime and night.

~~Uncertainty of the calculation in this paper mainly come from the measurement of copper concentration, radical concentration and aerosol liquid water content. The combined standrad uncertainty (uₜ) of the model calculations is a combination of uncertainties in the measurements used as model constraints and reaction rate constants. What's more, a series of tests based on Monte Carlo simulations show that the uncertainty of the model calculations is approximately 40% (for details, see Lu et al., 2012 and Tan et al., 2017).~~

|  |  |
|---|---|
|  |  |

| | |
|---|---|
|  |  |
|  |  |
|  |  |
|  |  |
|  |  |
|  |  |
|  |  |
|  |  |
|  |  |

 1090

$$\sim N = F(x, y, z, \dots) \tag{21}$$

$$\sim u_{r\_meas} = \frac{u_N}{N} = \sqrt{\left(\frac{\partial \ln F}{\partial x}\right)^2 (u_x)^2 + \left(\frac{\partial \ln F}{\partial y}\right)^2 (u_y)^2 + \left(\frac{\partial \ln F}{\partial z}\right)^2 (
[revised manuscript text omitted]

Anttila, T., Kiendler-Scharr, A., Tillmann, R., and Mentel, T. F.: On the Reactive Uptake of Gaseous Compounds by Organic-Coated Aqueous Aerosols: Theoretical Analysis and Application to the Heterogeneous Hydrolysis of N2O5, The Journal of Physical Chemistry A, 110, 10435-10443, 10.1021/jp062403c, 2006.

Baker, A. R., and Jickells, T. D.: Mineral particle size as a control on aerosol iron solubility, Geophysical Research Letters, 33, 10.1029/2006gl026557, 2006.

Berkemeier, T., Huisman, A. J., Ammann, M., Shiraiwa, M., Koop, T., and Poschl, U.: Kinetic regimes and limiting cases of gas uptake and heterogeneous reactions in atmospheric aerosols and clouds: a general classification scheme, Atmos. Chem. Phys., 13, 6663-6686, 10.5194/acp-13-6663-2013, 2013.

Berkemeier, T., Steimer, S. S., Krieger, U. K., Peter, T., Pöschl, U., Ammann, M., and Shiraiwa, M.: Ozone uptake on glassy, semi-solid and liquid organic matter and the role of reactive oxygen intermediates in atmospheric aerosol chemistry, Physical Chemistry Chemical Physics, 18, 12662-12674, 2016.

Bian, Y. X., Zhao, C. S., Ma, N., Chen, J., and Xu, W. Y.: A study of aerosol liquid water content based on hygroscopicity measurements at high relative humidity in the North China Plain, Atmos. Chem. Phys., 14, 6417-6426, 10.5194/acp-14-6417-2014, 2014.

Bielski, B. H., Cabelli, D. E., Arudi, R. L., and Ross, A. B.: Reactivity of $HO_2/O_2^-$ radicals in aqueous solution., Journal of physical and chemical reference data, 14, 1041-1100, 1985.

Capps, S., Henze, D., Hakami, A., Russell, A., and Nenes, A.: ANISORROPIA: the adjoint of the aerosol thermodynamic model ISORROPIA, Atmospheric Chemistry & Physics, 12, 2012.

Chameides, W. L., and Stelson, A. W.: Aqueous-phase chemical processes in deliquescent seasalt aerosols, Ber Bunsen Phys Chem, 96, 461-470, 1992.

Cheng, M. C., You, C. F., Cao, J. J., and Jin, Z. D.: Spatial and seasonal variability of water-soluble ions in PM2.5 aerosols in 14 major cities in China, Atmos. Environ., 60, 182-192, 10.1016/j.atmosenv.2012.06.037, 2012.

Clegg, S. L., Brimblecombe, P., and Wexler, A. S.: Thermodynamic model of the system $H^+-NH_4^+-SO_4^{2-}-NO_3^--H_2O$ at tropospheric temperatures, Journal of Physical Chemistry A, 102, 2137-2154, 10.1021/jp973042r, 1998.

Cooper, P. L., and Abbatt, J. P. D.: Heterogeneous interactions of OH and $HO_2$ radicals with surfaces characteristic of atmospheric particulate matter, J Phys Chem-Us, 100, 2249-2254, Doi 10.1021/Jp952142z, 1996.

Davidovits, P., Kolb, C. E., Williams, L. R., Jayne, J. T., and Worsnop, D. R.: Mass accommodation and chemical reactions at gas-liquid interfaces, Chemical Reviews, 106, 1323-1354, 10.1021/cr040366k, 2006.

[revised manuscript text omitted]

---

## Author Response (AR3)

**Response to the comments of reviewer #4**

This review covers the 2nd revised version of the manuscript ACP-0218

"Influence of aerosol copper on HO2 uptake: A novel parameterized equation" by Huang et al

We thank the reviewer for the helpful comments. The referee's comments are first given in black type, followed by our response to each in turn in blue type. Any changes to the manuscript in response to the comments are then given in quotation marks in red type.

The first author of this manuscript is Huan Song, not Huang.

This manuscript reports an explicit model of uptake of HO2 radicals to deliquesced inorganic aerosol particles to partially reconcile previous inconsistencies among experimental measurements and parameterizations. This includes a detailed treatment of the aqueous phase chemistry of HO2 and superoxide with Cu ions, as well as considering effects of Setchenov salting and ionic strength. The model is also used to interpret data from a field campaign. The previous review rounds already covered the general aspects regarding scientific relevance and significance the topic, which are undoubted. In response to previous comments related to a dataset also included to fit a parameterization, the authors of that dataset have joined the revised version of the paper and this version now includes an updated description of experimental details and revised data analysis related to that dataset. In response to another review, the authors have expanded the model description and the discussion related to sensitivity and uncertainty. Overall, this work provides valuable new information, especially the fact that taking into account the properties of the concentrated solutions of deliquesced aerosol particles allows to calculate HO2 uptake coefficients using known and well documented aqueous peroxy radical chemistry involving Cu ions. While the model includes some complexity by involving links to nitrogen oxides and and sulfur, it lacks inclusion of the interaction with the Fe(II)/Fe(III) redox couple, which usually is associated with the presence of Cu in atmospheric particles and may have important impacts. Nevertheless, the manuscript provides progress in understanding uptake of HO2 to aerosol particles.

In spite of this being the 2nd revised version of this manuscript, still a number of deficiencies exist that should be addressed. In the comment below, I list the line numbers of the pdf file of the revised manuscript version. In principle, these concerns are rather minor in character, but still numerous. They should be addressed before the manuscript may be accepted for ACP.

1) Language: the new text additions are sometimes misleading, mostly due to deficiencies in the English language. This should be fixed by a thorough work-through by the authors.

2) Abstract line 24: the IUPAC website provides a recommendation for a number of different deliquesced aerosol systems, not only cloud droplets. See also comment further below on the same aspect in the manuscript.

We made the modifications to this aspect in the manuscript. In the abstract line 24, we deleted "for cloud droplets". On line 253, we change the words to "deliquesced aerosol particles" and on line 262, we added the words "proposed for $HO_2$ uptake

35  for deliquesced aerosol particles" as recommended by the (24) opinion.

And the classical parameterized equation was confirmed by researches of reactive gas molecular uptake on dilute solution droplets (Magi et al., 1997) and on aqueous surfaces (Utter et al., 1992; Müller and Heal, 2002; Hu et al., 1995).

3) Line 26: not sure whether it is useful to have the parameterization in the abstract without explanation of symbols.

40  We added an instruction in the double brace in the abstract saying the explanations for the symbols are in the Appendix.

4) Line 27: The Wangdu campaign is not something the reader understands without introduction. Either explain in more detail, 'data from a campaign in the Wangdu region', or just 'data from a field campaign'.

We deleted the name of the field campaign on line 27.

45

5) Lines 42-45: these sentences should be split up, and the reason for the lower reactivity in absence of transition metals should be explained, including the self reaction of HO2 that has been parameterized by Thornton et al. In addition, a language issue here: The impact of HO2 uptake is not depending on a parameterization. It is the model output, or the calculated response of some parameters to HO2 uptake that changes. This is different.

50  We changed this part as:

"The model results of $HO_2$ uptake influence is lowered when a parameterized equation of $\gamma_{HO_2}$ is used without considering the influence of transition metal ions (TMIs) (Thornton et al., 2008). The reasons for the lower reactivity in absence of TMI including the lower reaction rate of aqueous $HO_2$ /$O_2^-$ reactions(Thornton et al., 2008). However , in spite of the lower $HO_2$ uptake coefficient used a significant impact on the calculated [OH] and $O_3$ production rate are suggested for air masses over

55  Chinese megacity areas (Macintyre and Evans, 2011)."

6) Line 73: symbols used in eq. (1) and (2) need to be explained

We added an instruction in the first of the paragraph saying the explanations for the symbols are in the Appendix.

60  7) Same paragraph: it does not become clear enough how the uptake coefficient was retrieved from the model output. This should be briefly described here.

We changed the sentences on line 73 as:

"The model directly calculates the averaged quasi-first order gas phase $HO_2$ uptake loss rate at steady state, $k_{het}$(s$^{-1}$), in Eq. (1). $\gamma_{HO_2}$ is retrieved by Eq.(2) considering the influence of aerosol liquid water content (ALWC) [g cm$^{-3}$] rather than surface

65  density because of the influence of the $RH$ on the uptake process (Kuang et al., 2018; Bian et al., 2014). "

8) Line 106: some numbers should be given here. What is the ratio between H^cc and H_0 over the RH range considered in this study? Same sentence: why does H^cc depend strongly on the Cu concentration? The dominant solute is ammonium sulfate, which should be the main driver of salting and ionic strength, isn't it.

We gave the ratio of $H^{cc}$ to $H_0$ of HO$_2$ in the revised MS as follows on the line 106:

"According to this correction, $H^{cc}$ of HO$_2$ increases with *RH*. The ratio of $H^{cc}$ to $H_0$ ranges from 0.03 (40% *RH*, *I*=16.7) to 0.34 (80% *RH*, *I*=5.5). Although the slating effect and the ionic strength are mainly driven by [NH$_4^+$] and [SO$_4^{2-}$], ionic strength increases quickly from 5.9 to 9.5 with [Cu$^{2+}$] from 0.1 M to 1 M and limits the solubility of HO$_2$ gas molecules."

9) Line 115: Two things here: the solubility of Fe is not a defined quantity, the authors may refer to the solubility of Fe-containing minerals. There are literature reports about what fraction of Fe is typically in dissolved form. In addition, there is a language issue here (many similar cases throughout the manuscript), the sentence reads like the solubility of Fe is related to its ratio to Cu, which is certainly not true.

10) Line 118: Neglecting the presence of Fe should lead to a stronger caveat for this work. The authors cite the Mao et al. (2013) work a few times, which clearly indicates a strong impact of the Cu/Fe ratio on the fate of HO2 products in the aqueous phase. Fe may reduce the contribution of recycling of peroxy radicals to lower the effective HO2 loss rate, and Fe might be relevant for the interpretation of the effect of HO2 uptake on the HOx budget in conjunction with the field data.

There is no direct evidence of the impact of the Cu/Fe ratio on the fate of HO2 products as proposed by Mao, so in this MS we used the model mechanisms for HO2 uptake to produce H2O as many other papers (Mozurkewich et al., 1987; Hanson et al., 1992; Thornton and Abbatt, 2005; Thornton et al., 2008; Taketani et al., 2009; Macintyre and Evans, 2011). On line 118, we change this part as:

"Since there is no direct evidence of the existence of Cu/Fe redox reactions of HO$_2$ which produce H$_2$O rather than H$_2$O$_2$ as proposed by (Mao et al., 2013), in the scope of this paper, HO$_2$ uptake produce H$_2$O separately by Cu and Fe free ions as proposed by many researches (Mozurkewich et al., 1987; Hanson et al., 1992; Thornton and Abbatt, 2005; Thornton et al., 2008; Taketani et al., 2009; Macintyre and Evans, 2011) and the mechanism summarized by IUPAC. Fe free ions can be seen as the equivalent Cu free ions in the application of the MARK model or the parameterized equation mentioned below."

11) Line 135: in eq. (10) and (11), the subscript 'equ' is misleading, since it should refer to 'effective' and not 'equilibrium' or 'equation'. So 'eff' would be better.

We changed 'equ' to 'eff' in this part and updated the meaning of $[Cu^{2+}]_{eff}$ which is now the sum of effective copper concentration and other TMI equivalent copper concentrations.

12) Line 148: this paragraph is not sufficiently clear. k_eff seems to be the apparent first order loss rate coefficient of HO2 in the aqueous phase (not involving solubility). Typical numbers should be provided; otherwise it would not be understandable,

why a 1 M Cu(II) solution would not lead to a very short reacto-diffusive length and thus strong concentration gradients in HO2. Also the reasons for the apparently low values should be explained, as this must result from complex recycling occurring, since the first reaction of Cu(II) with HO2 is very fast.

$k_{eff}$ is the comprehensive liquid phase reaction rate coefficient which encompasses both $HO_2$ dissolution equilibrium reactions and liquid phase chemical-physical reactions during $HO_2$ uptake process. It is calculated by Eq.19 and include the influence of the salting out effect of $HO_2$. Typical value is about $2.9 \times 10^6$ M$^{-1}$ s$^{-1}$ with 1 M Cu$^{2+}$ and $3.25 \times 10^4$ M$^{-1}$ s$^{-1}$ with 0.001M Cu$^{2+}$. $k_{eff}$ will change dramatically according to the concentration of equivalent copper ions and the diameter of particles.

We changed the relevant sentences and recalculated the ratio considering the high copper concentration on line 148 and following part:

"and $D_{aq}$ is the aqueous phase diffusion coefficient [cm$^2$s$^{-1}$], $k_{eff}$ is the comprehensive liquid phase reaction rate coefficient which encompasses both $HO_2$ dissolution equilibrium reactions and liquid phase chemical-physical reactions during $HO_2$ uptake process. $k_{eff}$ is calculated by Eq.19 and includes the influence of the salting out effect of $HO_2$ and the ionic strength effects on TMI. $k_{eff}$ will change dramatically according to the concentration of equivalent copper ions and the diameter of particles. Higher Cu concentration will make the ratio smaller and cause larger uncertainties, however, in the copper-doped aerosol particle, because of the high value of $k_{eff}$ (typical value is $2.9 \times 10^6$ M$^{-1}$ s$^{-1}$ with 1 M Cu$^{2+}$ and $3.25 \times 10^4$ M$^{-1}$ s$^{-1}$ with 0.001 M Cu$^{2+}$ ) and small Count Median Diameter ($R_d$) (usually smaller than 1 μm), the ratio $\frac{[\overline{HO_2}]}{[HO_{2(r)}]}$ is close to 1. At a diameter of 100nm, and a relative humidity between 40% and 90%, the condensed phase copper ion concentration varies from $10^{-5}$ to 1 M, the average ratio of the surface $HO_2$ concentration and the condensed phase $HO_2$ concentration is beyond 0.87 at every Cu concentration. The ratios are calculated by simulation of $k_{eff}$ and the accordingly calculations by Equation (12) and (13). Thus, in this model, we assume the surface concentration of $HO_2$ equals to the condensed phase average $HO_2$ concentration."

13) Line 158: the value of the accommodation coefficient is not related to the amount of Cu. The amount of Cu may control the uptake regime, with large amounts leading to uptake becoming accommodation limited. But the value of alpha is independent of Cu, unless it is involved in the process of surface to bulk transfer of HO2. Are the authors confusing accommodation coefficient with uptake coefficient? Also Table 1 is misleading, as the data in column seem to be uptake coefficient, as indicated in the Table caption, but inconsistent with the column header.

14) Line 162: probably related to the previous, the high accommodation coefficient for HO2 does not automatically mean it has a high loss rate. It only means that gamma may get large, if a strong sink is available in the condensed phase.

15) Line 164: language: The MARK model probably does not make the selection of alpha, but the authors selected it.

16) Line 167: as mentioned before, please clarify the column header of the table and the caption (alpha or gamma)

The accommodation coefficient is surely not influenced by the concentration of Cu ions if the viscosity of particles maintains. While the accurate accommodation coefficient can be only measured with no limitation of aqueous mass transfer flux in which

situation, a high aqueous reaction rate. We changed this part in section 2.3 to avoid misleading. We also changed the column header.

135 "The accommodation coefficient of $HO_2$ ($\alpha_{HO_2}$) is independent of the concentrations of free Cu ions if the viscosity of particles maintains. While the more accurate accommodation coefficient can be only measured with no limitation of aqueous mass transfer flux, in which situation, $\alpha_{HO_2}$ equals to $\gamma_{HO_2}$. $HO_2$ uptake coefficients are summarized for copper-doped inorganic aerosol particles from various previous laboratory studies. The uptake coefficient of $HO_2$ is approximately 0.5 in sulfate aerosol and even higher for chlorine or nitrate aerosol because of the catalytic effect of $Cu^{2+}$ on aqueous $HO_2/O_2^-$ (Table 1). In this

140 situation, the aqueous reactions are fast enough for the uptake process be limited primarily by the mass transport process (accommodation) and the uptake coefficient equals to the accommodation coefficient. Thus, the MARK model typically uses $\alpha_{HO_2}$ as 0.5. We also tested the influence of the accommodation coefficient on calculated $HO_2$ uptake coefficient in a field campaign, details please see the Supplementary Information."

145 17) Line 172: first sentence: language!
We changed the word as "included".

18) Line 178: SMPS = Scanning Mobility Particle Sizer or Scanning Mobility Particle Spectrometer
19) Line 179: aerosol particles were produced using a …

150 We changed this part as:
"…and the total aerosol surface area was determined with a Scanning Mobility Particle Sizer (SMPS) at the end of the flow tube. Aerosol particles were produced using a constant output atomizer…"

20) Line 198 and following: while the model is indeed used to explore the RH dependence of HO2 uptake and associated

155 aqueous phase chemistry, the comparison to experimental data is not really covering a substantial RH range. So that is essentially limited to the effect of the Cu(II) concentration.
At present, there are experimental measurements of $\gamma_{HO_2}$ at different RH but there is no experimental systematic study of this dependence where only RH is changed. Many researches proposed that $\gamma_{HO_2}$ is higher for aqueous inorganic aerosol than for dry inorganic aerosol. Although the previous experiments did not directly measure the RH´s dependence, the change of the

160 experimental uptake coefficients met the simulation trend (see Figure 2). Ambient RH would affect the activity coefficients of reactant ions in the aerosol particle condensed phase and the solubility of the gas phase reactant such as OH, $HO_2$ and $H_2O_2$. The MARK model was presented to simulate the change of the uptake coefficient with RH in Figure 1.

21) Line 230: HO2 uptake at low Cu content: the figure should be plotted in log y-scale to demonstrate that the remaining

165 uptake at low Cu content is driven by self reaction of HO2 (should be second order in HO2).

The aim of this MS is to explore the influence of TMI and RH on $HO_2$ uptake rather than $HO_2$ aqueous self-reaction mechanism. For this reason, we believe Figure 2 is better now to demonstrate the trend of $HO_2$ uptake coefficient under different copper ions gradient.

170    22) Line 231: What does the sentence 'The threshold is also consist in …' mean? What are droplets?

On line 231 the word should be 'consistent' and the droplets refer to cloud or rain droplets.

23) Line 233: I understand that increasing Cu content drives uptake to the accommodation limit; but why should that be determined by the solubility?

175    We changed the last sentence in the paragraph: "As the copper concentration increasing, the solution ionic strength increases and $\gamma_{HO_2}$ rapidly rises to the limit of the accommodation coefficient and the limitation of the $HO_2$ solubility."

24) Line 254: the Thornton et al. (2008) parameterization was developed for deliquesced aerosol particles, not cloud droplets. This repetitively comes up below again.

180    25) Line 262 and following: as mentioned above, the Thornton et al. and IUPAC recommendation has not been suggested for dilute aqueous droplets but actually for deliquesced aerosol particles. As discussed in Thornton et al. but also in the comments accompanying the IUPAC recommendations, the fact that k_TMI is lower than the actual known rate coefficient of HO2 or O2- with Cu(II), is assumed to likely result from the combined effects of solute strength effects. It is indeed the added value of this work to make this aspect more quantitative. The authors could emphasize this somewhat more to detail the individual

185    contributions of Setchenov salting, of ionic strength and of the recycling efficiency among the cupper and peroxy species to the reduction of the effective rate coefficient. The advantage of eq 15 is that it correctly represents the transition between the reacto-diffusive regime (when k_eff is higher) to the homogeneous bulk reaction regime covered by the parameterization suggested in this work (eqs 18-21)

Corrections about the word "droplets" to "deliquesced aerosol particles" are done in the revised MS mentions in the response

190    to opinion (2).

We deleted the discussion of the rate constant of $Cu^{2+}$ with $HO_2/O_2^-$ and added the sentence:

"The low reaction rates used here in the *CEq.* are assumed to likely result from the combined effects of solute strength effects as discussed by Lakey et al. (2016)."

The individual contributions are mentioned in the last paragraph of Section 3.3 as:

195    "The MARK model uses the same framework with the *CEq* and considers the Setchenov salting out and ionic strength effects on HO2 uptake more comprehensively and detailly and proposes $k_{eff}$ as the effective reaction coefficient (Eq.19). Considering the small RMSE between the MARK model and the laboratory studies, we proposed a novel parameterized equation (*NEq.*) to better describe the influence of $[Cu^{2+}]$ and *RH* on $\gamma_{HO_2}$.

"

200

26) Line 310 and following: the authors should clearly state that this parameterization is only reasonable as long k_eff remains sufficiently small, such that no HO2 gradients within particles develop. While they seem to show that this is valid with the mechanism involving Cu only, it is not granted that this is still true when for instance including Fe ions in the mechanism which could leading to an increasing sink for HO2 if the recycling efficiency is shut off; similar effects may occur in presence

205 of organics.

In the section 3.5.4 the MS discussed the possible sources of uncertainties of the novel equation including the approximate calculation of $HO_2$ concentration gradients and the influence of organics. We added the words: "or high copper equivalent concentration." in the first paragraph in this section.

About the influence of Fe ions is answered above.

[revised manuscript text omitted]